

# Intercomparison of Antarctic ice shelf, ocean, and sea ice interactions simulated by two models

Kaitlin A. Naughten[1,2,3], Katrin J. Meissner[1,2], Benjamin K. Galton-Fenzi[4,3], Matthew H. England[1,2], Ralph Timmermann[5], Hartmut H. Hellmer[5], Tore Hattermann[6,5], and Jens B. Debernard[7]

[1]Climate Change Research Centre, Level 4 Mathews Building, University of New South Wales, Sydney NSW 2052, Australia
[2]ARC Centre of Excellence for Climate System Science, Australia
[3]Antarctic Climate & Ecosystems Cooperative Research Centre, Private Bag 80, Hobart TAS 7001, Australia
[4]Australian Antarctic Division, 203 Channel Highway, Kingston TAS 7050, Australia
[5]Alfred Wegener Institut, Postfach 12 01 61, 27515 Bremerhaven, Germany
[6]Akvaplan-niva, P.O. Box 6606, Langnes, 9296 Tromsø, Norway
[7]Norwegian Meteorological Institute, P.O. Box 43, Blindern, N-0313 Oslo, Norway

*Correspondence to:* Kaitlin A. Naughten (k.naughten@unsw.edu.au)

**Abstract.** An increasing number of Southern Ocean models now include Antarctic ice shelf cavities, and simulate thermodynamics at the ice-shelf/ocean interface. This adds another level of complexity to Southern Ocean simulations, as ice shelves interact directly with the ocean and indirectly with sea ice. Here we present the first published model intercomparison and evaluation of present-day ocean/sea-ice/ice-shelf interactions, as simulated by two models: a circumpolar Antarctic config-
uration of MetROMS (ROMS: Regional Ocean Modelling System coupled to CICE: Community Ice CodE) and the global model FESOM (Finite Element Sea-ice/ice-shelf Ocean Model), where the latter is run at two different levels of horizontal resolution. From a circumpolar Antarctic perspective, we compare and evaluate simulated ice shelf basal melting and sub-ice shelf circulation, as well as sea ice properties and Southern Ocean water mass characteristics as they influence the sub-ice shelf processes. Despite their differing numerical methods, the two models produce broadly similar results, and share similar
biases in many cases. Both models reproduce many key features of observations, but struggle to reproduce others, such as the high melt rates observed in the small warm-cavity ice shelves of the Amundsen and Bellingshausen Seas. Several differences in model design show a particular influence on the simulations. For example, FESOM's greater topographic smoothing can alter the geometry of some ice shelf cavities enough to affect their melt rates; this improves at higher resolution, since less smoothing is required. In the interior Southern Ocean, the vertical coordinate system affects the degree of water mass erosion due to spurious diapycnal mixing, with MetROMS' terrain-following coordinates leading to more erosion than FESOM's z-
coordinates. Finally, increased horizontal resolution in FESOM leads to higher basal melt rates for small ice shelves, through a combination of stronger circulation and small-scale intrusions of warm water from offshore.

## 1 Introduction

The Antarctic Ice Sheet (AIS) has significant potential to drive sea level rise as climate change continues (Deconto and Pollard,
2016; Golledge et al., 2015; Rignot et al., 2014; Mengel and Levermann, 2014). Paleo records indicate that the AIS was a




major contributor to sea level change in past climate events (Cook et al., 2013; Miller et al., 2012; Raymo and Mitrovica, 2012; Dutton et al., 2015; O'Leary et al., 2013), and the mass balance of the modern-day AIS is already negative (Rignot et al., 2011; Zwally and Giovinetto, 2011). The rate of retreat of much of the AIS will be governed by the ocean (Golledge et al., 2017), as 40% of the ice sheet by area is grounded below sea level (Fretwell et al., 2013). This geometry provides the potential for

the ocean to melt large regions of the AIS from below. For example, the Amundsen sector of West Antarctica has bedrock geometry favourable for a marine ice sheet instability, and unstable retreat may have already begun (Rignot et al., 2014).

The ocean directly interacts with the AIS through ice shelves, which are the floating extensions of the land-based ice sheet. The properties of ice shelf cavities, the pockets of ocean between ice shelves and the seafloor, determine the basal melt rates of each ice shelf which ultimately affect the mass balance of the AIS through dynamical processes (Dupont, 2005). The seawater

in ice shelf cavities can be sourced from several different water masses, which affect its temperature and salinity. Many of these source water masses are influenced by sea ice processes (Jacobs et al., 1992; Nicholls et al., 2009).

A better understanding of ocean/sea-ice/ice-shelf interactions in Antarctica is crucial, particularly given their importance for future sea level rise. However, these interactions take place in observation-deficient regions. In particular, there are very few direct measurements inside ice shelf cavities, and observations are also scarce in the sea ice covered regions of the Southern

Ocean (Rintoul et al., 2010). While ice shelf basal melt rates can be inferred using remote sensing methods (Rignot et al., 2013; Depoorter et al., 2013), large uncertainties remain regarding the circulation patterns driving these melt rates, and no predictions for the future can be made based on these data.

Consequently, much of our understanding of ocean/sea-ice/ice-shelf interactions is based on numerical modelling. In recent years an increasing number of ocean models have begun to resolve ice shelf cavities and simulate thermodynamic processes

at the ice shelf base (Dinniman et al., 2016). Given the variety of models involved, and the relative lack of observations to constrain their tuning, it is desirable to conduct model intercomparison projects (MIPs, see e.g. Meehl et al. (2000)) by which several models run the same experiment and their output is compared. The resulting insights into model similarities and differences can ideally be attributed to model design choices, with the aim of guiding future development.

To date, the only MIPs considering ice shelf cavities are the ongoing ISOMIP experiments (Ice Shelf-Ocean Model In-

tercomparison Project) (Hunter, 2006; Asay-Davis et al., 2016) which use idealised domains and simplified forcing, and do not include coupled sea ice models. The ISOMIP experiments are undoubtedly valuable, and are likely to provide particular insights regarding the response of cavity circulation to warm versus cold forcing. However, idealised experiments such as ISOMIP should be complemented by intercomparisons over more realistic domains, with observationally derived forcing and coupled sea ice models. These model configurations are already being used to better understand processes in observed cavities

(Timmermann et al., 2012; Galton-Fenzi et al., 2012) and to provide future projections of ice shelf melt (Timmermann and Hellmer, 2013; Hellmer et al., 2012, 2017), so analysis of the similarities and differences between such models is timely. Another important benefit of realistic domains is the opportunity to compare model output to available observations, even if these observations are limited. Therefore, an element of model evaluation, as well as model intercomparison, can be included.

In this paper we present such an intercomparison of two ocean models, both including ice shelf thermodynamics and sea ice

components, from a circumpolar Antarctic perspective. We focus on ice shelf basal melt and sub-ice shelf circulation across





eight regions of the Antarctic coastline, but also consider interior Southern Ocean and sea ice processes as they affect ice shelf cavities. The model output is compared to relevant observations where available. Finally, key findings and their implications, as well as possibilities for future model development, are discussed.

## 2 Model descriptions

5  Two coupled ocean/sea-ice/ice-shelf models are included in this intercomparison: MetROMS and FESOM. We run FESOM at two different resolutions, for a total of three experiments (see Section 3). In this section we describe the two models and compare their scientific design.

### 2.1 Overview

MetROMS consists of the regional ocean model ROMS (Regional Ocean Modelling System) (Shchepetkin and McWilliams, 10  2005) including ice shelf thermodynamics (Galton-Fenzi et al., 2012), coupled to the sea ice model CICE (Community Ice CodE) (Hunke et al., 2015) using the coupler MCT (Model Coupling Toolkit) (Larson et al., 2005; Jacob et al., 2005). The coupling was implemented by the Norwegian Meteorological Institute (Debernard et al., 2017) and is described in Naughten et al. (2017). We use the development version 3.7 of the ROMS code, version 5.1.2 of CICE, and version 2.9 of MCT.

FESOM (Finite Element Sea-ice/ice-shelf Ocean Model) is a global ocean model with an internally coupled sea ice model 15  (Danilov et al., 2015; Timmermann et al., 2009) and ice shelf thermodynamics (Timmermann et al., 2012). It has an unstructured mesh in the horizontal, consisting of triangular elements which allow for spatially varying resolution. The numerical methods associated with the unstructured mesh are detailed by Wang et al. (2008) and Wang et al. (2014), while the implementation of the ice shelf component is discussed in Timmermann et al. (2012).

### 2.2 Domain and resolution

20  Our configuration of MetROMS has a circumpolar Antarctic domain with a northern boundary at $30°$S. Horizontal resolution is quarter-degree scaled by cosine of latitude, and the South Pole is relocated to achieve approximately equal resolution around the Antarctic coastline. This leads to resolutions (defined as the square root of the area of each grid box) of approximately 15-20 km in the Antarctic Circumpolar Current (ACC), 8-10 km on the Antarctic continental shelf, and 5 km or finer at the southernmost grounding lines of the Ross, Filchner-Ronne, and Amery Ice Shelves (Figure 1a).

25  Our FESOM setup has a global domain with spatially varying horizontal resolution. Here we define resolution in FESOM as the square root of the area of each triangular element, however this metric may not be truly comparable to MetROMS. When discussing resolution, the real question is how many features of fluid flow can be represented by a mesh of a certain spacing. In models with such different numerical methods as MetROMS (finite-volume) and FESOM (finite-element), the number of features resolved may scale differently with the mesh spacing. Numerical dissipation and stabilisation built into different time-30  stepping routines can also influence this "effective resolution". Furthermore, MetROMS employs a staggered Arakawa C-grid for the ocean and B-grid for the sea ice (Arakawa and Lamb, 1977), by which different variables are calculated at different



locations within each grid box. In FESOM, all variables are calculated at the same locations (nodes), analogous to the Arakawa A-grid. There is some evidence that this design tends to resolve fewer features of fluid flow (Haidvogel and Beckmann, 1999), and indeed FESOM appears to have a lower effective resolution than finite-difference C-grid models with comparable nominal grid spacing. These differences should be lessened in eddy-resolving simulations, where advection of momentum (which is

essentially the same in staggered and unstaggered meshes) dominates. However, the simulations presented here are not eddy-resolving, so some influence would be expected.

To account for these uncertainties, as well as to investigate the importance of resolution on FESOM's performance, we have prepared two meshes: "low-resolution" (Figure 1b) and "high-resolution" (Figure 1c). The high-resolution mesh has approximately double the number of 2D nodes as the low-resolution mesh, but these extra nodes are not evenly spaced throughout the

domain. Outside the Southern Ocean, the two meshes have virtually identical resolution (not shown), ranging from 150-225 km in the abyssal Pacific, Atlantic, and Indian Oceans, and 50-75 km along coastlines. In the open Southern Ocean, resolution ranges from 20-100 km for the low-resolution mesh and 15-50 km for the high-resolution mesh. Both meshes have finer resolution on the Antarctic continental shelf (approximately 8-10 km for low-resolution, 5-7 km for high-resolution) and in ice shelf cavities (5-10 km for low-resolution, 3-7 km for high-resolution). The greatest difference between the two meshes occurs

in the Amundsen and Bellingshausen Seas, with approximate resolution of 11 km for the low-resolution mesh and 4 km for high-resolution.

In the vertical, MetROMS uses 31 terrain-following levels using the s-coordinate system, with increasing vertical resolution near the surface and bottom, and coarsest resolution in the interior. FESOM employs a hybrid z-sigma vertical coordinate system, with the same discretisation for both the low- and high-resolution meshes. The region south of the 2500 m isobath

surrounding Antarctica, which includes all ice shelf cavities as well as the continental shelf and slope, has sigma-coordinates with 22 levels. The remainder of the domain has z-coordinates, comprised of 38 levels weighted towards the surface. Bottom nodes are allowed to deviate from the standard z-levels in order to match the given bathymetry. Both models are free-surface, which leads to time-varying vertical levels in MetROMS, but only affects the uppermost layer in FESOM.

In both models, the use of terrain-following coordinates in the thin water columns of ice shelf cavities leads to extremely

high vertical resolution, often finer than 1 m in MetROMS, which limits the timestep. Our configuration of ROMS requires a baroclinic timestep of 5 minutes for stability, with 30 barotropic timesteps for each baroclinic. In CICE, the timestep is 30 minutes for both dynamic and thermodynamic processes, and ocean/sea-ice coupling is also performed every 30 minutes. FESOM is run with a timestep of 10 minutes for the low-resolution mesh and 9 minutes for high-resolution. The sea ice model operates on the same time step as the ocean component.

**2.3 Smoothing of bathymetry and ice shelf draft**

Steep bathymetry can be problematic for terrain-following coordinate ocean models, as it has the potential to introduce pressure gradient errors (Haney, 1991). Both ROMS (Shchepetkin, 2003) and FESOM (Wang et al., 2008) are designed to minimise this issue with the splines density Jacobian method for the calculation of the pressure gradient force, which reduces errors compared to the standard density Jacobian method. Nevertheless, a particular challenge arises at ice shelf fronts, which in





reality are cliff faces that can reach several hundred metres in depth, but which models must represent as sloping surfaces. This substantial change in surface layer depth over as little as one grid cell creates steeply sloping vertical layers with a large pressure gradient, and numerical errors in the pressure gradient calculation could drive spurious circulation patterns across the given ice shelf front.

In both models, some amount of smoothing of the bathymetry and ice shelf draft is necessary for numerical stability and to reduce pressure gradient errors. On the other hand, excessive smoothing could alter the geometry of the ice shelf cavities to the point where circulation is affected. An oversmoothed ice shelf front would be too shallow and gently sloping, providing a pathway for warm surface waters to easily enter the cavity, where in reality a physical barrier exists. Near the grounding lines at the back of ice shelf cavities, oversmoothing would remove the deepest ice which melts most easily (Lewis and Perkin,
1986). In this situation the water column thickness would be overestimated, allowing for greater transport of warm water to the grounding line. In a coupled ice-sheet/ocean model, Timmermann and Goeller (2017) demonstrated that increased water column thickness due to a thinning ice shelf can more than compensate for the reduced melting expected from the elevated in-situ freezing point at the ice shelf base. Therefore, a delicate balance must be struck when smoothing model topographies, in order to achieve the most accurate simulation.

We prepared the MetROMS and FESOM domains using bathymetry, ice shelf draft, and land/sea masks from the RTopo-1.05 dataset (Timmermann et al., 2010). MetROMS follows a 3-step smoothing procedure similar to that of Lemarié et al. (2012). First, the "deep ocean filter" consists of a single pass of a Hanning filter (window size 3) over the bathymetry $h$, with variable coefficients designed to remove isolated seamounts. Next, a selective Hanning filter is repeatedly applied to both $\log(h)$ and $\log(z_{ice})$, where $z_{ice}$ is the ice shelf draft, until the slope parameter $r = (h_i - h_{i+1})/(h_i + h_{i+1})$ satisfies the
condition $r < 0.25$ everywhere (and similarly for $z_{ice}$). This selective filter has coefficients scaled by the gradient of $h$ or $z_{ice}$, meaning that regions which are already smooth enough will not become oversmoothed. Finally, both $h$ and $z_{ice}$ undergo a final two passes of a regular Hanning smoother to remove 2D noise. Note that this separate treatment of bathymetry and ice shelf draft does not directly consider water column thickness, for which some large gradients may remain.

The smoothing procedure in FESOM is the same as described by Nakayama et al. (2014). The source bathymetry and
ice shelf draft are first averaged over 4-minute ($\frac{1}{15}^{\circ}$) intervals. Then Gaussian filters are applied to both fields, with spatially-varying radii scaled by the desired final resolution. For this reason, high-resolution regions of the domain receive less smoothing than lower-resolution regions. The ice shelf draft undergoes one pass of the Gaussian filter, while the bathymetry undergoes four passes with larger radii. Following interpolation to the unstructured mesh, the ice shelf draft field receives selective smoothing to satisfy the critical steepness limitation of Haney (1991) at all points. This procedure limits the slope of the ice shelf draft,
and extremely high resolution may be necessary to preserve steep slopes.

Another region of concern is the grounding line, where water column thickness approaches zero and vertical layers converge. Estimates of pressure gradient error, such as that of Haney (1991), scale inversely with the vertical layer thickness and can therefore diverge near the grounding line. To alleviate this problem, a minimum water column thickness of 50 m is enforced. In both models, the bathymetry is artificially deepened where necessary to satisfy this condition during the smoothing process.





## 2.4 Ocean mixing

ROMS includes several options for tracer advection (Shchepetkin and McWilliams, 2005), and the choice of advection scheme is known to impact the simulation. The centered and Akima fourth-order tracer advection schemes are dominated by dispersive error, which can lead to undershoots of the freezing point and spurious sea ice formation in our MetROMS configuration

(Naughten et al., 2017). On the other hand, the upwind third-order tracer advection scheme is dominated by dissipative error, which can result in high levels of diapycnal mixing for some simulations (Lemarié et al., 2012; Marchesiello et al., 2009). Indeed, problematic diapycnal mixing related to the upwind third-order scheme was observed in decadal-scale simulations with our configuration of MetROMS (not shown). Therefore, the 25-year MetROMS simulation we present here uses the Akima fourth-order tracer advection scheme, combined with explicitly parameterised Laplacian diffusion applied along isoneutral

surfaces, at a level strong enough to smooth out most dispersive oscillations. This configuration shows very minimal spurious sea ice formation compared to a simulation with flux-limited (i.e. locally monotonic) upwind third-order advection (Naughten et al., 2017), and exhibits less spurious diapycnal mixing than the upwind scheme. The diffusivity coefficient is 150 m$^2$/s, which applies to the largest grid cell (approx. 24 km resolution) and is scaled linearly for smaller cells. Advection of momentum uses the upwind third-order scheme in the horizontal and the centered fourth-order scheme in the vertical (Shchepetkin and

McWilliams, 2005), and is combined with parameterised biharmonic viscosity along geopotential surfaces, with a coefficient of 10$^7$ m$^4$/s (scaled by grid size as with diffusivity).

FESOM computes advection of momentum using the characteristic Galerkin method, and advection of tracers using the explicit second-order flux-corrected-transport scheme (Wang et al., 2014). The Laplacian approach is used to explicitly parameterise both diffusivity and viscosity, with coefficients 600 m$^2$/s and 6000 m$^2$/s respectively. These values apply to a reference

area of 5800 km$^2$, and are scaled to the area of each triangular element, scaling with the square root for diffusivity and linearly for viscosity. At 10 km resolution (element area of 100 km$^2$), the resulting diffusivity is 78.8 m$^2$/s, compared to 62.4 m$^2$/s in ROMS. The analogous viscosity terms cannot be directly compared between ROMS and FESOM, since they do not use the same parameterisation.

A flow-dependent Smagorinsky viscosity term is also applied in FESOM (Smagorinsky, 1963, 1993; Wang et al., 2014).

In z-coordinate regions, tracer diffusion is rotated along isoneutrals, and the Gent-McWilliams eddy parameterisation is used (Gent and McWilliams, 1990; Gent et al., 1995; Wang et al., 2014). In sigma-coordinate regions, diffusivity and viscosity are both applied along constant-sigma surfaces.

For weakly stratified regions such as the Southern Ocean, the choice of vertical mixing parameterisation can have a significant impact on simulated convection (Timmermann and Beckmann, 2004). MetROMS employs the Large-McWilliams-Doney

interior closure scheme (Large et al., 1994) which includes the KPP boundary layer parameterisation. We implement the same KPP modification as in Dinniman et al. (2011), which imposes a minimum surface boundary layer depth based on surface stress, in the case of stabilising conditions. This modification is designed to address problems with excessive stratification during periods of rapid sea ice melt, and follows principles similar to the FESOM vertical mixing parameterisation discussed





below. A shallow bias in mixed layer depths during the melt season is problematic for the accurate simulation of Southern
Ocean water masses, particularly in the Weddell Sea (Timmermann and Beckmann, 2004).

The vertical mixing scheme in our configuration of FESOM (Timmermann et al., 2009) consists of the Richardson number
dependent parameterisation of Pacanowski and Philander (1981), modified to have maximum vertical diffusivities and viscosi-

ties of 0.05 m²/s. Over a depth defined by the Monin-Obukhov length, calculated as a function of surface stress and buoyancy
forcing, an extra 0.01 m²/s is applied to both vertical diffusivity and viscosity. This combination was found by Timmermann
and Beckmann (2004) to produce the most realistic representation of water masses in the Weddell Sea, avoiding the excessive
open-ocean convection which is characteristic of traditional convective adjustment. We also tested the KPP parameterisation
(without the modification used by MetROMS) in short simulations with our FESOM configuration (not shown). Compared to

the modified Pacanowski-Philander scheme, this had no significant impacts on Weddell Sea convection, at least on the 5-year
timescale.

## 2.5   Ice shelf thermodynamics

With terrain-following coordinates, it is relatively straightforward to include ice shelf cavities in an Antarctic domain. In both
ROMS and FESOM, all of the terrain-following vertical layers subduct beneath the ice shelves. The pressure of the ice shelf

draft must be considered in the calculation of the pressure gradient. ROMS vertically integrates the density of water displaced
by ice, and assumes the density of this displaced water is a linear function of depth, with coefficient $\frac{\partial \rho}{\partial z} = 4.78 \times 10^{-3}$ kg/m⁴
and intercept given by the density in the first model layer. FESOM computes the pressure gradient force from the vertically
integrated horizontal density gradient and assumes that the horizontal pressure gradient is zero at the ice shelf base. High-order
interpolation for density is done in the vertical to compute horizontal density gradients as accurately as possible.

ROMS and FESOM simulate ice shelf thermodynamics: the heat and salt fluxes associated with melting and refreezing at the
ice shelf base. However, any net melting or freezing is not actually applied to the ice shelf geometry. It is assumed that glacial
flow of the ice shelf, surface accumulation, and basal melting are in dynamic equilibrium such that the geometry remains
constant. Removing this assumption necessitates coupling with an ice sheet model, which has recently been accomplished
for FESOM (Timmermann and Goeller, 2017) and is under development for ROMS (Gladstone et al., 2017). Ice-sheet/ocean

coupling is an emerging field of climate modelling, and the first generation of models will be compared and evaluated as part
of the MISOMIP experiments (Marine Ice Sheet-Ocean Model Intercomparison Project) (Asay-Davis et al., 2016).

Both ROMS and FESOM (Galton-Fenzi, 2009; Galton-Fenzi et al., 2012; Timmermann et al., 2012) implement the 3-
equation parameterisation of Hellmer and Olbers (1989) refined by Holland and Jenkins (1999). The heat and salt exchange
coefficients $\gamma_T$ and $\gamma_S$ have the form

$$\gamma_T = \frac{u^*}{\kappa + 12.5\,\mathrm{Pr}^{\frac{2}{3}} - 6} \qquad \text{and} \qquad \gamma_S = \frac{u^*}{\kappa + 12.5\,\mathrm{Sc}^{\frac{2}{3}} - 6} \tag{1}$$





where Pr is the Prandtl number and Sc is the Schmidt number (both dimensionless constants) and $u^*$ is the friction velocity, calculated as

$$u^* = \max\left(\sqrt{C_d\left(u^2 + v^2\right)}, u^*_{min}\right) \tag{2}$$

where $C_d$ is the drag coefficient ($3 \times 10^{-3}$ in ROMS, $2.5 \times 10^{-3}$ in FESOM), $u$ and $v$ are the horizontal ocean velocity
components in the uppermost vertical layer, and $u^*_{min}$ is a lower bound for $u^*$ which represents molecular diffusion ($10^{-3}$ in
ROMS, $2.5 \times 10^{-4}$ in FESOM). While the effect of the different drag coefficient between the models is likely to be negligible,
the larger minimum $u^*$ in ROMS will cause stronger melting in locations with very weak flow, such as at the grounding line
(Gwyther et al., 2016).

The turbulence term $\kappa$ in Equation 1 has a different formulation between the two models. FESOM follows a very similar
approach to Jenkins (1991), by which

$$\kappa = 2.12 \log\left(u^* \frac{D}{\nu}\right) - 3 \tag{3}$$

where $D = 10$ m is a reference boundary layer depth, and $\nu = 1.95 \times 10^{-6}$ m²/s is the kinematic viscosity. ROMS instead
uses a simplified version of McPhee et al. (1987)'s approach, by which

$$\kappa = \begin{cases} 2.5 \log\left(\frac{5300(u^*)^2}{|f|}\right) + 7.12 & \text{if } u^* > 10^{-3} \text{ and } |f| > 10^{-8} \\ 0 & \text{otherwise} \end{cases} \tag{4}$$

where $f$ is the Coriolis parameter.

While refreezing is implicit in the 3-equation formulation, none of our configurations include an explicit frazil ice model
such as that of Smedsrud and Jenkins (2004) or Galton-Fenzi et al. (2012).

### 2.6 Sea ice

MetROMS includes the sea ice model CICE (Hunke et al., 2015) which is a multi-layer, multi-category model widely used in
global coupled models as well as regional and uncoupled setups. Our configuration of CICE has seven ice layers plus one snow
layer, and five ice thickness categories. It is externally coupled to ROMS, i.e. runs on separate processors, with communication
driven by the coupler MCT (Larson et al., 2005; Jacob et al., 2005). There are six baroclinic ocean timesteps (5 minutes) for
each sea ice timestep (30 minutes), and the coupler exchanges fields every sea ice timestep. Having longer timesteps for the
sea ice than for the ocean is computationally favourable, but it also introduces lags in ocean/sea-ice interactions, because the
coupled fields are time-averaged over the previous 30 minutes.

FESOM's sea ice model is described by Danilov et al. (2015). It has a single ice layer (plus one snow layer) and a single
thickness category. It is internally coupled with the ocean, running on the same processors and the same timestep. While



the FESOM sea ice model is generally less complex than CICE, it nonetheless has been shown to reproduce key features of observed Arctic and Antarctic sea ice (Timmermann et al., 2009).

Our configuration of CICE uses the "mushy" thermodynamics scheme of Turner et al. (2013a). It also includes the level-ice melt pond parameterisation of Hunke et al. (2013), and the Delta-Eddington radiation scheme (Briegleb and Light, 2007). In FESOM, sea ice thermodynamics follows Parkinson and Washington (1979) with the zero-layer approach to heat conduction (Semtner, 1976).

For sea ice dynamics, CICE uses elastic-anisotropic-plastic rheology (Tsamados et al., 2013) with the ridging-based ice strength formulation of Rothrock (1975). FESOM has elastic-viscous-plastic rheology (Hunke and Dukowicz, 1997; Hunke, 2001) including a linear formulation of ice strength with coefficient $P^* = 15,000$ N/m$^2$. Sea ice transport follows an incremental remapping approach in CICE (Lipscomb and Hunke, 2004), with the ridging participation and redistribution functions of Lipscomb et al. (2007). FESOM uses a backward Euler implicit advection scheme for sea ice transport.

## 3 Experimental design

For this intercomparison, we simulated the 25-year period 1992-2016 using three model configurations: MetROMS, low-resolution FESOM, and high-resolution FESOM (Figure 1).

### 3.1 Initial conditions

All simulations are initialised using monthly-averaged observational or reanalysis products for January 1992. Initial ocean temperature and salinity are taken from the ECCO2 reanalysis (Menemenlis et al., 2008; Wunsch et al., 2009), and extrapolated into ice shelf cavities using a nearest-neighbour method in Cartesian space. Initial ocean velocity and sea surface height are set to zero.

Sea ice is initialised using the NOAA/NSIDC Climate Data Record for Passive Microwave Sea Ice Concentration (Meier et al., 2013). Wherever the observed Antarctic sea ice concentration exceeds 0.15, the model is initialised with concentration 1, ice thickness of 1 m, and snow thickness of 0.2 m. This is the same method used by Naughten et al. (2017) and is similar to that of Kjellsson et al. (2015). FESOM, having a global domain, also requires initial conditions for Arctic sea ice. We follow the same method as for the Antarctic, but set the initial ice thickness to 2 m, since Arctic sea ice tends to be thicker (Kwok and Cunningham, 2008; Worby et al., 2008). Initial sea ice velocity is set to zero.

Our experiments do not include a proper spinup to a quasi-equilibrium state. For the purposes of this intercomparison around the Antarctic margin and continental shelf, as well as in the ice shelf cavities, we argue a full spinup is not worth the computational expense. That is because the processes we focus on - onshore flow, dense shelf water formation, and ocean/ice-shelf interaction - equilibrate much more quickly than the interior ocean. Indeed, area-averaged basal melt rates in our experiments stabilise within 5-10 years for most ice shelves.

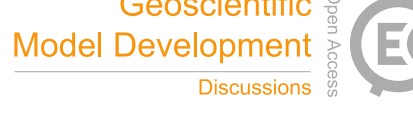



## 3.2 Atmospheric forcing

MetROMS and FESOM are both forced with the ERA-Interim atmospheric reanalysis (Dee et al., 2011) using 6- and 12-hourly fields over the years 1992-2016. Due to differing implementations of model thermodynamics, the two models are forced with different combinations of atmospheric variables. Both models utilise 6-hourly fields for near-surface air temperature, pressure, and winds, which are linearly interpolated to each timestep. Near-surface humidity is derived from ERA-Interim's 6-hourly fields for dew point temperature; this conversion is performed in advance for MetROMS, but at run-time for FESOM. Both models read 12-hourly fields for precipitation (split into rain and snow) and evaporation, which are not interpolated in time but rather applied at a constant rate with a step change every 12 hours, as they represent total fluxes over the given 12-hour period.

MetROMS diagnoses incoming shortwave radiation from ERA-Interim's 6-hourly total cloud cover, which is interpolated to each timestep. Incoming longwave radiation is calculated internally. In FESOM, incoming shortwave and longwave radiation are read directly from ERA-Interim, as 12-hourly fields which are applied as step changes.

To account for the influence of iceberg calving on the Southern Ocean freshwater budget, both models are forced with an additional surface freshwater flux representing iceberg melt. For this field we use the monthly climatology of Martin and Adcroft (2010), interpolated to each timestep, and repeated annually. River runoff from other continents is not considered.

## 3.3 Surface salinity restoring

A persistent feature of many Southern Ocean models (Kjellsson et al., 2015; Heuzé et al., 2015; Sallée et al., 2013; Turner et al., 2013b) is spuriously deep convection in the Weddell Sea, leading to an unrealistic open-ocean polynya as warm Circumpolar Deep Water is brought to the surface. The possible causes of this widespread model bias include insufficient surface freshwater flux (Kjellsson et al., 2015) as well as insufficient summer mixed layer depths (Timmermann and Beckmann, 2004). In both circumstances, salinity in the subsurface Winter Water layer increases until the weakly stratified water column becomes unstable and overturns.

MetROMS is prone to deep convection in the Weddell Sea, and while tuning of the sea ice dynamics and ocean vertical mixing helped to delay the onset of convection, the only permanent solution we found was surface salinity restoring. We restore MetROMS to the World Ocean Atlas 2013 monthly climatology of surface salinity (Zweng et al., 2013), linearly interpolated to each model timestep and repeated annually. Restoring has a timescale of 30 days and affects the uppermost layer, whose thickness is time-varying but generally ranges from 1-3 m. We exclude the Antarctic continental shelf from this restoring (defined as regions south of $60°$S with bathymetry shallower than 1500 m, as well as all ice shelf cavities), as significant freshening of Antarctic Bottom Water occurs otherwise. Given the relatively scant observations on the continental shelf making up the World Ocean Atlas products, restoring in this region may not be appropriate.

FESOM does not develop spurious deep convection in the Weddell Sea, even for long simulations without restoring. Possible reasons for this differing behaviour between MetROMS and FESOM are discussed in Section 4.2.1. Nonetheless, we apply the same surface salinity restoring to FESOM as we do to MetROMS, so that the experiments are as similar as possible. Restoring





in FESOM is scaled with a constant depth of 10 m, which is the depth of the surface layer in z-coordinate regions, neglecting free surface variations. We do not restore north of $30°$S, as this region is outside the MetROMS domain.

### 3.4 Northern boundary conditions

MetROMS, with its regional circumpolar domain, has lateral boundary conditions at $30°$S. The ECCO2 reanalysis (Menemenlis
et al., 2008; Wunsch et al., 2009) provides temperature, salinity, and meridional velocity ($v$) as monthly averages over the transient period 1992-2016. Sea surface height is taken from the AVISO climatology (AVISO, 2011) which is a single time record. Note that tides are not considered, as discussed further in Section 5.

We follow the method described in Naughten et al. (2017) to ensure stability at the open boundary: zonal velocity $u$ is clamped to zero, the bathymetry is modified to be constant in $y$ over the northernmost 15 rows of the domain, and a sponge
layer is applied over these rows (in which the diffusivity coefficient linearly increases to 10 times its background value, and the viscosity coefficient to 100 times). Northern boundary conditions are applied using the Chapman scheme for sea surface height (Chapman, 1985), the Flather scheme for barotropic $v$ (Flather, 1976), and the radiation-nudging scheme for baroclinic $v$, temperature, and salinity (Marchesiello et al., 2001).

The presence of lateral boundary conditions derived from observations may give MetROMS an advantage for the accurate
simulation of Southern Ocean water masses, compared to FESOM which has a global domain. However, this intercomparison focuses on the continental shelf and ice shelf cavities. These regions are relatively far-field from $30°$S, compared to the ACC and the interior Southern Ocean which are more tightly coupled to the boundary conditions. For the relatively short (25-year) simulations shown here, it is unlikely that continental shelf water masses will be significantly influenced by nudging at $30°$S. Longer simulations would likely show a larger response.

# 4  Results

## 4.1  Ocean

### 4.1.1  Drake Passage transport

Zonal transport through Drake Passage is calculated at $67°$W over the period 2002-2016. The first 10 years of the simulation (1992-2001) are excluded as spinup. This time-averaged Drake Passage transport, including the standard deviation in annual
averages, is $126.8 \pm 10.8$ Sv in MetROMS, $158.6 \pm 8.0$ Sv in low-resolution FESOM, and $152.6 \pm 8.2$ Sv in high-resolution FESOM.

With regards to observations, Drake Passage transport was previously thought to lie around 134 Sv (Cunningham et al., 2003). However, recent improvements in measuring systems have suggested a higher value. As part of the cDrake project (Dynamics and Transport of the Antarctic Circumpolar Current in Drake Passage), Donohue et al. (2016) determined a Drake
Passage transport of $173.3 \pm 10.7$ Sv. Compared to these observations, the values from all three of our simulations are too low, especially in MetROMS. Additionally, all three simulations exhibit downward trends in Drake Passage transport over 2002-





2016, which are statistically significant at the 95% level: -0.29 Sv/y in MetROMS, -0.19 Sv/y in low-resolution FESOM, and -0.41 Sv/y in high-resolution FESOM. This weakening of the ACC may be driven by degradation of Southern Ocean interior water masses due to spurious diapycnal mixing, as discussed in Section 4.1.4.

### 4.1.2 Mixed layer depth

We calculate mixed layer depth using the density criterion of Sallée et al. (2013): the shallowest depth at which the potential density is at least 0.03 kg/m$^3$ greater than at the surface (or at the ice shelf interface, in the case of ice shelf cavities). Summer (DJF) and winter (JJA) mixed layer depth in each simulation, averaged over the period 2002-2016, are shown in Figure 2 for the entire Southern Ocean, and Figure 3 zoomed into the Antarctic continental shelf. Figure 2 also includes climatological observations by Pellichero et al. (2017) recalculated to use the same definition of mixed layer depth as the models. We have

not included these observations in Figure 3, as they are less reliable on the continental shelf due to insufficient measurements.

In the ACC in summer (top row of Figure 2), MetROMS shows a ring of deeper mixed layers around 100 m surrounding the region stratified by sea ice meltwater. This spatial pattern agrees well with observations, but the magnitude somewhat disagrees, as MetROMS' mixed layers are too deep in the ACC and too shallow elsewhere. FESOM has a much more uniform summer mixed layer depth which is 45 m (corresponding to the fourth layer in z-coordinate regions) throughout most of the

ACC, and generally shallower in the sigma-coordinate region of the continental shelf. Both models have significantly deeper mixed layers in winter (bottom row of Figure 2, note different colour scale) with the largest values in the northern branch of the ACC where mode and intermediate waters subduct. Observations indicate this feature should be strongest in the Pacific and Australian sectors. MetROMS shows local maxima in both regions, but the magnitude in the Pacific sector (approx. 250 m) is still quite low. FESOM only captures this feature in the Pacific sector, but here it attains mixed layer depths in excess of 500

m which exceeds observations. This overestimation is less pronounced at high resolution. Elsewhere in the ACC, FESOM's mixed layer depths (approx. 100 m) more or less agree with observations, while in MetROMS they are too deep (approx. 200 m).

Zooming into the continental shelf, the water column is largely stratified in summer (top row of Figure 3) but shows active regions of dense water formation in winter (bottom row of Figure 3, note different colour scale). Both MetROMS and FESOM

form dense water in the inner Ross and Weddell Seas, with regions of mixed layer depth exceeding 500 m, although the convection appears to be stronger in FESOM where these regions are deeper and more widespread. In the Weddell Sea, dense water formation is split into two regions on either side of the Filchner-Ronne Ice Shelf front, with shallower mixed layers in the middle. In the Ross Sea, both models show somewhat stronger convection on the western side of the Ross Ice Shelf front, near McMurdo Sound, in agreement with observations (Jacobs et al., 1979). A small region of dense water formation in western

Prydz Bay, adjacent to the Amery Ice Shelf, is also present in both models. These three regions are in agreement with observed bottom water formation sites (Foldvik et al., 2004; Gordon et al., 2015; Herraiz-Borreguero et al., 2016). FESOM also exhibits deep mixed layers ($> 500$ m) in the Amundsen Sea, which were observed in 2012 but are not a consistent feature of this region (Dutrieux et al., 2014). The presence of CDW on the Amundsen Sea continental shelf is sensitive to mixed layer depth, as a completely destratified water column filled with cold shelf water will prevent the development of a warmer bottom layer





(Petty et al., 2013, 2014). This mechanism has been proposed as a cause of cold biases in Amundsen Sea ice shelf cavities, and subsequent underestimation of ice shelf melt rates, in FESOM (Nakayama et al., 2014). In our simulations these deep mixed layers have some dependence on resolution, as they cover nearly the entire Amundsen Sea at low resolution but are more restricted to the ice shelf fronts at high resolution. Similarly, low-resolution FESOM exhibits locally deepened mixed layers

(approx. 250 m) near the southern entrance of George VI Ice Shelf in the Bellingshausen Sea, while this feature is absent at high resolution.

Mixed layer depths in ice shelf cavities show no significant seasonality (note the different colour scales for summer and winter in Figure 3), and are generally shallow ($< 50$ m) except near regions of persistent refreezing, which forms marine ice. This process increases salinity at the ice shelf base as fresh water is removed in the form of frazil ice, providing a buoyancy

forcing. Regions of marine ice formation are detailed in Section 4.3, but their signature can be seen here. The most affected region is the central Ronne Ice Shelf, which has mixed layer depths of 300-400 m in MetROMS, 50-80 m in low-resolution FESOM, and 70-120 m in high-resolution FESOM. Refreezing in this region is indeed stronger and more widespread in MetROMS than in FESOM (Section 4.3.1). All three simulations exhibit mixed layer depths exceeding 50 m in much of the Ross Ice Shelf, which has large areas of refreezing (Section 4.3.5). Only MetROMS shows increased mixed layer depths

(approx. 70 m) along the western edge of the Amery Ice Shelf, which is a region of refreezing in MetROMS but not in FESOM (Section 4.3.3). Due to the lack of observations in ice shelf cavities, the true mixed layer depths in these regions are unknown.

### 4.1.3 Water mass properties

Figure 4 plots the temperature/salinity (T/S) distribution south of $65°$S in each simulation, averaged over 2002-2016, and colour-coded based on depth. In this section we identify the different water masses represented in Figure 4, and compare their

properties between the two models. Due to a scarcity of year-round measurements on the continental shelf, it is not feasible to create a comparable figure using observations. However, limited observations of some water masses exist, and are compared to the simulated water mass properties in the text below.

Antarctic Bottom Water (AABW) is the deepest water mass (1000 m or deeper) with simulated salinity $> 34.5$ psu and intermediate temperature ($-1°$C to $1.5°$C). In both MetROMS and FESOM, the deepest AABW (below 2000 m) forks into

two distinct branches on either side of 34.7 psu. The lower-salinity branch is Weddell Sea Bottom Water (WSBW) and the higher-salinity branch is Ross Sea Bottom Water (RSBW). Limited observations of these two water masses are available through the World Ocean Circulation Experiment (WOCE) Atlas (Koltermann et al., 2011; Talley, 2007): track A23 through the Weddell Sea (considering only the section south of $65°$S, which has approximate longitude $20°$W, and below 2000 m) and track S4P through the Ross Sea (considering only the section between $150°$E and $130°$W, which has latitude $67°$S, and

below 2000 m). In these tracks, the salinity of WSBW ranges from 34.65 to 34.7 psu, and RSBW from 34.68 to 34.72 psu. The models' tendency for WSBW to be fresher than RSBW is therefore supported by observations, and both models are also in agreement with the observed salinity of WSBW. However, they both overestimate the salinity of RSBW compared to these observations, particularly FESOM which approaches 34.8 psu. Both water masses have more uniform salinity in MetROMS than in FESOM, which is reflected by narrower red lines in Figure 4.





The same WOCE tracks measure temperatures from $-0.8°$C to $-0.2°$C for WSBW, and $-0.4°$C to $0.8°$C for RSBW. The observed tendency for WSBW to be colder than RSBW is apparent in MetROMS ($-0.5°$C to $0.75°$C for WSBW, $0.25°$C to $0.75°$C for RSBW) but the two water masses have approximately the same temperature in FESOM ($-1°$C to $1°$C). In both models, RSBW temperatures more or less agree with observations, but simulated WSBW is too warm. However, these

observations do not sample the full spatial extent of the water masses, so the true temperature and salinity may have a larger range.

Circumpolar Deep Water (CDW) is shallower than AABW (200-1000 m) and warmer ($> 0°$C). In MetROMS, the temperature of CDW can exceed $3°$C, while it stays below approx. $2.5°$C in FESOM. The warmer CDW in MetROMS is consistent with increased southward spreading of warmer CDW from the north around most of the continent, as discussed in Section 4.1.4.

Observations of CDW in this region suggest a temperature range of $0.3°$C to $2.5°$C (Schmidtko et al., 2014). Both models exhibit curling, finger-like structures on the low-salinity (left) side of the CDW distribution. These features represent meanders of the ACC over the boundary $65°$S, and these meanders transport different properties southward in different geographical locations. As CDW enters the subpolar gyres, it mixes with other water masses to produce cooler Modified Circumpolar Deep Water (MCDW).

Just above the surface freezing temperature (dashed black lines in Figure 4, approx. $-2°$C) are two subsurface water masses (100-500 m depth). Low Salinity Shelf Water (LSSW, $< 34.5$ psu) and High Salinity Shelf Water (HSSW, $> 34.5$ psu) are both the result of sea ice formation, but HSSW is more affected by strong brine rejection. HSSW is saltier in low-resolution FESOM (up to 35.1 psu) than in high-resolution FESOM (up to 35 psu). This is the main difference between the two FESOM simulations, which are otherwise very similar in terms of water mass properties. MetROMS has fresher HSSW than either

FESOM simulation, with maximum salinities of approximately 34.8 psu.

At the higher end of the HSSW salinity range, and with temperatures up to $-1°$C, is surface water (0-50 m) from the Ross Sea polynya. This water mass is more prominent in the FESOM distributions than in MetROMS, due to its higher salinity. As with HSSW, FESOM's Ross Sea polynya is saltier at low resolution.

The remainder of the surface water (50 m or shallower) is Antarctic Surface Water (AASW) which has lower salinity,

generally $< 34$ psu, with temperatures between the surface freezing point and $1°$C. A spread of points with particularly low salinity ($< 33.7$ psu) represents narrow embayments on the western side of the Antarctic Peninsula, from which meltwater cannot easily escape.

Water masses below the surface freezing temperature are called Ice Shelf Water (ISW). The only way that a water mass can fall below this line (neglecting numerical error in tracer advection) is from interaction with an ice shelf base. The freezing

temperature of seawater decreases with depth, due to enhanced pressure, and at the deepest grounding lines it can approach $-3.5°$C. Water which melts or refreezes at the ice shelf base will retain this freezing temperature until it is modified by mixing or by melting/freezing at a different depth.

The temperature/salinity distributions of ISW follow distinct diagonals, where the slope is the dilution ratio of melting/freezing ice in seawater (Gade, 1979). The three deepest ice shelf cavities form the most prominent diagonals in Figure

4: in order of increasing salinity, they are the Amery, the Filchner-Ronne, and the Ross. ISW beneath the Ross Ice Shelf is





saltiest in low-resolution FESOM and freshest in MetROMS, consistent with the HSSW which feeds the cavity. In the Amery and Filchner-Ronne cavities, high-resolution FESOM displays deeper water masses than low-resolution FESOM, which is due to its better representation of deep ice near the grounding line (Sections 4.3.1 and 4.3.3).

### 4.1.4   Deep ocean drift

As our experiments do not include a full spin-up, it is useful to examine changes in the properties of deep water masses during the simulations, and compare the different ways the models are drifting. Some of these changes may be forced, as our forcing period 1992-2016 is not a steady-state climate. Other changes may be due to model deficiencies, such as artificial diapycnal mixing (by which water masses over-mix) or sea ice biases affecting deep water formation.

Figure 5 shows meridional slices of temperature and salinity along $0°$E (Greenwich Meridian), comparing the ECCO2 initial
conditions for January 1992 (a) with the January 2016 monthly average for MetROMS (b), low-resolution FESOM (c), and high-resolution FESOM (d). Greater smoothing of the FESOM bathymetry compared to MetROMS or ECCO2 is apparent, as deep ocean seamounts in the coarse-resolution regions north of $55°$S are less pronounced. This is somewhat alleviated with higher resolution.

Antarctic Intermediate Water (AAIW), the subsurface water mass north of approx. $50°$S characterised by relatively low
salinity ($< 34.5$ psu, shown as a black contour in Figure 5), shows some degree of erosion in all three simulations. Difficulty preserving AAIW is a very common problem among ocean models and is generally attributed to spurious diapycnal mixing (England, 1993; England et al., 1993) with a potential contribution from errors in surface forcing (Griffies et al., 2009). The erosion is most severe in MetROMS, and is combined with freshening of the underlying North Atlantic Deep Water (NADW). Since MetROMS has terrain-following coordinates throughout the entire domain, whereas FESOM has z-coordinates every-
where except the Antarctic continental shelf, MetROMS would indeed be expected to be more prone to diapycnal mixing in the deep ocean (Griffies et al., 2000), particularly around steep regions of bathymetry such as seamounts. The degree of AAIW erosion in MetROMS depends on the tracer advection scheme (Marchesiello et al., 2009; Lemarié et al., 2012), and our choice of the Akima advection scheme over the upwind third-order scheme (Section 2.4) was motivated by the less severe diapycnal mixing in Akima. In FESOM, AAIW is slightly better preserved at low resolution than at high resolution.

Another notable feature in Figure 5 is the larger volume of warm CDW ($> 0.75°$C, shown as a black contour) south of $60°$S in MetROMS. A slight warming of the underlying AABW is also apparent, likely due to spurious entrainment of the CDW through diapycnal mixing. The cause of this increased CDW upwelling in MetROMS is unclear. Warming and shoaling of CDW around most regions of Antarctica has been observed over recent decades, and attributed to changes in wind stress (Schmidtko et al., 2014; Spence et al., 2014, 2017). With these observations in mind, it is possible that this feature is a forced
trend, although FESOM is forced with the same winds and does not display this behaviour - in fact, the small amount of warm CDW south of $60°$S present in the initial conditions has mixed into the water column by the end of both FESOM simulations. However, CDW upwelling is also sensitive to the tracer advection scheme in MetROMS, and is more severe with the upwind third-order advection scheme (not shown). Therefore, some component of numerical error could be an additional contributing factor.





## 4.2 Sea ice

### 4.2.1 Concentration and extent

Sea ice concentration (fraction of each grid cell covered by ice) and extent (area of grid cells with concentration exceeding 0.15) are the most convenient variables for model evaluation, due to the availability of satellite observations. Here we compare with

the NOAA/NSIDC Climate Data Record of Passive Microwave Sea Ice Concentration (Meier et al., 2013) and the NSIDC Sea Ice Index version 2 for sea ice extent (Fetterer et al., 2016). We examine monthly averages for February and September, which are the months of minimum and maximum Antarctic sea ice extent, respectively, over the period 1992-2015 (observations for 2016 were not yet available at the time of writing).

Figure 6 compares time-averaged sea ice concentration (a) as well as timeseries of total sea ice extent (b) for February and

September, between NSIDC observations, MetROMS, low-resolution FESOM, and high-resolution FESOM. All three of our simulations underestimate the sea ice minimum, which is a common bias among sea ice models (Turner et al., 2013b). The majority of simulated February sea ice is in the Weddell Sea (Figure 6a, top row), which agrees with observations, although in both FESOM simulations it extends too far northeast into the Weddell Gyre. Observed patches of coastal ice in the Amundsen and Bellingshausen Seas, as well as along the coast of East Antarctica, are largely absent in MetROMS and almost completely

absent in FESOM.

Note also in February the dominance of intermediate concentrations, between 0.25 and 0.75, with virtually no regions of concentration 1 in either model. This characteristic is an artifact of time-averaging over multiple years, and signifies regions which have near-total ice cover in some years (concentration $\approx 1$) but open water in other years (concentration $\approx 0$). February sea ice extent in any individual year, therefore, is lower than this time-averaged figure may suggest. Indeed, the timeseries in

Figure 6b (top panel) reveals that all three simulations underestimate February sea ice extent by approximately a factor of 2 compared to observations. However, they all display some of the observed interannual variability, such as the high in 2008 and the low in 2011. These year-to-year variations are driven by seasonal atmospheric conditions which manifest in ERA-Interim.

In FESOM, the sea ice minimum is slightly greater at high resolution. This difference is driven by summertime conditions in the southern Weddell Sea and the east coast of the Antarctic Peninsula. In the low-resolution mesh, smoother bathymetry

near the peninsula allows a southward excursion of the southern boundary of the ACC in summer, which carries warmer water into the region and melts more sea ice.

The sea ice maximum in September is well captured by all three simulations, which exhibit zonal asymmetry in line with observations (Figure 6a, bottom row). Sea ice concentrations throughout most of the ice pack are slightly too low in MetROMS (approx. 0.94) and slightly too high in both FESOM simulations (approx. 0.995) compared to NSIDC observations (approx.

0.97). This difference between the models influences the air-sea fluxes, which are modulated by the sea ice concentration. For example, the ocean in MetROMS will experience slightly greater wind stress than in FESOM, and therefore more turbulent mixing. In particular, sea ice concentration affects the air-sea heat fluxes, which may shed some light on the spurious Weddell Sea deep convection seen in MetROMS (without surface salinity restoring) but not in FESOM, as described in Section 3.3. Winter sea ice concentrations far below 1 in MetROMS allow frazil ice to form in the middle of the ice pack, rather than being

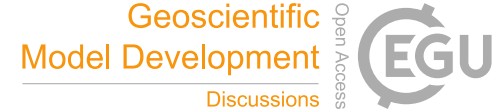

restricted to coastal polynyas. This introduces a positive feedback by which brine rejection increases the sea surface salinity, causing destabilisation of the water column and upwelling of warm water, which melts surrounding sea ice and exposes more open water to the cold atmosphere. By contrast, FESOM's winter sea ice has concentrations near 1 almost everywhere, which shields the ocean surface from atmospheric heat fluxes and the resulting frazil ice formation and brine rejection. However,

differences in vertical mixing schemes between the two models could also affect their sensitivity to spurious Weddell Sea deep convection (Timmermann and Beckmann, 2004), as discussed in Section 2.4.

While the general pattern of both models' September sea ice agrees with observations, the northern edge of the ice pack is too far south in MetROMS and too far north in FESOM. These discrepancies are reflected in the timeseries of September sea ice extent (Figure 6b, bottom panel) where the NSIDC observations fall between the MetROMS and FESOM simulations. In-

terannual variability is well represented, with both models reproducing many of the highs and lows seen in the observations. No significant difference in winter sea ice cover is apparent between the low-resolution and high-resolution FESOM simulations.

### 4.2.2 Thickness

Observations of sea ice thickness are scarce and have large uncertainties (Holland et al., 2014). A comprehensive evaluation of MetROMS and FESOM with respect to sea ice thickness is therefore difficult, although a comparison of the two models can

still be made. Figure 7 shows seasonal averages of sea ice effective thickness (concentration times height) in each simulation, averaged over 1992-2016.

Sea ice is generally thicker in MetROMS than in either FESOM simulation, particularly in the Weddell Sea, the Amundsen and Bellingshausen Seas, and along the coastline of East Antarctica. This difference may be due to complex dynamic processes such as ridging and rafting, which are not considered by single-layer sea ice models such as the one used in FESOM. However,

FESOM's coastal sea ice is slightly thicker at high resolution, particularly in the Amundsen and Bellingshausen Seas.

In MetROMS, a particularly thick region of sea ice (approx. 3 m) exists on the western edge of the Weddell Sea, along the Antarctic Peninsula. This feature is also present in IceSAT observations (Kurtz and Markus, 2012; Holland et al., 2014), and in-situ measurements of second-year ice in the western Weddell Sea find thicknesses of 2.4 to 2.9 m (Haas et al., 2008). The region of thick ice is less pronounced, but still visible, in the high-resolution FESOM simulation. In low-resolution FESOM,

the southward excursion of the southern boundary of the ACC in summer (see Section 4.2.1) prevents multi-year ice from building up in this region, so the feature is mostly absent. All three simulations show some sign of the Ronne Polynya in winter (JJA) and spring (SON), with thinner sea ice near the Ronne Depression.

### 4.3 Ice shelf cavities

Considering Antarctica as a whole, all three model simulations underestimate the contribution of ice shelf basal melting to

the Southern Ocean freshwater budget, roughly by a factor of two. The simulated total mass loss from ice shelf basal melting, averaged over 2002-2016, is 642 Gt/y for MetROMS, 586 Gt/y for low-resolution FESOM, and 739 Gt/y for high-resolution FESOM. In comparison, the observation-based estimate of Rignot et al. (2013) was $1325 \pm 235$ Gt/y for the period 2003-2008. Depoorter et al. (2013) have an even larger estimate, of $1454 \pm 174$ Gt/y, mainly due to higher estimated mass loss from the



Abbot, George VI, Brunt, and Riiser-Larsen Ice Shelves (although estimates for the Ross and Filchner-Ronne Ice Shelves are lower than in Rignot et al.).

A closer examination of individual ice shelves shows that the bias in our simulations is a regional phenomenon. Table 1 compares simulated basal mass loss to Rignot et al.'s estimates for 25 ice shelves, organised into eight regions. The model biases
are summarised in Figure 8, which plots the difference between the simulated values and Rignot et al.'s central estimates, as well as the uncertainty range, for each ice shelf. All three simulations underestimate mass loss for ice shelves in the Amundsen Sea, Bellingshausen Sea, and Australian Sector. These three regions include many warm-cavity ice shelves which, despite their small areas, exhibit substantial basal mass loss in observations. However, ice shelves in the remaining five regions generally show either agreement between our model simulations and Rignot et al.'s observations, or an overestimation of mass loss by the
models (with the main exception being MetROMS' underestimation of the Filchner-Ronne Ice Shelf). The following sections will analyse these eight regions in more detail.

The average annual minimum in total basal mass loss (calculated over 5-day averages between 2002 and 2016) is 490 Gt/y for MetROMS, 323 Gt/y for low-resolution FESOM, and 379 Gt/y for high-resolution FESOM. The corresponding average annual maximum values are 1017 Gt/y, 1589 Gt/y, and 1988 Gt/y respectively. Note that the seasonal cycle is larger in FESOM
than in MetROMS, which is likely related to greater summertime melting near ice shelf fronts. Transport of warm AASW into ice shelf cavities is enhanced by FESOM's more significant smoothing of the ice shelf front, as discussed in the following sections.

The area of a given ice shelf in model simulations does not necessarily agree with the area used in Rignot et al.'s calculations, particularly for small ice shelves which are not well resolved by the models. Such disagreements may bias our comparison.
However, a comparison of area-averaged basal melt rates rather than area-integrated basal mass loss (not shown) shows essentially the same biases. We also must acknowledge the uncertainties inherent in Rignot et al.'s estimates, which largely neglect mass balance at the ice shelf front, and rely on atmospheric reanalyses which in turn have limited observations from which to downscale. Finally, neither MetROMS nor FESOM considers the effects of tides. Since the heat and salt transfer coefficients in both models depend on ocean velocity adjacent to the ice shelf base, tidal currents would be expected to increase melt rates
in all ice shelf cavities. Tides also cause enhanced vertical mixing, which further influences melt rates (Gwyther et al., 2016).

To aid our intercomparison, we have categorised the water in ice shelf cavities into five water masses based on discrete temperature and salinity bounds, defined in Table 2: ISW, MCDW, HSSW, LSSW, and AASW. Figure 9 plots the percent volume of each water mass in ice shelf cavities, for the eight regions specified in Table 1 as well as the total for all Antarctic ice shelf cavities. These proportions are based on temperature and salinity fields averaged over 2002-2016 for each simulation, and
neglect the seasonal cycle. AASW, which is mostly a summertime phenomenon, may therefore be obscured. For Antarctica as a whole (Figure 9i), both FESOM simulations have more MCDW and HSSW, and less ISW and LSSW, than in MetROMS. These differences are more pronounced in low-resolution FESOM, while high-resolution FESOM is more similar to MetROMS. The water mass proportions in each region (Figure 9a to 9h), and consequently the reasons for the overarching differences between the three models, will be analysed in the following sections.



### 4.3.1 Filchner-Ronne Ice Shelf

The Filchner-Ronne Ice Shelf (FRIS) is the largest ice shelf in the Weddell Sea region and the second largest (by area) in Antarctica. However, its melt rates are quite low away from the grounding line, leading to relatively modest basal mass loss for its size. For both FESOM simulations, basal mass loss for FRIS falls within the range of observations given by Rignot et al. (Table 1.1). MetROMS significantly underestimates this rate, simulating about half the lower bound given by Rignot et al.. Figure 10a shows the spatial distribution of this mass loss in the three simulations, with two-dimensional ice shelf melt/freeze fields averaged over 2002-2016.

Circulation patterns in the FRIS cavity have been inferred from a few sub-ice shelf observations (Nicholls and Østerhus, 2004; Nicholls and Johnson, 2001) and feature anticyclonic flow around Berkner Island, a cyclonic gyre in the Filchner Ice Shelf cavity, and HSSW inflow into the western Ronne Ice Shelf cavity via the Ronne Depression. MetROMS displays the anticyclonic flow around Berkner Island (Figure 10b), which is stronger on the western and southern sides, corresponding with locally increased melt rates. Circulation in the Filchner Ice Shelf cavity is weak but mostly cyclonic. However, the observed refreezing immediately east of Berkner Island (Joughin and Padman, 2003; Rignot et al., 2013) is absent.

In FESOM, southward flow of HSSW on the western flank of the Filchner Depression drives a relatively strong anticyclonic gyre in the Filchner cavity. Melting is therefore apparent on the western side of the Filchner Ice Shelf front, with refreezing associated with outflow in the east. This pattern is opposite to observations and may be caused by FESOM's strong HSSW formation in the Filchner Depression, as evidenced by deep wintertime mixed layers in Figure 3. Melting at the Filchner Ice Shelf front is stronger in the low-resolution FESOM simulation than the high-resolution, with vigorous melting immediately east of Berkner Island. This feature is likely due to the slightly smoother ice shelf front in the low-resolution simulation (Figure 10e), which allows for greater transport of warm AASW into the cavity. At the southern coast of Berkner Island, FESOM's inflowing current splits into two branches, one continuing westward along the southern edge of the Ronne Ice Shelf cavity, and the other turning northward to continue the cyclonic flow around the island.

In the Ronne Ice Shelf cavity, observations by Rignot et al. as well as Joughin and Padman indicate significant areas of refreezing in the interior combined with melting at the ice shelf front. MetROMS captures both of these features. FESOM exhibits refreezing in the interior (albeit weaker than in MetROMS) but also at the ice shelf front, with a band of melting between the two regions. At the western edge of the cavity, a narrow band of refreezing associated with outflow is present in both MetROMS and FESOM, which agrees with observations by Nicholls et al. (2004).

In MetROMS, the strongest melting occurs in the pockets of deep ice at the back of the cavity. Compared to the remote sensing observations of Joughin and Padman (2003), MetROMS overestimates melt rates in these regions of the Filchner Ice Shelf, but underestimates them for the Ronne Ice Shelf. In FESOM, melting is weaker and more widespread along these grounding line regions, which more or less agrees with Joughin and Padman's observations for the Filchner grounding line, although melt rates in the interior Filchner Ice Shelf are too high. These grounding line regions are largely bypassed by the gyre transporting HSSW through the Filchner cavity, and therefore remain cooler and fresher. There may also be a small effect from FESOM's greater smoothing of the ice shelf draft, which causes the deepest ice to shoal by approximately 100 m. All





else being equal, shallower ice melts more slowly due to its increased in-situ freezing point. In this situation, the increase is approximately $0.076°C$, which is likely to be overwhelmed by other factors. The pockets of deep ice are better preserved in the high-resolution FESOM mesh than the low-resolution, and show no significant changes in melt rate. Instead, the slightly thinner water column further inhibits HSSW transport to these grounding line regions, where the temperature is slightly cooler.

This tendency for changes in velocity to offset changes in the in-situ freezing point in the FRIS cavity was demonstrated by Timmermann and Goeller (2017) for a fully coupled configuration with an evolving ice shelf draft.

Other FESOM simulations focusing on FRIS exhibit somewhat different melt rate patterns. For example, Timmermann and Goeller (2017) simulate more vigorous melting near the grounding line, as well as a larger area of refreezing in the interior Ronne Ice Shelf which does not quite extend to the ice shelf front. However, these simulations used a different atmospheric

forcing dataset, which may influence sea ice formation patterns and consequently sub-ice shelf circulation. Additionally, the mesh used by Timmermann and Goeller has extremely high resolution at the FRIS grounding line (approx. 1 km), which may allow for better representation of ocean velocities beneath the deepest ice.

With respect to water mass properties, Figure 9a reveals that the FRIS cavity in MetROMS is almost entirely filled with ISW, with a small contribution from HSSW (<5%). This dominance of ISW indicates that it has a relatively long simulated

residence time in the FRIS cavity. FESOM's stronger dense water formation in the Filchner Depression means that HSSW has a much larger presence (approx. 40%), indicating more rapid flushing of the cavity; this effect is slightly lessened at high resolution. As a result, FESOM has warmer and saltier bottom water than MetROMS in most of the FRIS cavity (Figures 10c and 10d). However, observations south of Berkner Island (Nicholls and Johnson, 2001) reveal bottom water temperatures around $-2.2°C$, which implies that both MetROMS and FESOM are too warm. The fact that MetROMS still underestimates

total basal mass loss, despite a warm bias, suggests that its relatively weak circulation and long residence time are to blame for low melt rates.

The exceptionally large tides of the Weddell Sea (Foldvik et al., 1990), with tidal velocities up to 1 m/s (Robertson et al., 1998), are understood to have a strong impact on FRIS melt rates. Indeed, inclusion of tides in one regional model of the southern Weddell Sea caused basal mass loss from FRIS to approximately double (Makinson et al., 2011). Another model

(Mueller et al., 2017) found no significant change in total mass loss, but rather an amplification of existing melt and freeze patterns. As mentioned previously, neither MetROMS nor FESOM considers these effects.

### 4.3.2 Eastern Weddell Region

We define the Eastern Weddell region as the line of ice shelves east of FRIS, stretching along the coastline of Queen Maud Land, from the Brunt Ice Shelf in the west to the Prince Harald Ice Shelf in the east. Compared with the values given by Rignot

et al. (Table 1.2), all three simulations overestimate basal mass loss from the combined Brunt and Riiser-Larsen Ice Shelves as well as the Prince Harald Ice Shelf. Both FESOM simulations also overestimate mass loss from the combined Fimbul, Jelbart, and Ekstrom Ice Shelves. Simulated mass loss for the other ice shelves in the Eastern Weddell region generally falls within the observational estimates. However, some of the ice shelves are so small that they are barely resolved by the MetROMS grid or the low-resolution FESOM mesh, particularly the Nivl, Lazarev, and Prince Harald Ice Shelves.



A notable feature of the Eastern Weddell region is an overhang of some ice shelf fronts past the continental shelf break, which in our simulations allows the Antarctic Coastal Current to undercut the ice shelf (Figure 11b). This process is particularly strong in MetROMS, where increased velocity corresponds with vigorous melting at the fronts of the Brunt, Riiser-Larsen, and Fimbul Ice Shelves, with weaker melting or refreezing further back in the cavities (Figure11a). This pattern more or less agrees with

observations by Langley et al. (2014) for the Fimbul Ice Shelf. FESOM has a weaker coastal current than MetROMS in this region, and melting is less concentrated at the ice shelf fronts. In general, melting is stronger in the high-resolution FESOM simulation than the low-resolution.

Observations of Eastern Weddell ice shelf cavities are scarce, but mooring data does exist for the Fimbul Ice Shelf (Hattermann et al., 2012). It reveals a generally cold cavity with temperatures around $-1.9°C$ and salinities around 34.3 psu, with

occasional intrusions of warmer, saltier MCDW at depth, and seasonal melting at the ice shelf front due to sun-warmed AASW. For comparison, Figures 11c and 11d plot meridional slices of simulated temperature and salinity through the Fimbul Ice Shelf cavity at $1°W$. All three of our simulations show a warmer and (in the case of FESOM) saltier cavity than seen in the range of mooring readings (see Figure 2 of Hattermann et al.). It appears that MCDW is mixing too readily into the cavity, particularly in FESOM where the continental shelf break is more gently sloping due to smoothing of the bathymetry (Figures 11c and 11d,

first column vs. second and third columns). The high-resolution FESOM mesh has a slightly steeper continental shelf break than the low-resolution mesh, and the cavity is slightly cooler and fresher, indicating that oversmoothing of the continental shelf break may be a contributing factor in the increased transport of MCDW.

For the Eastern Weddell region as a whole, FESOM has proportionally more MCDW in ice shelf cavities than MetROMS, as evidenced by Figure 9b. As for the Fimbul, this effect is slightly lessened at high resolution. The Eastern Weddell cavities

in MetROMS are more dominated by LSSW (approx. 50%) with a small amount of ISW (<5%). Traces of AASW are present in both FESOM simulations but not in MetROMS, although it may still exist on a seasonal basis.

### 4.3.3   Amery Ice Shelf

All three model simulations overestimate basal mass loss from the Amery Ice Shelf, by about 50% above the upper bound given by Rignot et al. for MetROMS, and about 20% for both FESOM simulations (Table 1.3). However, the sources of this

bias are quite different between the two models.

In MetROMS, the majority of melting occurs near the grounding line (Figure 12a), which has the one of the deepest ice shelf drafts in Antarctica ($> 2000$ m) (Galton-Fenzi et al., 2008). A cyclonic circulation pattern is apparent (Figure 12b), with refreezing along the western side of the cavity. This spatial distribution of melting and freezing agrees with observations (Wen et al., 2010; Rignot et al., 2013) and with previous modelling (Galton-Fenzi et al., 2012), but the melting at the grounding line

appears to be too strong ($> 40$ m/y) and the refreezing too weak and over an insufficient area. These two biases combine to cause the overestimation of total basal mass loss simulated by MetROMS. It is possible that both biases could be addressed with an explicit frazil ice parameterisation with multiple size classes, as Galton-Fenzi et al. (2012) showed that including such a parameterisation in ROMS both reduced melting at the grounding line and increased refreezing on the western side of the





cavity. Additionally, the back of the Amery Ice Shelf cavity is very steep and not well resolved by MetROMS, which could lead to pressure gradient errors causing excessive melt.

FESOM exhibits much weaker melting than MetROMS at the back of the cavity, even though bottom water temperatures are at least as warm (Figure 12c). We attribute this discrepancy at least party to to shoaling of the ice shelf draft from oversmoothing in our FESOM setups, which raises the in-situ freezing point and therefore reduces melting. The Amery draft is so steep that in order for a FESOM mesh to preserve the >2000 m deep ice near the grounding line, resolution of 1.5 km or finer is required throughout the entire cavity. However, experimentation with such a mesh revealed severe timestep limitations: even a 1 minute timestep was prone to numerical instabilities, compared to the 10 minute and 9 minute timesteps which are stable for the two FESOM meshes we present here. Due to the computational expense, we chose not to pursue simulations with this experimental mesh. Nonetheless, some improvement can be seen between the low-resolution mesh and the high-resolution mesh (Figure 12e), in which the deep ice has shoaled by approximately 1000 m and 800 m respectively compared to the source topography. The corresponding increases in the in-situ freezing point are $0.76°C$ and $0.61°C$. In the high-resolution simulation, melt rates near the grounding line exceed 10 m/y on the eastern flank, which is approximately double that of the low-resolution simulation. Such a strong response is not due to the in-situ freezing point alone, which is only modestly different between the two simulations. Increased velocities near the back of the cavity, possibly due to the steeper ice draft, also have an effect.

In both FESOM simulations, significant melting also occurs near the ice shelf front and throughout the outer third of the ice shelf, at a higher rate (up to 5 m/y) than in MetROMS. This melting is somewhat lessened at high resolution, offsetting the increased melt rates at the grounding line. It occurs primarily in summer, leading to a large seasonal cycle in total basal mass loss for the entire ice shelf. Average annual minimums and maximums (calculated over 5-day averages between 2002 and 2016) are 27 Gt/y and 451 Gt/y for low-resolution FESOM, and 30 Gt/y and 309 Gt/y for high-resolution FESOM. In comparison, MetROMS has an average annual minimum of 78 Gt/y and maximum of 119 Gt/y. The seasonality and resolution-dependence of melting in the outer third of the cavity suggest that it is driven by warm AASW subducting beneath an oversmoothed ice shelf front. Circulation in FESOM is predominantly anticyclonic, with no significant areas of refreezing. This reversed circulation shows little sensitivity to the improved cavity geometry at high resolution, indicating that it is more likely driven by hydrography, as for the Filchner Ice Shelf (Section 4.3.1).

Bottom water in FESOM is warmer and saltier than in MetROMS throughout the cavity (Figures 12c and 12d); these differences are lessened at higher resolution. Figure 9c indicates that the Amery cavity is dominated by LSSW in both MetROMS and high-resolution FESOM, with smaller contributions from ISW (5-20%) and MCDW (<5%). Low-resolution FESOM has significantly more MCDW in the cavity (approx. 40%), which may be tied to oversmoothing of the continental shelf break as for the Fimbul Ice Shelf (Section 4.3.2).

### 4.3.4 Australian Sector

Travelling east from the Amery, the remainder of the Australian sector of Antarctica contains numerous small ice shelves along the coast of Wilkes Land, including the West, Shackleton, Totten, and Mertz Ice Shelves. Rignot et al. estimate relatively high melt rates for these ice shelves, which all three of our simulations fail to capture (Table 1.4).





There are several potential reasons for this consistent underestimation. First, the observed production of HSSW which drives the majority of melting for cold-cavity ice shelves (Jacobs et al., 1992) is enhanced in this region by many small polynyas which are kept open by grounded icebergs (Kusahara et al., 2010; Gwyther et al., 2014) and coastline geometry (Tamura and Ohshima, 2008). Neither MetROMS nor FESOM considers grounded icebergs in the configurations used here, so it is likely that both sea ice and HSSW production in this region are underestimated. In fact, Figure 9d reveals no year-round presence of HSSW in Australian sector ice shelf cavities for any of the three simulations. The dominant water mass from sea ice formation is instead LSSW, which is fresher and less dense.

Whether enhanced production of HSSW would lead to a decrease or increase in basal melting depends on the local hydrography. For example, flooding a cavity with relatively cold HSSW formed in coastal polynyas could prevent any nearby MCDW from accessing the cavity, in which case stronger polynyas would lead to decreased melt rates (Cougnon et al., 2013; Khazendar et al., 2013; Gwyther et al., 2014). Indeed, MCDW has recently been observed in front of the Totten Ice Shelf (Greenbaum et al., 2015; Rintoul et al., 2016; Silvano et al., 2017) which could be a factor in its considerable melt rate. However, MCDW can only access the cavity through a trough in the continental shelf which was previously unknown and therefore not included in RTopo-1.05 (Greenbaum et al., 2015). Even if this trough was included in the bathymetry datasets used by MetROMS and FESOM, the models would likely not resolve such a small-scale feature without increased resolution. As seen in Figure 9d, a significant amount of MCDW is still present in Australian sector ice shelf cavities for all three simulations (approx. 20% in MetROMS, 85% in low-res FESOM, and 75% in high-res FESOM). However, the fairly wide temperature range of water masses we consider to be MCDW (Table 2) means that the degree of modification is important for ice shelf melting.

Simulated ice shelf melt rates are shown for the Australian sector in Figure 13a. Melt rates in MetROMS are generally more concentrated at the ice shelf front, and in FESOM more uniform throughout the cavities. As suggested by FESOM's greater proportion of MCDW in Figure 9d, FESOM has slightly warmer and (in the case of low-res FESOM) saltier bottom water in most ice shelf cavities (Figures 13b and 13c). This is despite the fact that bottom water offshore of the continental shelf is warmer in MetROMS (approx. $0.5°C$) than in either FESOM simulation (approx. $-0.2°C$), and indicates that cross-shelf heat transport is stronger in FESOM. Melt rates are enhanced in the high-resolution FESOM simulation compared to low-resolution, and in most cases are also higher than MetROMS. This pattern is likely due to stronger circulation as shown for the Totten Ice Shelf in Figure 13d. The MetROMS grid and low-resolution FESOM mesh cannot adequately resolve circulation in such a small cavity, which is represented by only a few dozen grid boxes or triangular elements. Resolution is still less than ideal for the high-resolution FESOM mesh, but this simulation manages to develop an anticyclonic gyre beneath the ice shelf. Stronger transport through the cavity at high resolution, as well as increased transfer coefficients due to the faster velocity, causes basal mass loss for the Totten Ice Shelf to more than double compared to the low-resolution FESOM simulation, even though temperatures are slightly higher in the latter simulation. By comparison, the ROMS configuration of Gwyther et al. (2014) for the Totten region was even higher resolution (approx. 3 km), and instead simulated a cyclonic gyre.





### 4.3.5 Ross Sea

For the Ross Ice Shelf, the largest in Antarctica (by area), MetROMS falls within Rignot et al.'s estimate of basal mass loss while both FESOM simulations produce an overestimate (Table 1.5). In this region we also include the nearby Sulzberger Ice Shelf, for which all three simulations underestimate basal mass loss, and the Nickerson Ice Shelf, for which MetROMS and

high-resolution FESOM agree with Rignot et al.'s observations, but low-resolution FESOM produces a slight underestimate.

All simulations show predominantly anticyclonic circulation beneath the Ross Ice Shelf, with inflow in the west and outflow in the east (Figure 14b), in agreement with observations (Reddy et al., 2010). A similar system of interconnected gyres is seen in all three simulations, although circulation near the back of the cavity is stronger in MetROMS.

Patterns of refreezing are similar between simulations (Figure 14a), with large areas of refreezing near the back of the

cavity concentrated around Steers Head and Crary Ice Rise. FESOM also displays relatively strong refreezing near the edge of McMurdo Ice Shelf, particularly in the high-resolution simulation, which has been observed (Langhorne et al., 2015).

The Ross Ice Shelf front is exceptionally steep and requires smoothing in all three simulations for numerical stability. This smoothing allows relatively warm AASW to slide under the ice shelf front, which is visible as tongues of warm water in meridional temperature slices through $180°E$ (Figure 14c). The low-resolution FESOM simulation exhibits this problem most

severely, but it is reduced at high resolution as the ice shelf front requires less smoothing. The interior of the cavity is warmer in FESOM than in MetROMS, and (particularly at low resolution) saltier (Figure 14d), due to stronger sea ice formation as discussed in Section 4.1.2. Figure 9e confirms that both FESOM simulations, but particularly low-resolution FESOM, have more HSSW and less ISW than MetROMS in Ross Sea ice shelf cavities. These differences in hydrography explain why melting in the outer third of the cavity remains stronger in FESOM than in MetROMS even when the oversmoothing of the ice

shelf front has been addressed with higher resolution.

FESOM also displays stronger melting east of Roosevelt Island, where a small amount of relatively warm MCDW enters the cavity (not shown). This water mass has been observed (Reddy et al., 2010; Jacobs and Comiso, 1989) and is likely not a model artifact. Indeed, Paolo et al. (2015) detect ice shelf thinning in this region over the period 1994-2012.

### 4.3.6 Amundsen Sea

Ice shelves in the Amundsen Sea have been the subject of much attention in recent years, due to observed intrusions of unmodified CDW causing rapid basal melting and grounding line retreat (Hellmer et al., 1998; Jacobs et al., 2011; Jenkins et al., 2010; Wåhlin et al., 2010). The Amundsen Sea has the highest ice shelf melt rates of any sector of Antarctica, corresponding to large basal mass loss coming from a handful of relatively small ice shelves. However, all three of our simulations severely underestimate these mass loss values, as shown in Table 1.6.

There are several likely reasons for this systematic bias, the first and most well-studied being resolution. Intrusion of CDW onto the continental shelf of the Amundsen Sea depends on small-scale features in the bathymetry, which cannot be resolved by model grids coarser than approx. 5 km (Nakayama et al., 2014). Of the three simulations in this intercomparison, only



high-resolution FESOM falls within this threshold. Eddy transport of heat is also an important factor for cross-shelf CDW exchange, and this process requires resolutions of 1 km or less (St-Laurent et al., 2013), which none of our simulations have.

In FESOM, CDW transport into the Amundsen Sea has been shown to be sensitive to the depth of transition between sigma-coordinates and z-coordinates, with a shallower transition favouring the transport of warmer CDW due to the better alignment
of z-coordinates with isopycnals in this region (Nakayama et al., 2014). Our simulations have a relatively deep transition of 2500 m, which supports the on-shore transport of cooler CDW. Additionally, the very deep mixed layers in the Amundsen Sea which develop in both FESOM simulations (Section 4.1.2) produce a slope front which further blocks CDW from the continental shelf. Any warm water flowing along the bottom is eroded by the convection of cold LSSW.

All three simulations underestimate bottom water temperature throughout the continental shelf (Figure 15b), which has
been observed at approx. $1°C$ in the Pine Island Ice Shelf cavity (Jacobs et al., 2011; Jenkins et al., 2010) and $0.5$ to $1.2°C$ throughout the Amundsen Sea (Dutrieux et al., 2014). Temperatures are warmer in MetROMS (approx. $-0.2°C$) than in both FESOM simulations (approx. $-1.7°C$ for low-resolution and $-1°C$ for high-resolution), leading to higher melt rates (Figure 15a). All three simulations are dominated by MCDW in this region (Figure 9f), however FESOM has a larger presence of LSSW than MetROMS due to the deep mixed layers discussed previously. Consistent with the reduced cross-shelf transport of
CDW, the Antarctic Slope Front current (not shown) is stronger in FESOM compared to MetROMS, in the Amundsen as well as Bellingshausen Sea sectors.

Increasing the resolution in FESOM causes a substantial increase in basal mass loss for all Amundsen Sea ice shelves (Table 1.6). These changes are most pronounced for the Pine Island Ice Shelf, where mass loss increases by approximately a factor of 5. A major contributor to this increased melting is better resolution of troughs in the bathymetry which provide a pathway for
warmer water to access the continental shelf. In the easternmost trough, near the Thwaites and Pine Island Ice Shelves, bottom water temperature increases by approximately $1°C$ as a result of increased resolution. Salinity also decreases throughout the Amundsen Sea, due to entrainment from additional meltwater (Figure 15c).

Melting is further enhanced in the high-resolution FESOM simulation due to stronger circulation, as shown for the Pine Island Ice Shelf in Figure 15d. As for the Totten Ice Shelf (Section 4.3.4), circulation in this small cavity is not resolved by
MetROMS or low-resolution FESOM, but high-resolution FESOM develops an anticyclonic gyre which increases melt rates due to larger friction velocities.

### 4.3.7 Bellingshausen Sea

Similarly to the Amundsen Sea, the nearby Bellingshausen Sea experiences intrusions of unmodified CDW, particularly beneath George VI Ice Shelf in the east (Jenkins and Jacobs, 2008) as well as into Margeurite Bay (Moffat et al., 2009). Again, all three
model simulations largely fail to capture the CDW intrusions, and as a result underestimate basal mass loss for the Stange and George VI Ice Shelves (Table 1.7). MetROMS and low-resolution FESOM also produce an underestimate for the Abbot Ice Shelf, while high-resolution FESOM agrees with Rignot et al.'s observations. All three simulations agree with observations for the Wilkins Ice Shelf. The increase in melting between the low-resolution and high-resolution FESOM simulations is





substantial, with basal mass loss more than doubling for the Stange and George VI Ice Shelves. However, this still falls below the range of observations.

As in the Amundsen Sea, insufficient resolution as well as the depth of the sigma-z transition in FESOM could be playing a role in the simulated lack of CDW intrusions. Observations suggest that the dominant mechanism for cross-shelf CDW

transport in this region is eddies shed from the ACC (Martinson and McKee, 2012), which none of our simulations resolve. However, the FESOM simulations of Timmermann et al. (2012) had much higher melt rates in the Bellingshausen Sea, despite similar resolution and the same sigma-z transition as the simulations presented here. Since Timmermann et al. forced FESOM with the NCEP/NCAR atmospheric reanalysis rather than ERA-Interim, differences in atmospheric forcing could also be a factor.

MetROMS has generally warmer bottom water in the Bellingshausen Sea than either FESOM simulation (Figure 16b), leading to stronger melting in some regions (Figure 16a). In Ronne Entrance, extending into the southern end of the channel-shaped George VI Ice Shelf, MetROMS displays an intrusion of MCDW (approx $0°C$). High-resolution FESOM also has warmer bottom water here than low-resolution FESOM (approx $-0.25°C$ and $-0.75°$ respectively). For all three simulations, however, bottom water temperatures in this region and in Marguerite Bay fall well below the observed $1°C$ (Jenkins and Jacobs,

2008). As in the Amundsen Sea, the increased meltwater in high-resolution FESOM compared to low-resolution FESOM leads to lower salinities in most regions of the Bellingshausen Sea (Figure 16c). MetROMS displays pockets of exceptionally cold and fresh water (approx. $-2°C$ and 33.6 psu) in some ice shelf cavities, which is the result of poor resolution preventing meltwater from efficiently circulating out of semi-isolated regions, as discussed in Section 4.1.3. All three simulations show a dominance of MCDW in the Bellingshausen Sea region (Figure 9g), with a smaller presence of AASW (<5%).

Beneath George VI Ice Shelf (Figure 16d), FESOM displays inflow at the southern end of the channel in Ronne Entrance, and outflow at the northern end into Marguerite Bay. This circulation agrees with the direction of net transport inferred from observations (Jenkins and Jacobs, 2008). Some outflow is also apparent in Ronne Entrance, as part of an anticyclonic gyre, but the rest of the gyre splits off and flows through the cavity. These circulation patterns are similar in both FESOM simulations, but they are stronger at high resolution. In MetROMS, south-to-north transport is apparent along the western edge of the

cavity, but also north-to-south transport along the eastern edge. This channel-shaped cavity is narrower and not well resolved by MetROMS, sometimes only 2 grid boxes wide.

### 4.3.8   Larsen Ice Shelves

The Larsen Ice Shelves on the eastern coast of the Antarctic Peninsula are undergoing a period of dramatic change, with the collapse of Larsen A in 1995 and Larsen B in 2002 (Rott et al., 1996; Rack and Rott, 2004), followed by a major calving of

Larsen C in July 2017 (Hogg and Gudmundsson, 2017). However, these breakup events are thought to be mainly driven by atmospheric processes rather than basal melting (Pritchard et al., 2012). Basal processes are actually thought to stabilise the Larsen C Ice Shelf through the production of marine ice by refreezing (Holland et al., 2009). Additionally, tidal forcing is likely to be important for the spatial distribution of melting (Mueller et al., 2012).





For all three simulations presented here, simulated basal mass loss for the Larsen C and Larsen D Ice Shelves falls within the range of estimates given by Rignot et al. (Table 1.8). For the larger Larsen C Ice Shelf, basal mass loss is approximately doubled in the low-resolution FESOM simulation compared to MetROMS, with a further increase of approximately 50% in high-resolution FESOM. Note that the RTopo-1 dataset used to generate the model domains does not include Larsen A or

Larsen B, as it was published following their collapse.

Ice shelf melt rate (a), vertically averaged velocity (b), and bottom water temperature (c) are shown for all three simulations in Figure 17. The stronger melting in FESOM corresponds to stronger circulation, travelling from north to south beneath the ice shelf. These differences are due to the lower summer sea ice concentration along the peninsula in FESOM compared to MetROMS, as seen in Figure 6. The absence of sea ice means the ocean surface is less sheltered from wind stress, and develops

stronger seasonal currents which extend into the cavity. Additionally, more AASW is allowed to develop in FESOM compared to MetROMS, as seen in Figure 9h. By contrast, wintertime area-averaged melt rates are similar in all three simulations (not shown). The low-resolution FESOM simulation also has warmer bottom water and more MCDW in the cavities than either MetROMS or high-resolution FESOM, consistent with the southward excursion of the southern boundary of the ACC as discussed in Section 4.2.2.

Observations suggest extensive areas of refreezing beneath the southern Larsen C Ice Shelf (Holland et al., 2009; Rignot et al., 2013). Both FESOM simulations show small regions of refreezing here, but not to the same extent as observations. MetROMS shows no regions of net refreezing at all. This discrepancy may be due to the lack of small-scale ice shelf thickness variability in the smoothed ice shelf drafts.

## 5   Discussion

Despite large variations in ocean/ice-shelf interaction, sub-ice shelf circulation, and continental shelf processes across different regions of the Antarctic coastline, several consistent themes have emerged in the simulation of these regions by the MetROMS and FESOM models. In some cases these patterns can be directly linked to model design, and can therefore provide guidance for future model development. In other cases, the interaction of several different model design choices makes this attribution more difficult.

Apparent in nearly all regions is the influence of sea ice on ocean/ice-shelf interactions. Since many of the prominent Southern Ocean water masses are governed by sea ice formation and melt, it is not surprising that simulated sea ice would have a discernible effect on the processes in ice shelf cavities. First, the location and rate of sea ice formation impacts the properties of shelf water masses flowing into the ice shelf cavities. MetROMS generally exhibits weaker dense water formation than either FESOM simulation, as evidenced by shallower mixed layers on the Antarctic continental shelf (Figure 3) and

a reduced presence of HSSW in ice shelf cavities (Figure 9). For the Filchner-Ronne and Ross Ice Shelves, this results in colder, fresher cavities with generally lower basal melt rates. Depending on the ice shelf, this may lead to better or worse agreement with observed basal mass loss. In FESOM, the excessive volume of HSSW produced by sea ice formation in the Filchner Depression is likely the culprit for its reversed direction of transport through the FRIS cavity. In the Amundsen and





Bellingshausen Seas, FESOM has unrealistically deep mixed layers driven by sea ice formation, which fill the shelf with LSSW and extract the heat of any warm bottom layer. This contributes to the cold bias in these regions and the underestimation of basal melt rates. This mechanism is known to be sensitive to the atmospheric forcing (Petty et al., 2013, 2014; Nakayama et al., 2014), although there is clearly some model-dependence since these deep mixed layers are not present in MetROMS.

The relationship between sea ice production, cross-shelf CDW transport, and ice shelf basal melting was also investigated by Timmermann and Hellmer (2013) in the context of future climate projections.

Simulated summer sea ice extent is too low in all three simulations, which is a common problem among sea ice models (Turner et al., 2013b). A larger area of the surface ocean is exposed to surface heating, which then drives increased summertime melting of ice shelf fronts. This behaviour is exacerbated by smoothing of the ice shelf front which allows the warm
surface waters to slide further back into the cavity, as seen for the Ross Ice Shelf in both models and the Amery Ice Shelf in FESOM. The question of ice shelf smoothing will be discussed later in this section, but it is also worth considering the potential causes of low summer sea ice and how this might be ameliorated. The fact that both models underestimate the sea ice minimum by a similar amount, despite MetROMS' more sophisticated sea ice physics, suggests that atmospheric forcing could be part of the problem. Atmospheric general circulation models (GCMs) consistently overestimate the amount of solar
radiation reaching the Southern Ocean in summertime due to biases in cloud cover (Trenberth and Fasullo, 2010), and while atmospheric reanalyses such as ERA-Interim perform better than GCMs, some biases persist (Naud et al., 2014). For future work, it would be worthwhile to force MetROMS or FESOM with the output of a high-resolution regional atmospheric model, such as RACMO (Regional Atmospheric Climate Model) which has previously been used for downscaling ECMWF reanalyses over Antarctica (Lenaerts et al., 2012). The resulting impact (if any) on cloud cover, radiation, and summer sea ice extent would
be useful to quantify. Another potential contributor to low summer sea ice in our simulations is the fact that grounded icebergs are not considered. Grounded icebergs increase the extent of fast ice, which is anchored to the coast and better survives the summer melt (Fraser et al., 2012).

Another model characteristic which influences ice shelf melt rates is the degree of smoothing of the bathymetry and the ice shelf draft. This problem is more severe in FESOM, which for a given horizontal resolution requires more smoothing than
MetROMS to ensure numerical stability, but it is also apparent for MetROMS at the Ross Ice Shelf front. Indeed, steep ice shelf fronts are some of the most affected regions, as smoothing allows warm AASW to slide into the ice shelf cavity and cause seasonal spikes in melt rates. However, a lack of observations means that the true amplitude of this seasonal cycle is unknown, and strong summertime melting is not necessarily unrealistic. This smoothing might also partially compensate for the absence of tides in both models, given that tidal processes similarly drive water mass exchanges across the ice shelf front.
Another environment in which smoothing is problematic is the continental shelf break, which is too gently sloping in some regions of the FESOM mesh, such as near the Fimbul Ice Shelf (Figure 11c). This allows MCDW to mix up the slope more easily, and indeed FESOM has a greater presence of MCDW than MetROMS in many ice shelf cavities (Figure 9). Finally, the steeply sloping Amery Ice Shelf draft is a particularly challenging feature for the FESOM mesh to adequately represent without considerable computational expense. In order to satisfy steepness limitations, the deep ice near the back of the cavity



shoals, particularly in the coarse-resolution mesh, which raises the in-situ freezing point at the ice shelf base and contributes to reduced melting.

All three of these problems improve with increased resolution, as less smoothing is required. High-resolution FESOM represents steep ice shelf drafts more accurately than low-resolution FESOM, leading to fewer AASW intrusions beneath the
Ross and Amery Ice Shelf fronts, and higher melt rates near the back of the Amery. The continental shelf break is also better preserved, and as a result the proportion of MCDW in ice shelf cavities decreases in almost every sector (Figure 9). Increased resolution is a straightforward solution to oversmoothing, and FESOM's unstructured mesh is ideal for targeting problematic regions without requiring increased resolution everywhere. However, since the maximum stable timestep of the model is a function of the smallest element rather than the average element, the impact of this approach on computational cost can still be
substantial, as we found when experimenting with resolution in the Amery Ice Shelf cavity. Therefore, in the future it may be worthwhile to experiment with different topographic smoothing methods, which may uncover options to minimise the trade-off between numerical stability and geometric accuracy.

Another way in which resolution impacts ice shelf melt rates in FESOM is by affecting the strength of sub-ice shelf circulation and the friction velocity at the ice shelf base. The high-resolution FESOM simulation is able to resolve circulation patterns
beneath small ice shelves, such as the Totten, Pine Island, and George VI Ice Shelves, whereas the low-resolution simulation shows more or less stagnant cavities with lower melt rates. Furthermore, the well-studied impact of resolution on Amundsen Sea CDW intrusions (Nakayama et al., 2014) is apparent in our simulations, with bottom water temperature on the continental shelf increasing by up to $1°C$ as a result of better-resolved troughs in the bathymetry. However, both FESOM simulations still have a cold bias in the Amundsen Sea, due to deep wintertime mixed layers in this region. Convection fills the continental shelf
with LSSW and erodes any CDW intruding into the bottom layer. This behaviour is driven by sea ice formation, and has been shown to be sensitive to the atmospheric dataset used (Nakayama et al., 2014). The relatively deep transition between sigma and z coordinates in our FESOM simulations is also known to inhibit CDW intrusions. Finally, Stewart and Thompson (2015) found that mesoscale eddies are vital to onshore CDW transport. None of our simulations resolve eddies on the Antarctic continental shelf, which would require resolution of approximately 1 km (St-Laurent et al., 2013).

FESOM's hybrid vertical coordinate system, with z-coordinates in most of the domain, is advantageous for the accurate simulation of the interior Southern Ocean and the ACC. Both FESOM and MetROMS exhibit some erosion of AAIW during the simulation, but the erosion is more severe in MetROMS due to spurious diapycnal mixing associated with terrain-following coordinates in the deep ocean. This degradation of deep water masses could explain the relatively weak Drake Passage transport simulated by MetROMS. However, terrain-following coordinates as used by both MetROMS and FESOM have other benefits
on the continental shelf/slope and within ice shelf cavities. In particular, they avoid the considerable sensitivity of ice shelf melt rates to vertical resolution which is seen in z-coordinate models of ice shelf cavities (Mathiot et al., 2017; Gwyther, 2016).

None of our simulations include tides, which would increase ice shelf melt rates through increased mixing and higher friction velocities. Tidal amplitudes are particularly large in the Weddell Sea, and the inclusion of tides has been shown to have a significant effect on simulated FRIS melt rates (Makinson et al., 2011; Mueller et al., 2017). While tides do exist as an
option in the ROMS code, their implementation requires specifying tidal elevation and/or tidal currents at lateral boundaries,



from which the tidal signal propagates throughout the domain. In our MetROMS domain, the northern boundary at $30°$S requires a strong sponge layer of increased diffusivity and viscosity to prevent numerical instabilities, which is likely to modify any tidal signal specified at the boundary. Additionally, the ice shelf cavities are much further from the boundary than is typical for simulations investigating tide/ice-shelf interactions, which generally have smaller domains focusing on a single

ice shelf (Padman et al., 2009; Mueller et al., 2017). We experimented with tides in our MetROMS setup, but they triggered northern boundary instabilities leading to large oscillations in ACC transport and subsequently CDW upwelling. In FESOM, tides have only been applied in regional configurations such as the Ross Sea (Wang et al., 2013) using a similar method to ROMS. Nonetheless, future model development to successfully implement tides in our MetROMS and FESOM domains would be valuable, and in particular should improve MetROMS' simulated basal mass loss for FRIS. An alternative approach

better suited to large domains could be to implement the complete lunisolar tides of Thomas et al. (2001), which have been successfully incorporated into two global ocean models (Müller et al., 2010).

   Finally, we must acknowledge that while the three-equation parameterisation for ice-shelf/ocean interaction is widely used, it relies on turbulent transfer coefficients which are largely unconstrained by observations. Little is known about the basal roughness of ice shelves, and thus the spatial and temporal variations in the corresponding drag coefficient are not generally

included in models. The thermodynamics of the boundary layer at the ice-shelf/ocean interface are also influenced by the choice of vertical mixing scheme (Gwyther et al., 2015). A more sophisticated treatment of marine ice formation, such as the explicit frazil ice models of (Smedsrud and Jenkins, 2004) and (Galton-Fenzi et al., 2012), may improve simulated patterns of refreezing. Furthermore, alternative parameterisations of ice shelf basal melt are being explored by the community (Jenkins, 2011), which may provide valuable intercomparisons with the three-equation parameterisation in the future.

## 6   Conclusions

We have presented the first published model intercomparison of circumpolar Antarctic ocean/sea-ice/ice-shelf interactions over a realistic domain. While we find that both MetROMS and FESOM underestimate total basal mass loss from ice shelves, this is a regional bias largely confined to small, warm-cavity ice shelves which are not well resolved by the model configurations considered here. With respect to simulated sub-ice shelf circulation, some ice shelf cavities show agreement with the direction

of transport inferred from observations (such as the Ross Ice Shelf in both models, the Amery Ice Shelf in MetROMS, and the George VI Ice Shelf in FESOM) and others show disagreement (such as the George VI Ice Shelf in MetROMS, and the Amery and Filchner-Ronne Ice Shelves in FESOM). FESOM's simulation of ice shelf cavities improves at higher resolution, suggesting that further refinement of resolution is justified as available computational power continues to increase. Sea ice in both MetROMS and FESOM mostly agrees with observations, although both models underestimate the summer sea ice

minimum, and MetROMS requires surface salinity restoring to prevent a spurious open-ocean polynya from forming in the Weddell Sea. In the interior Southern Ocean and the ACC, FESOM has an advantage due to its vertical coordinate system. We conclude that realistic intercomparisons of simulated ice shelf cavities are valuable for guiding model development. Future




studies including a greater variety of models, more reliable atmospheric forcing, and ideally more observations, would be worthwhile as these coupled ocean/sea-ice/ice-shelf models continue to be developed by the community.

*Code availability.* Access to the ice shelf branch of MetROMS is available upon request from Kaitlin Naughten (k.naughten@unsw.edu.au), and the FESOM source code from Ralph Timmermann (ralph.timmermann@awi.de).

*Competing interests.* The authors declare that they have no conflict of interest.

*Acknowledgements.* We would like to thank Michael Dinnimann, Paul Holland, Petra Heil, Elizabeth Hunke, Matt Mazloff, and Xylar Asay-Davis for their advice while tuning and debugging MetROMS, and Nicholas Hannah for technical support. Marta Kasper, Yoshihiro Nakayama, and Lukrecia Stulic provided assistance with installing and configuring FESOM, and Dmitry Sein with generating the FESOM mesh. We are also grateful to Sergey Danilov and Qiang Wang for helpful discussions regarding effective resolution and pressure gradient
calculation in FESOM. Finally, Jean-Baptiste Sallée and Violaine Pellichero provided observations of climatological mixed layer depth. This research was supported by an Australian Government scholarship under the Australian Postgraduate Award and Research Training Program schemes, a UNSW Research Excellence Award, and UNSW Science Silver Star and Gold Star Awards. Support was also provided by the Australian Research Council's Centre of Excellence for Climate System Science, and the Australian Government's Cooperative Research Centre Programme through the Antarctic Climate & Ecosystems Cooperative Research Centre. Computational resources were provided by
the NCI National Facility at the Australian National University, through awards under the Merit Allocation Scheme, the Intersect allocation scheme, and the UNSW HPC at NCI Scheme. Ben Galton-Fenzi is supported under the Australian Research Council's Special Research Initiative for the Antarctic Gateway Partnership SRI40300001, which also contributes to the World Climate Research Programme (WCRP) Climate and Cryosphere (CliC) project Targeted Activity "Linkages Between Cryosphere Elements". Ralph Timmermann acknowledges funding by the Helmholtz Climate Initiative REKLIM (Regional Climate Change), a joint research project of the Helmholtz Association of
German Research Centres (HGF).



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





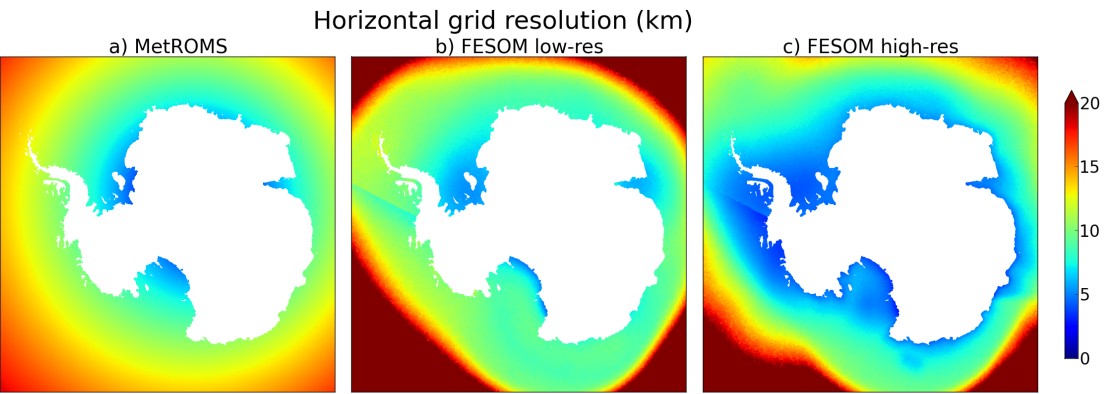

**Figure 1.** Horizontal resolution (km) of the MetROMS grid and both FESOM meshes around Antarctica. Resolution is defined as the square root of the area of each grid box (MetROMS) or triangular element (FESOM). Note that values above 20 km are not differentiated.





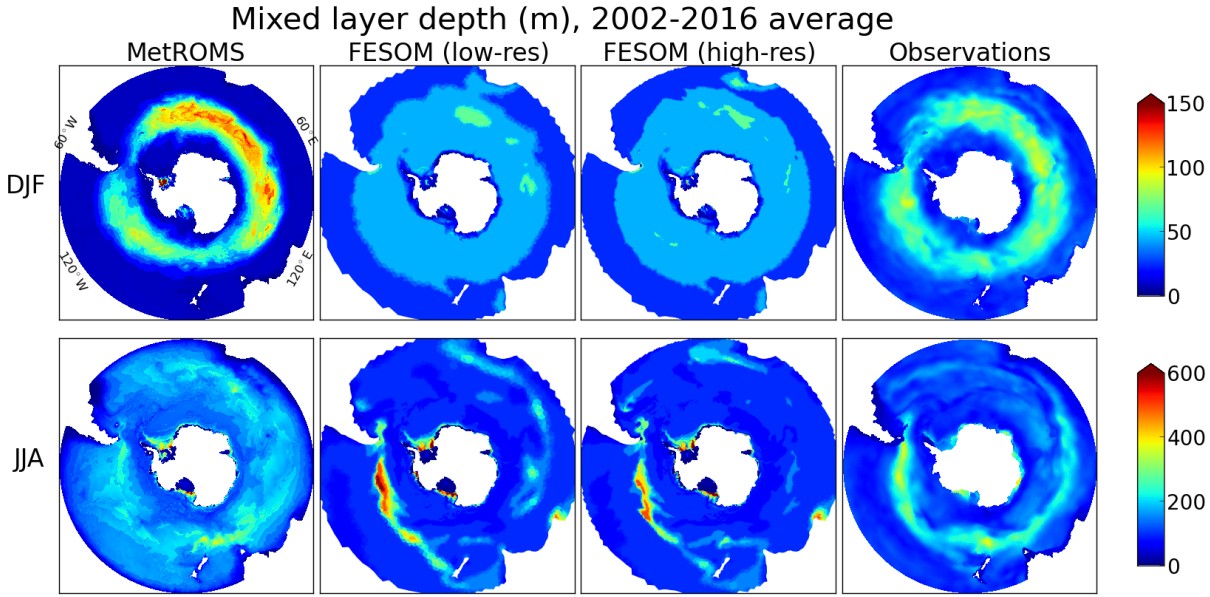

**Figure 2.** Mixed layer depth (m), calculated as the shallowest depth where potential density is at least 0.03 kg/m$^3$ greater than at the surface or ice shelf base. Results are shown for MetROMS, low-resolution FESOM, and high-resolution FESOM averaged over the years 2002-2016 for summer (DJF) and winter (JJA), as well as climatological observations by Pellichero et al. (2017) recalculated with the same definition of mixed layer depth. Note the different colour scale for summer and winter.





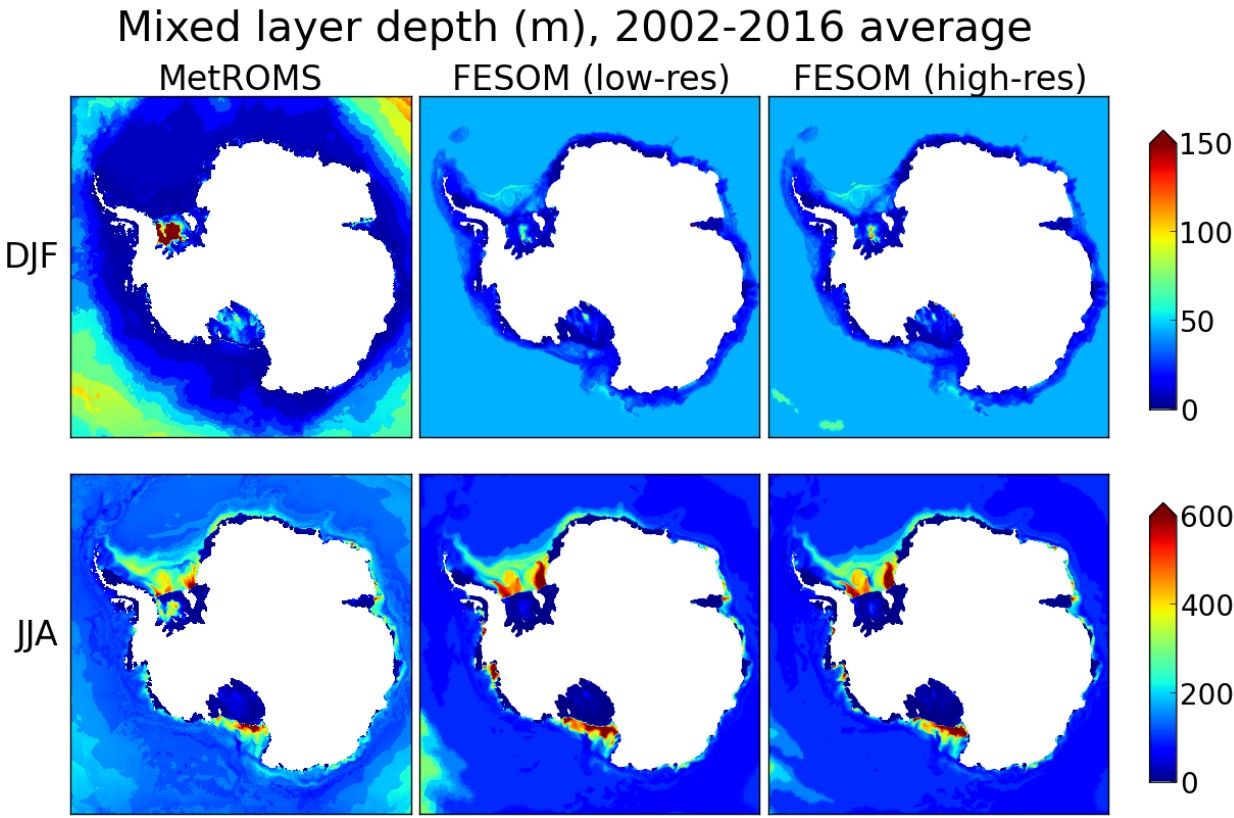

**Figure 3.** As Figure 2 for each model simulation, zoomed into the Antarctic continental shelf.





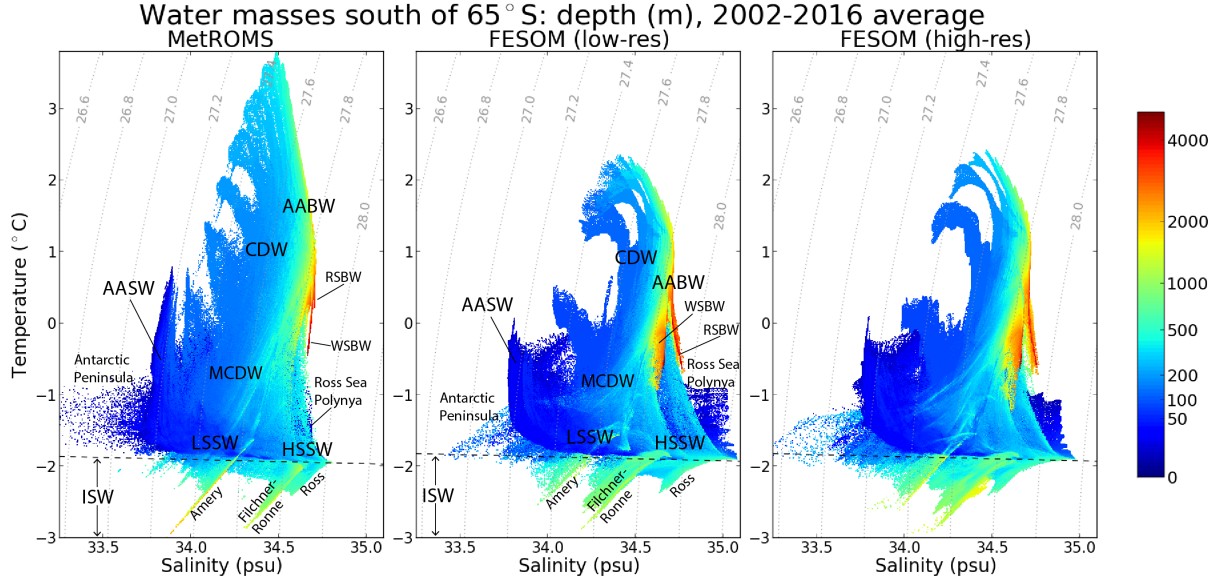

**Figure 4.** Temperature-salinity distribution south of $65°$S for MetROMS, low-resolution FESOM, and high-resolution FESOM, averaged over the years 2002-2016, and coloured based on depth (note nonlinear colour scale). Each grid box (in MetROMS) or triangular prism (in FESOM) is sorted into $1000 \times 1000$ temperature and salinity bins. The depth shown for each bin is the volume-weighted average of the depths of the grid boxes or triangular prisms within that bin. The dashed black line in each plot is the surface freezing point, which has a slightly different formulation between MetROMS and FESOM due to the different sea ice thermodynamics schemes. The dotted grey lines are potential density contours. Labels show different water masses: AABW = Antarctic Bottom Water, WSBW = Weddell Sea Bottom Water, RSBW = Ross Sea Bottom Water, CDW = Circumpolar Deep Water, MCDW = Modified Circumpolar Deep Water, LSSW = Low Salinity Shelf Water, HSSW = High Salinity Shelf Water, AASW = Antarctic Surface Water, ISW = Ice Shelf Water. Slanted labels below the freezing point line show specific ice shelves' contributions to ISW.





**Figure 5.** Temperature in °C (left) and salinity in psu (right) interpolated to 0°E (Greenwich Meridian). Black contours show the 0.75°C isotherm and the 34.5 psu isohaline. (a) Initial conditions for January 1992, from the ECCO2 reanalysis (Menemenlis et al., 2008; Wunsch et al., 2009). (b), (c), (d) January 2016 monthly average for MetROMS, low-resolution FESOM, and high-resolution FESOM respectively.





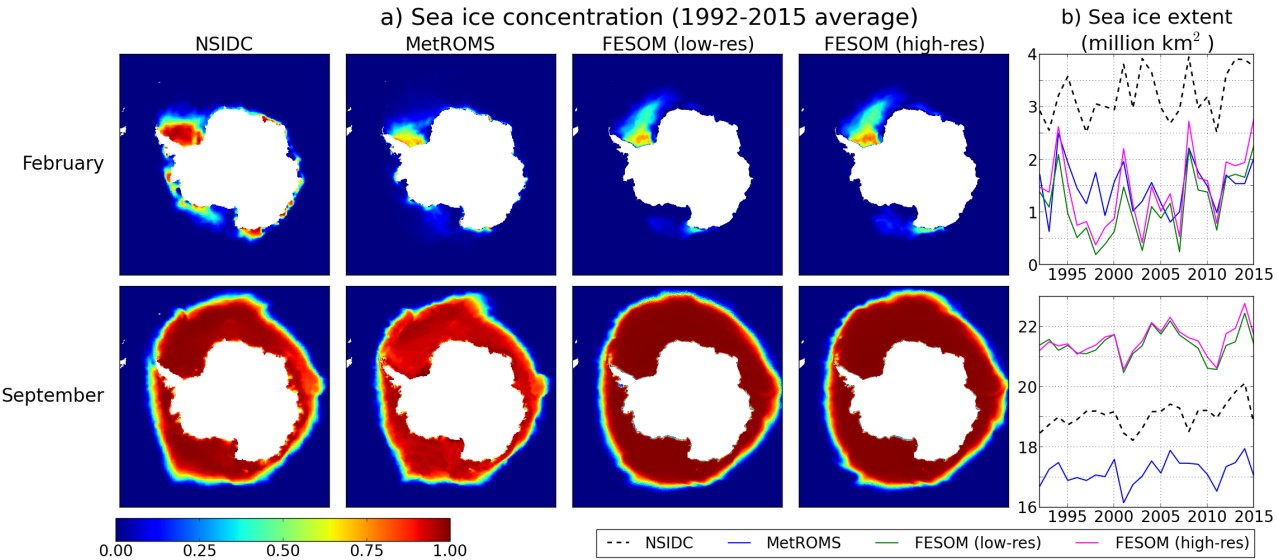

**Figure 6.** (a) 1992-2015 mean Antarctic sea ice concentration for February (top) and September (bottom), comparing NSIDC observations (NOAA/NSIDC Climate Data Record of Passive Microwave Sea Ice Concentration) (Meier et al., 2013), MetROMS, low-resolution FESOM, and high-resolution FESOM. (b) Timeseries of total Antarctic sea ice extent in millions of $km^2$ for February (top) and September (bottom), comparing NSIDC observations (NSIDC Sea Ice Index version 2) (Fetterer et al., 2016), MetROMS, low-resolution FESOM, and high-resolution FESOM.





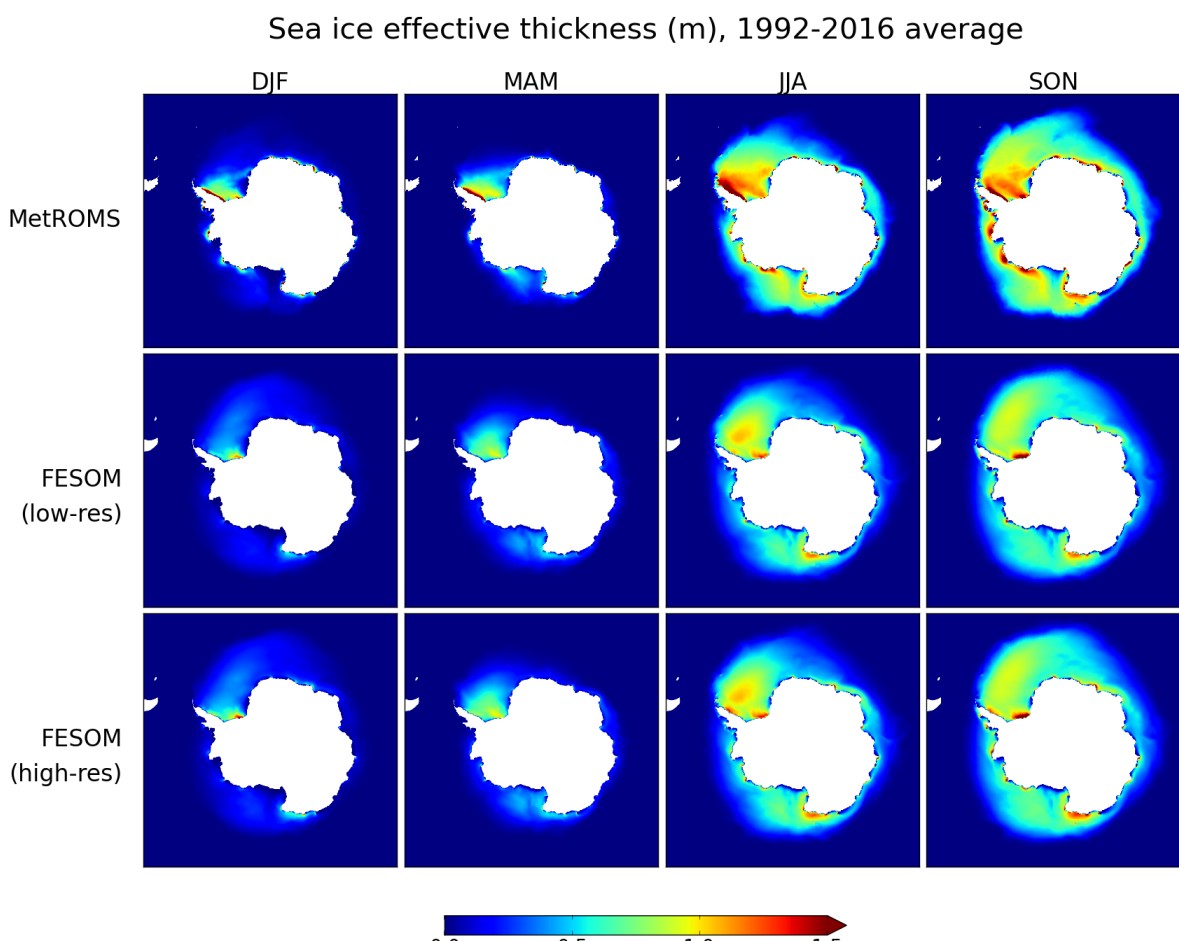

**Figure 7.** 1992-2016 mean seasonal Antarctic sea ice effective thickness (concentration times height, measured in metres) for MetROMS, low-resolution FESOM, and high-resolution FESOM.





**Table 1.** Ice shelf basal mass loss (Gt/y) for all ice shelves with area exceeding 5000 km$^2$ as measured by Rignot et al. (2013). In some cases multiple ice shelves have been combined (eg Brunt & Riiser-Larsen) because the boundaries between them in the model domains are not distinct. The ice shelves have been sorted into the eight regions analysed in Section 4.3. Values are shown for the MetROMS, low-resolution FESOM, and high-resolution FESOM simulations averaged over the years 2002-2016, and are compared to the range of observational estimates given by Rignot et al. for the period 2003-2008. Mass loss values from model simulations are marked with (-) or (+) if they fall below or above (respectively) the range given by Rignot et al..

| | MetROMS | FESOM low-res | FESOM high-res | Rignot et al. |
|---|---|---|---|---|
| **1. Filchner-Ronne** | 46.0 (-) | 113.8 | 115.4 | $155.4 \pm 45$ |
| **2. Eastern Weddell Region** | | | | |
| Brunt & Riiser-Larsen | 29.2 (+) | 33.4 (+) | 34.6 (+) | $9.7 \pm 16$ |
| Fimbul & Jelbart & Ekstrom | 30.3 | 41.8 (+) | 52.4 (+) | $26.8 \pm 14$ |
| Nivl | 3.4 | 5.4 | 5.9 | $3.9 \pm 2$ |
| Lazarev | 2.9 (-) | 4.9 | 4.9 | $6.3 \pm 2$ |
| Baudouin & Borchgevink | 28.4 | 35.7 | 36.5 | $21.6 \pm 18$ |
| Prince Harald | 5.4 (+) | 2.1 (+) | 2.6 (+) | $-2 \pm 3$ |
| **3. Amery** | 91.0 (+) | 71.0 (+) | 71.4 (+) | $35.5 \pm 23$ |
| **4. Australian Sector** | | | | |
| West | 11.5 (-) | 10.2 (-) | 12.7 (-) | $27.2 \pm 10$ |
| Shackleton | 14.3 (-) | 17.4 (-) | 22.7 (-) | $72.6 \pm 15$ |
| Totten & Moscow University | 9.5 (-) | 4.4 (-) | 9.3 (-) | $90.6 \pm 8$ |
| Mertz | 4.1 (-) | 2.6 (-) | 4.6 (-) | $7.9 \pm 3$ |
| **5. Ross Sea** | | | | |
| Ross | 53.8 | 95.1 (+) | 112.0 (+) | $47.7 \pm 34$ |
| Sulzberger | 14.0 (-) | 6.5 (-) | 9.2 (-) | $18.2 \pm 3$ |
| Nickerson | 5.3 | 2.0 (-) | 4.1 | $4.2 \pm 2$ |
| **6. Amundsen Sea** | | | | |
| Getz | 88.1 (-) | 21.3 (-) | 30.6 (-) | $144.9 \pm 14$ |
| Dotson | 9.1 (-) | 1.6 (-) | 3.7 (-) | $45.2 \pm 4$ |
| Thwaites | 7.4 (-) | 2.5 (-) | 5.9 (-) | $97.5 \pm 7$ |
| Pine Island | 20.5 (-) | 1.9 (-) | 9.5 (-) | $101.2 \pm 8$ |
| **7. Bellingshausen Sea** | | | | |
| Abbot | 25.0 (-) | 21.9 (-) | 36.3 | $51.8 \pm 19$ |
| Stange | 6.1 (-) | 5.1 (-) | 10.9 (-) | $28.0 \pm 6$ |
| George VI | 48.4 (-) | 14.0 (-) | 32.5 (-) | $89.0 \pm 17$ |
| Wilkins | 8.1 | 8.6 | 11.3 | $18.4 \pm 17$ |
| **8. Larsen Ice Shelves** | | | | |
| Larsen C | 18.2 | 35.2 | 54.7 | $20.7 \pm 67$ |
| Larsen D | 2.9 | 2.1 | 3.3 | $1.4 \pm 14$ |
| **Total Antarctica** | 642 (-) | 586 (-) | 739 (-) | $1325 \pm 235$ |





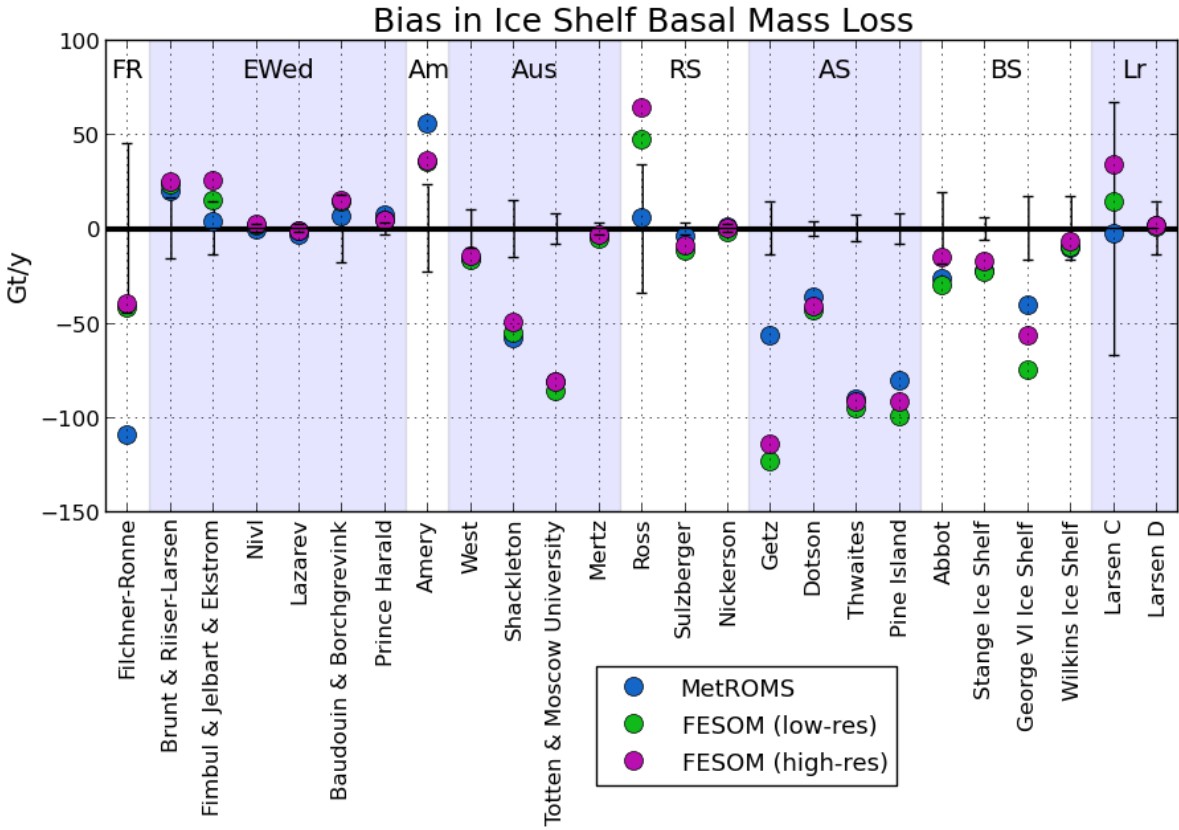

**Figure 8.** Difference between simulated ice shelf basal mass loss (2002-2016 average) and the central estimate given by Rignot et al. (2013) for each ice shelf in Table 1, in MetROMS (blue), low-resolution FESOM (purple), and high-resolution FESOM (green). The uncertainty ranges of Rignot et al. are also shown with black error bars. The eight regions specified in Table 1 are labelled as follows: FR = Filchner-Ronne, EWed = Eastern Weddell Region, Am = Amery, Aus = Australian Sector, RS = Ross Sea, AS = Amundsen Sea, BS = Bellingshausen Sea, Lr = Larsen Ice Shelves.

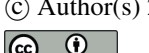



**Table 2.** Potential temperature ($T$) and salinity ($S$) ranges used to categorise water masses in ice shelf cavities in Figure 9. $T_f$ is the surface freezing point as in Figure 4. All acronyms are the same as in Figure 4.

|  | $T$ (°C) | $S$ (psu) |
|---|---|---|
| ISW | $T < T_f$ | |
| AASW | $T \geq T_f$ | $S < 34$ |
| LSSW | $T_f \leq T \leq -1.5$ | $34 \leq S < 34.5$ |
| HSSW | $T_f \leq T \leq -1.5$ | $S \geq 34.5$ |
| MCDW | $T > -1.5$ | $S \geq 34$ |

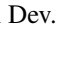

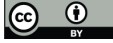

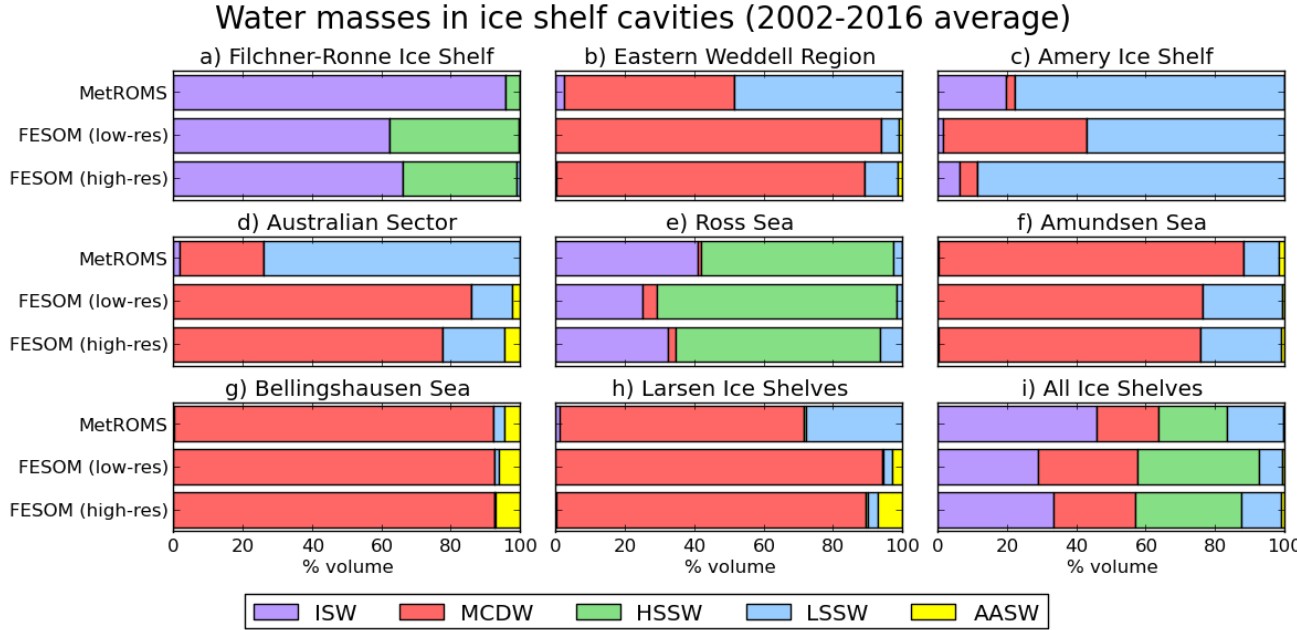

**Figure 9.** Proportions of different water masses (defined in Table 2) as percentage volumes in ice shelf cavities for each simulation, based on temperature and salinity fields averaged over 2002-2016. Results are shown for the eight regions specified in Table 1 (a-h) as well as the total for all Antarctic ice shelves (i). All acronyms are the same as in Figure 4.



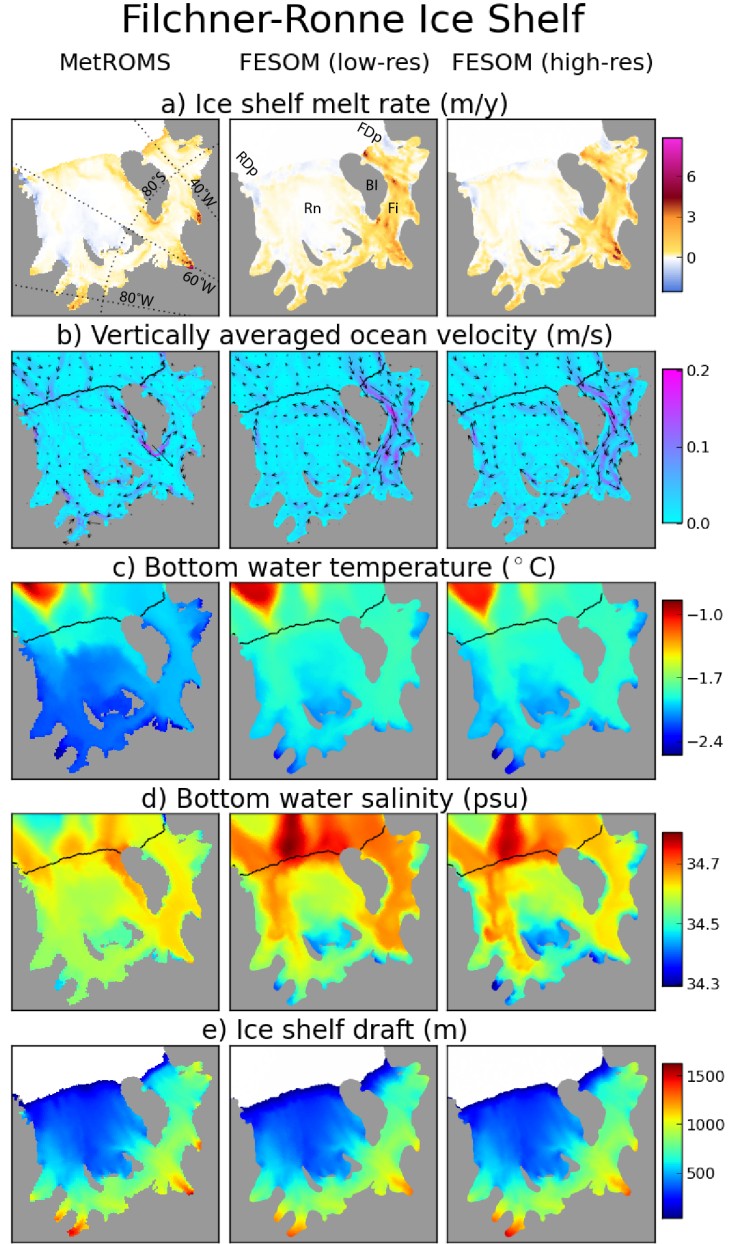

**Figure 10.** The Filchner-Ronne Ice Shelf cavity in MetROMS (left), low-resolution FESOM (middle), and high-resolution FESOM (right). All fields are averaged over the period 2002-2016. (a) Ice shelf melt rate (m/y). (b) Vertically averaged ocean velocity (m/s), where the colour scale shows magnitude and the arrows show direction. (c) Bottom water temperature (°C). (d) Bottom water salinity (psu). (e) Ice shelf draft (m) as seen by each model. In (b), (c), and (d), the ice shelf front is contoured in black. Rn = Ronne Ice Shelf, Fi = Filchner Ice Shelf, RDp = Ronne Depression, FDp = Filchner Depression, BI = Berkner Island.



**Figure 11.** (a), (b): As Figure 10a and 10b for the Eastern Weddell ice shelf cavities. (c) Temperature (°C) and (d) salinity (psu) interpolated to 1°W, through the Fimbul Ice Shelf. Br = Brunt Ice Shelf, RiL = Riiser-Larsen Ice Shelf, Ek = Ekström Ice Shelf, Je = Jelbart Ice Shelf, Fm = Fimbul Ice Shelf, Nv = Nivl Ice Shelf, Lz = Lazarev Ice Shelf, Bo = Borchgrevink Ice Shelf, Bd = Baudouin Ice Shelf, PH = Prince Harald Ice Shelf.





**Figure 12.** As Figure 10, for the Amery Ice Shelf cavity. PB = Prydz Bay.



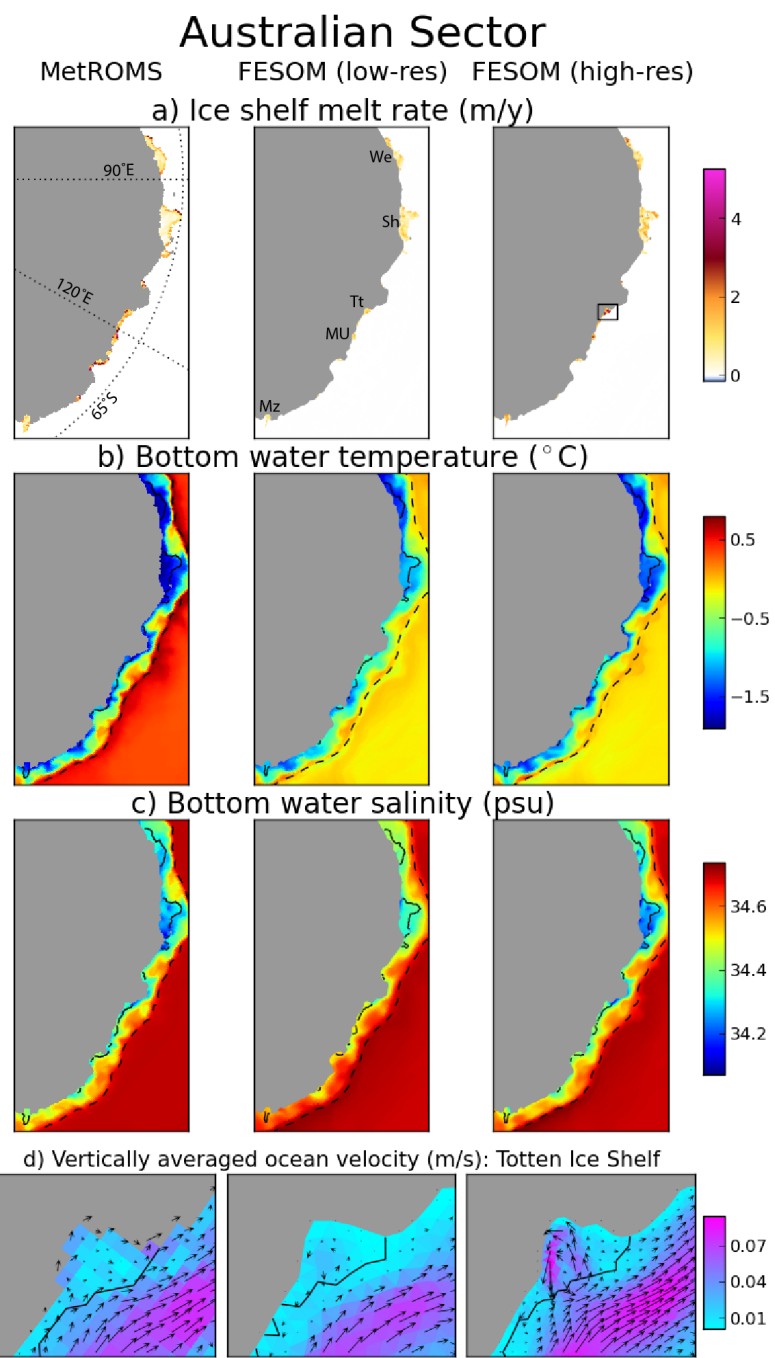

**Figure 13.** (a), (b), (c): As Figure 10a, 10c, and 10d for the Australian Sector ice shelf cavities. The dashed black lines in (b) and (c) show the 1500 m isobath, which approximates the continental shelf break. (d) As Figure 10b, zoomed into the Totten Ice Shelf cavity (region outlined in the rightmost panel of (a)). We = West Ice Shelf, Sh = Shackleton Ice Shelf, Tt = Totten Ice Shelf, MU = Moscow University Ice Shelf, Mz = Mertz Ice Shelf.



**Figure 14.** (a), (b): As Figure 10a and 10b for the Ross Sea ice shelf cavities. (c) Temperature (°C) and (d) salinity (psu) interpolated to 180°E, through the Ross Ice Shelf. Rs = Ross Ice Shelf, Sz = Sulzberger Ice Shelf, Nk = Nickerson Ice Shelf, McM = McMurdo Ice Shelf, RI = Roosevelt Island, CIR = Crary Ice Rise, SH = Steers Head.

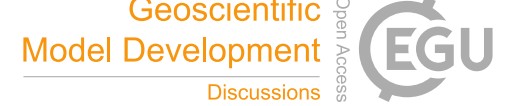



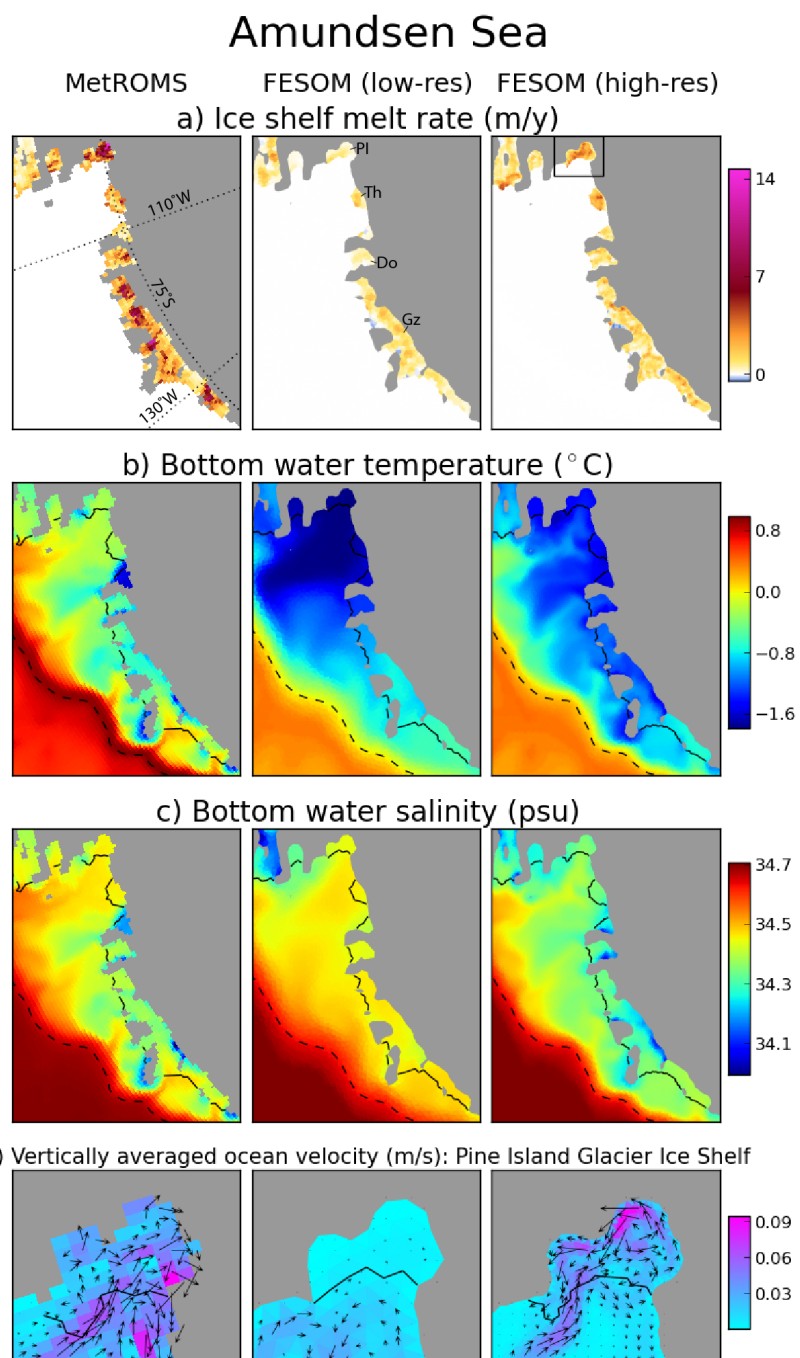

**Figure 15.** (a), (b), (c): As Figure 13a, 13b, and 13c for the Amundsen Sea ice shelf cavities. The dashed black lines in (b) and (c) show the 1500 m isobath. (d) As Figure 13d, zoomed into the Pine Island Ice Shelf cavity (region outlined in the rightmost panel of (a)). PI = Pine Island Ice Shelf, Th = Thwaites Ice Shelf, Do = Dotson Ice Shelf, Gz = Getz Ice Shelf.





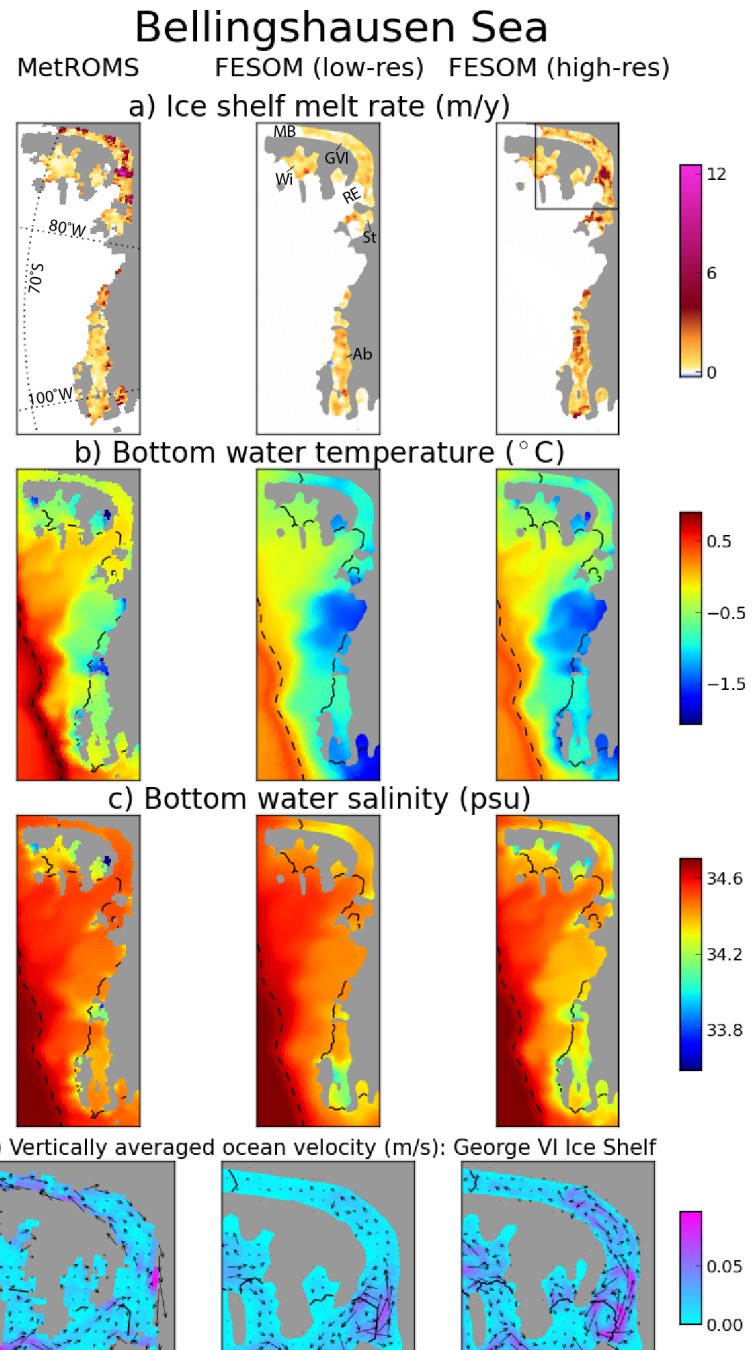

**Figure 16.** (a), (b), (c): As Figure 13a, 13b, and 13c for the Bellingshausen Sea ice shelf cavities. The dashed black lines in (b) and (c) show the 1500 m isobath. (d) As Figure 13d, zoomed into the George VI Ice Shelf cavity (region outlined in the rightmost panel of (a)). GVI = George VI Ice Shelf, Wi = Wilkins Ice Shelf, St = Stange Ice Shelf, Ab = Abbot Ice Shelf, MB = Marguerite Bay, RE = Ronne Entrance.



**Figure 17.** (a), (b), (c): As Figure 10a, 10b, and 13b for the Larsen ice shelf cavities. The dashed black line in (c) shows the 1500 m isobath.
LrC = Larsen C Ice Shelf, LrD = Larsen D Ice Shelf, AAP = Antarctic Peninsula.