# Peer review of "Intercomparison of Antarctic ice shelf, ocean, and sea ice interactions simulated by two models"

_Geoscientific Model Development, 2017_

## Referee Comment (RC1) · Anonymous Referee #1 · 7 Nov 2017

Reviewer comment: Kaitlin Naughten : Intercomparison of Antarctic ice shelf, ocean, and sea ice interactions simulated by two models. This paper is primarily about the ice-shelf melt rates from two different ocean models. The difference in sea ice and ocean states is also described but the interactions are merely hinted at rather than proven. The message is that with the only reference point between the models as the imposed surface forcing (from ERA-I) it is difficult to unravel the differences in behaviour of ice shelf basal melt. The models have not been spun-up and so the ice shelf cavity water masses may not be in equilibrium. It is thus difficult to assess how significant are the results. By allocating a large chunk of the paper to model assessment, it becomes unwieldy and overly long. (although in formatted text probably not much so than that

of Mathiot et al. also in GMD) The model assessment should be almost perfunctory to show that there are differences in the mean state and drifts. This allows the reader to focus on the ice shelf comparison, however, this section needs to be tidied up and given a specific focus to draw strongly form Figure 8 (expanded on below). I would not suggest any more model simulations, just a simplification of how this data is presented.

Specific comments

2:2-3 Please quote the more comprehensive observational study of the mass balance;

Shepherd, A. et al. A reconciled estimate of ice-sheet mass balance. Science 338, 1183–1189, 2012.

2:3 Golledge is a strange reference to use, a better summary of the processes is provided by

Joughin & Alley, Stability of the West Antarctic ice sheet in a warming world, Nature Geoscience, 4, 506–513 doi:10.1038/ngeo1194, 2011

2:14 It is true that there are not many such measurements, but tribute should be made to those that are done (at great expense) e.g. Nicholls et al., 2006; McPhail et al 2009; Venables et al., 2014

2:20 Add references to cavity models in GCMs;

Losch, M.: Modeling ice shelf cavities in a z coordinate ocean general circulation model, J. Geophys. Res.-Oceans, 113, C08043, https://doi.org/10.1029/2007JC004368, 2008.

Mathiot, P., Jenkins, A., Harris, C., and Madec, G.: Explicit representation and parametrised impacts of under ice shelf seas in the zâĹŮ coordinate ocean model NEMO 3.6, Geosci. Model Dev., 10, 2849-2874, https://doi.org/10.5194/gmd-10-2849-2017, 2017.

9:8 Actually FESOM uses a different version of EVP than Hunke and so I suggest you

instead reference Bouillon et al. (2013), as Danilov does.

9:12 Add a brief section on surface exchange scheme. Since these are forced models (ERA-I) then the difference in the bulk formula for surface exchange, particularly momentum, can contribute to differences in sea ice drift/formation, ACC transport and Ekman pumping/drift. A comparison of wind stress (on the ocean, not from ERA-I directly) from the two models would illustrate this.

9:27-30 The Intermediate waters which are pumped up on to the shelves will not be spun-up and since the initialization of these waters is based on almost no observations, the fidelity of the experiments is it doubt. However, this does not invalidate the model inter-comparison. I suggest a figure here which shows the far field anomaly of mean ocean temperature 300-1000m depth, compared with that of the initial conditions, for both models (see figure 14 of Mathiot et al., 2017).

10: 15-30 Consider including a brief note on the downside of salinity restoring – bulk salinity drift and impacts on ACC.

11:23 A lead in is required for this section describe the diagnostics that are going to be used and why. For example, the ACC "The Antarctic Circumpolar Current (ACC) is the most important current in the Southern Ocean, and the only current that flows completely around the globe. It is evaluated in models and observations at the Drake Passage, where by convention; all flow through the Passage is the ACC. The strength of the ACC is associated with the circumpolar winds (and the Southern Annular Mode - SAM) and the north-south oceanic salinity gradient. Various observations suggest the time-averaged Drake Passage transport is in between 134+-27 Sv (Cunningham et al., 2003) and 173+-11 Sv (Donohue et al., 2016)" – likewise for the MLD (continental shelf break mixing, heat exchange and sea ice formation in coastal polynyas) and sea ice properties.

12:1. Forced ocean simulations have significant drifts in the ACC because the bulk formulae do not constrain the surface fresh water balance, even with salinity restoration,

and hence the density structure changes. A stronger note of caution about validating these models with no spin-up is required.

13:17 – 15:34 There really is very little point in discussing the water mass properties and deep ocean when the model has not spun-up (a process requiring 300+yrs)! These do not add much to the discussion and should be removed to restrict the intercomparison to near surface characteristics.

16:11 Turner (2013) is refereeing to coupled models where the issue is more likely associated with the coupled forcing and mixed –layer depth, not due to the sea ice model itself. It is thus misleading to attribute this to the sea ice model, when it is rather the response of the sea ice to the climate model forcing. In the case of a standalone ocean model, where the surface forcing should act to restore the sea ice to that in ERA-I, such a lack of summer sea ice serious deficit in the models or bulk formula.

16:16-19 Delete. It is not necessary to reiterate that the sea ice concentration is a time average (as it is for the inferred observations).

16:21-22 This must happen because ERAI surface fluxes are attempting to restore sea ice to observations, so providing the mechanism them makes the variability self explanatory.

16:23-26 The difference between high and low resolution is hardly significant in Fig 6b and not worthy of mention unless the bathymetry is causing the low ice extent in the first place. That ACC water is entering the Weddell Sea gyre from this direction would be unusual, and as a consequence be worth discussing as a model bias.

16:27-34 Comparing sea ice concentrations is not valid unless there is a strong point to make. The observations are uncertain to 0.07 anyway and the concentration is consequently capped. The point about strong winter heat loss causing a sensible heat polynya and deep convection is supported by:

Goosse, H. and Fichefet, T.: Open-ocean convection and polynya formation in a largescale ice-ocean model, Tellus A, 53, 94–11, doi:10.1034/j.1600-0870.2001.01061.x, 2001.

Timmermann, R. and Beckmann, A.: Parameterization of vertical mixing in the Weddell Sea, Ocean Model., 6, 83–100, doi:10.1016/S1463-5003(02)00061-6, 2004

Consequently, the authors could strengthen this argument, but it is really a sideline which should be avoided and allow a stronger continuity to the paper.

17:7-11 Compared with coupled models the ice extents are pretty good, but they are still outside the decadal variability in the observations. Consequently the difference must be related to SST. This could be due to the path of the ACC (bathymetry or bulk formulae) or even due to the salinity restoration method being applied differently in the two models.

17: 15 Ice thickness is indeed a good means to inter-compare processes between models. I would be far more concerned with the thick ice in FESOM (and MetROMS) sitting directly in front of the Filchner_Ronne ice shelf. This should be a region of thin ice in JJA due to the winter latent heat polynya, which will not form in the model due to the thick ice. Consequently the Weddell sea ice formation and drift would be abnormal as is evidenced by the ice extent and thickness. On the other hand the Ross Ice-shelf polynya looks fine.

17:29 This launch into the cavity melt is too abrupt. It would be better to start with how this is assess observationally and the uncertainties (the Deporter and Rignot estimates are easily inside the error bounds and no comparison is required).

17:29-18:34 The intriguing aspect of figure 8 is that both models deviate from Rignot in the same direction. We would expect differences associated with the individual model resolution (vertical and horizontal), bulk formulae of momentum at the continental shelf break, basal melt parameterisations (and bathymetry). A description of possible sources of discrepancies is required before launching into the individual shelf

regions. For both models to deviate from Rignot in the same way either says that there are regional specific biases (e.g. lack of tides in models), ERAI is regionally biased from an unknown reality, or that Rignot is methodology is flawed. The latter could be tested by determining if the entire pattern changes with the Depoorter results. A comparison with Table 3 of Mathiot et al. (2017) might also be instructive.

19-25 The descriptions of ice shelves by region This type of analysis lends itself to repetition and long-winded descriptions. Instead I suggest that the analysis focus on specific ice shelves that show particular characteristics in Figure 8. For example Pine Island shows all the models closely clustered. The analysis should then show that the actual behavior of the models is quite different. Another might be the Ross where the melt rates are quite different and there is resolution sensitivity. Another might be, say, West, which is a poorly resolved small (shallow draft) ice shelf.

Keep the figures presented internally consistent, the style used for Figure 11 is good with a cross-section of water properties (including of-shelf) to reveal the bathymetry issues, perhaps showing the vertical overturning stream function and horizontal barotropic stream function. The small vertically integrated velocity vectors are not easy to interpret and if this approach is used a schematic representation of circulation may be better.

27:20-30:19 Discussion It is certainly true that all models, even ERAI show a cloud bias associated with cyclones, see:

Bodas-Salcedo, A., T. Andrews, A. V. Karmalkar, and M. A. Ringer (2016), Cloud liquid water path and radiative feedbacks over the Southern Ocean, Geophys. Res. Lett., 43, 10,938–10,946, doi:10.1002/2016GL070770.

Williams, K. D., and Coauthors, 2013: The Transpose-AMIP II experiment and its application to the understanding of Southern Ocean cloud biases in climate models. J. Climate, 26, 3258–3274, doi:https://doi.org/10.1175/JCLI-D-12-00429.1.

Include reference to (Graham et al., 2016) when discussing resolution.

Graham, JA, Dinniman, MS and Klinck, JM : Impact of model resolution for on-shelf heat transport along the West Antarctic Peninsula, J. Geophys. Res., 121, 7880-7897, doi: 10.1002/2016JC011875, 2016.

Refer to Mathiot et al., 2017 to compare with a pure z-level model.

Some discussion of the variability is required. Is the melt rate dominated by a few years of periodic high melt, or is the interannual variability small (probably varies according to distance from the shelf break and strength of the ocean baratropic circulation on shelf

---

## Short Comment (SC1) · 7 Nov 2017

Dear authors,

In my role as Executive editor of GMD, I would like to bring to your attention our Editorial version 1.1:

http://www.geosci-model-dev.net/8/3487/2015/gmd-8-3487-2015.html

This highlights some requirements of papers published in GMD, which is also available on the GMD website in the 'Manuscript Types' section:

http://www.geoscientific-model-development.net/submission/manuscript_types.html

In particular, please note that for your paper, the following requirements have not been met in the Discussions paper:

- "The main paper must give the model name and version number (or other unique identifier) in the title."

- "All papers must include a section, at the end of the paper, entitled 'Code availability'. Here, either instructions for obtaining the code, or the reasons why the code is not available should be clearly stated. It is preferred for the code to be uploaded as a supplement or to be made available at a data repository with an associated DOI (digital object identifier) for the exact model version described in the paper. Alternatively, for established models, there may be an existing means of accessing the code through a particular system. In this case, there must exist a means of permanently accessing the precise model version described in the paper. In some cases, authors may prefer to put models on their own website, or to act as a point of contact for obtaining the code. Given the impermanence of websites and email addresses, this is not encouraged, and authors should consider improving the availability with a more permanent arrangement. After the paper is accepted the model archive should be updated to include a link to the GMD paper."

As evaluation results also depend on the model and the model version, please provide the model names and their version numbers in the title of your article. Additionally, please ensure that the exact model versions, the evaluation was performed with, are permanently archived (best in a permanent archive providing a DOI (e.g. Zenodo)). Please make the model code also available to the public. If legal issues prevent this, please state the explicit reasons in the code availability section.

Yours,

Astrid Kerkweg

---

## Referee Comment (RC2) · X. Asay-Davis (Referee) · 3 Dec 2017

Review of Naughten et al. "Intercomparison of Antarctic ice shelf, ocean, and sea ice interactions simulated by two models"

Reviewer: Xylar Asay-Davis

I wish my name to be relayed to the authors, as I do not support the practice of anonymous review.

**General comments:**

This paper presents a comparison of two sea ice/ocean models with ice-shelf cavities, MetROMS and FESOM. Biases are analyzed in detail and across many regions where observations are available, including a discussion of directions for future model improvements that might reduce these biases. Where direct observations are not available, (e.g. for flow patterns under ice shelves), a comparison is made between models and biases are inferred from more indirect observations (e.g. the observed locations of inflow into and outflow from ice-shelf cavities). The paper is very well written, very clearly organized and makes for a compelling read. I feel it is nearly ready for publication, requiring only a few minor changes.

Perhaps the most significant change I would wish for the authors to consider is the addition of a figure of the rate of sea-ice production from each model, possibly compared with satellite-derived estimates (see my specific comment below). While the figure of mixed-layer depth serves as a proxy for this quantity, I feel like a figure showing sea-ice production would more directly get at the source of biases seen in both models that are inferred to come from too much or too little (or incorrectly located) sea-ice production in different regions.

As an aside, given that it is not (any longer) a requirement from GMD to put the figures and tables at the end of the manuscript, as a reviewer I would have liked to have the figures interspersed with the text for easier reviewing and reference. This is just something to keep in mind for any future submissions to a Copernicus journal.

**Specific comments:**

p. 2 l. 3 "The rate of retreat of much of the AIS will be governed by the ocean." While I agree that the ocean will almost certainly play an important role in AIS retreat, there are a couple of reasons I would suggest toning down this statement. First, internal ice dynamics and topography may govern the rate of retreat at least as much if not more than the ocean, with the ocean acting more as a trigger mechanism or an intermittent forcing. Second, we should not discount the potentially important role that atmospheric forcing will likely play in this process (e.g. via surface melting and potential hydrofracture).

p. 3 l. 27-29 "how many features of fluid flow can be represented by a mesh of a certain spacing" and "the number of features resolved". Flow features are not easily countable in the way this wording implies, and increasing resolution does not typically lead to a binary transition from unresolved to resolved. Instead, there is a messy transition from unresolved

through partially resolved to fully resolved.  I would suggest something like, "smallest flow features that are captured by a mesh of a certain spacing" and "the smallest resolved feature" for these.

p. 4 l. 4-5 "These differences... dominates" My experience with MISOMIP and my own pan-Antarctic modeling is that eddy-permitting and eddy-resolving simulations do *not* necessarily behave more similarly, and other modeling choices still play an important (if not dominant role) even when eddies are included.  For example, even fairly subtle differences in how topography is represented can lead to changes in how eddies are shed or small-scale currents interact with the topography.

p. 4 l. 21-22 and Fig. 5 "Bottom nodes are allowed to deviate from the standard z-levels in order to match the given bathymetry" This is not obvious in Fig. 5.  If this is a plotting artifact in Fig. 5, it would be best to fix that so the true nature of how FESOM represents bathymetry is shown in the figure.  If not, it is unclear why there are full-cell jumps in the bathymetry in Fig. 5, given that FESOM should have some kind of equivalent of the partial-cell methods used in other z-level models.

p. 4 l. 24-25 "extremely high vertical resolution." I would suggest avoiding subjective phrases like this that include an implicit value judgement.  What might seem today like "extremely high" vertical resolution is likely to become closer to standard resolution in the not-too-distant future.  We frequently run global ocean simulations with the Model for Prediction Across Scales Ocean (MPAS-O) including 1 meter vertical resolution in the upper mixed layer.

p. 4 l. 26 "with 30 barotropic timesteps for each baroclinic"  This is largely an aside for your future work, not a request for a change in the paper.  Did you experiment at all with fewer barotropic time steps per baroclinic?  If vertical resolution is controlling the time step, it should follow that the barotropic time step would not be strongly affected and fewer subcycles might be possible.

p. 6 l. 10-11 "very minimal spurious sea ice formation" maybe remove "very" (since it sounds kind of subjective)

p. 6 l. 10-12 I'm kind of confused by this sentence.  My understanding was that that the flux-limited advection scheme in Naughten et al. 2017 reduced the spurious sea-ice formation.  Do you perhaps mean "comparable to" instead of "compared to"?  Then, I would understand a bit better.

p. 7 l. 9-11 It is a little unclear what "no significant impacts on Weddell Sea convection" means in this context.  I take it to mean that KPP appears to work just as well as Pacanowski-Philander, at least over the 5 years.  If that is the case maybe a comment is warranted here to the effect that this deserves further investigation.  I'm a little unclear on what you (or I as the reader) should take from this.

p. 8 l. 5-6: The values of u* need units (presumably m/s)

p. 8 eq. (4): 530, 10^-3 and 10^-8 need units, since they appear to all be dimensional.

p. 10 l. 6-8: "which are not interpolated in time...as they represent total fluxes…" This is a fine approach but there would be ways to interpolate these data in time while preserving the 12-hourly mean.

p. 10 l. 13: "additional surface freshwater flux representing iceberg melt" Could you give a description in a sentence or 2 of what the characteristics are of this climatology? It seems like it comes from an iceberg model, rather than from observations, which is perfectly reasonable but probably deserves a mention.

p. 11 l. 6-7 "...taken from the AVISO climatology...which is a single time record." If I understand right, you use the annual mean rather than the monthly climatology? (Near as I can tell, both are available from AVISO.)

p. 11 l. 9 "in y" I would probably change this to "in latitude" even though I understand that "y" is not exactly the same as latitude over the whole domain. At the northern boundary, they are presumably the same and I think that would be clearer to the reader.

p. 14 l. 3-4 "In both models RSBW temperatures more or less agree with observations...". This seems a little generous to me. MetROMS is clearly missing some colder temperatures, while FESOM reaches temperatures that are significantly colder than observed in WOCE. I guess that is what the "more or less" is meant to cover.

p. 14 l. 16 LSSW isn't really analyzed here. Is this because Fig. 4 doesn't really have anything to say about this water mass?

p. 15 l. 24: "In FESOM, AAIW is slightly better preserved at low resolution than at high resolution." Do you care to speculate on why this is?

p. 20 l. 11: "extremely high resolution." Again, I would recommend a less subjective wording.

p. 21 l. 4 and Fig. 11a: I'm not sure what can be one but I found the figure to be too small to clearly see the features that are being described. When I zoomed in to 400% in the pdf viewer, the figure was kind of pixelated, meaning this didn't really help.

p. 22 l. 15: "steeper ice draft" I would also suggest that better resolved currents may be the reason.

p. 25 l. 2: "and this process requires resolutions of 1 km or less" This is stated in a couple of places. My experience is that model properties improve significantly even at eddy-permitting resolution (~2-4 km) even when the largest eddies are not fully resolved. My point is that eddy transport is not typically absent in eddy-permitting simulations, it is just diminished from what it would be at higher resolution.

p. 27 l. 27-28: "First, the location and rate of sea ice formation impacts the properties of shelf water masses flowing into the ice shelf cavities." You did a very thorough job of plotting a lot of fields from the 2 models. If there is one field I wished you'd included, it is the rate of sea-ice formation, perhaps comparing with a satellite-derived data set such as Tamura et al. (2016). I will not insist that you add such a plot but I would appreciate it if you would consider it at least.

p. 29 l. 10-12: "Therefore, in the future it may be worthwhile to experiment with different topographic smoothing methods, which may uncover options to minimise the trade-off between numerical stability and geometric accuracy." I would suggest adding to this that it might be worth investigating other numerical methods for computing the horizontal pressure-gradient force (HPGF) in FESOM, since this is an area of active research (Engwirda, Kelley, and Marshall 2017). I would have more confidence that improvements in the HPGF would lead to less topographic smoothing than that a better smoothing algorithm can solve the problem on its own.

p. 29 l. 23-24: "None of our simulations resolve eddies on the Antarctic continental shelf, which would require resolution of approximately 1 km (St-Laurent et al., 2013)." See my previous comment about 1 km resolution. I think the higher resolution FESOM simulation is probably at least eddy permitting. This is suggested at least by some of the features shown in the zoomed-in velocity plots. If this is the case, it might be worth mentioning. If not, it might be worth mentioning that none of the simulations is even eddy permitting.

p. 30 l. 18-19: "Furthermore, alternative parameterisations of ice shelf basal melt are being explored by the community (Jenkins, 2011), which may provide valuable intercomparisons with the three-equation parameterisation in the future." I would suggest including Jenkins (2016) here. I think this work is more likely to lead to an alternative to the three equations than the plume-model approach as in Jenkins (2011).

p. 30 l. 28-29: "Sea ice in both MetROMS and FESOM mostly agrees with observations..." Can you be more specific about which fields were compared with observations? For example, I don't think the rates and locations of sea-ice production are likely to agree with satellite-derived estimates (see my suggestion for a figure above), as you discuss in the context of mixed-layer depth in the ocean.

p. 30 l. 31: "In the interior Southern Ocean and the ACC, FESOM has an advantage due to its vertical coordinate system." I think it would be worth restating that FESOM's vertical coordinate is z-level (and perhaps also that MetROMS' is terrain-following) in this region.

p. 31 l. 1: "...more reliable atmospheric forcing..." I think this needs some clarification. What does this mean to you? I realize this has been covered in the discussion but it's worth summarizing in at least a little more detail here.

**Typographic and grammatical corrections:**

p. 1  l. 15-16: In my experience, it is customary to refer to a model's "terrain-following coordinate" or "z-coordinate" (both singular).  Plural would be appropriate if we were referring to coordinates in multiple dimensions (e.g. spherical coordinates).

p. 4 l. 26-29 (and possibly elsewhere) I would suggest using only "time step" and not "timestep".  You use both in this paragraph.

p. 8 l. 1, 4, 12, 15: There is an incorrect new paragraph on each of these lines causing an indentation.

p. 21 l. 26: "which has  one of the deepest ice shelf drafts in Antarctica" an extra "the"

**References**

Engwirda, Darren, Maxwell Kelley, and John Marshall. 2017. "High-Order Accurate Finite-Volume Formulations for the Pressure Gradient Force in Layered Ocean Models." *Ocean Modelling* 116 (August):1–15.

Jenkins, Adrian. 2016. "A Simple Model of the Ice Shelf–Ocean Boundary Layer and Current." *Journal of Physical Oceanography* 46 (6):1785–1803.

Tamura, Takeshi, Kay I. Ohshima, Alexander D. Fraser, and Guy D. Williams. 2016. "Sea Ice Production Variability in Antarctic Coastal Polynyas: ANTARCTIC SEA ICE PRODUCTION VARIABILITY." *Journal of Geophysical Research, C: Oceans* 121 (5):2967–79.

---

## Author Comment (AC1) · 20 Feb 2018

**Response to Reviewers**

This document is colour-coded as follows:

- Comments by reviewers are in **blue**.
- Our responses are in **black**.
- Blocks of text we have added to the manuscript are in **red**.

Note that all figure numbers refer to the new version of the manuscript. For Figures 9 and above, this is offset by one compared to the original submission, due to the addition of Figure 8.

**Reviewer 1**

This paper is primarily about the iceshelf melt rates from two different ocean models. The difference in sea ice and ocean states is also described but the interactions are merely hinted at rather than proven. The message is that with the only reference point between the models as the imposed surface forcing (from ERA-I) it is difficult to unravel the differences in behaviour of ice shelf basal melt. The models have not been spun-up and so the ice shelf cavity water masses may not be in equilibrium. It is thus difficult to assess how significant are the results. By allocating a large chunk of the paper to model assessment, it becomes unwieldy and overly long. (although in formatted text probably not much so than that of Mathiot et al. also in GMD) The model assessment should be almost perfunctory to show that there are differences in the mean state and drifts. This allows the reader to focus on the ice shelf comparison, however, this section needs to be tidied up and given a specific focus to draw strongly form Figure 8 (expanded on below). I would not suggest any more model simulations, just a simplification of how this data is presented.

We would like to thank the reviewer for these general comments, all of which are addressed in our responses to specific comments below.

Specific comments

2:2-3 Please quote the more comprehensive observational study of the mass balance; Shepherd, A. et al. A reconciled estimate of ice-sheet mass balance. Science 338, 1183–1189, 2012.

We have added this reference as suggested (page 2, line 3).

2:3 Golledge is a strange reference to use, a better summary of the processes is provided by Joughin & Alley, Stability of the West Antarctic ice sheet in a warming world, Nature Geoscience, 4, 506–513 doi:10.1038/ngeo1194, 2011

We have added this reference as suggested (page 2, line 4).

2:14 It is true that there are not many such measurements, but tribute should be made to those that are done (at great expense) e.g. Nicholls et al., 2006; McPhail et al 2009; Venables et al., 2014

We have added the following sentence to this paragraph (page 2, lines 16-17):

Nonetheless, some measurements have been made at great expense (e.g. Nicholls et al., 2006; McPhail et al., 2009; Venables and Meredith, 2014).

2:20 Add references to cavity models in GCMs;
Losch, M.: Modeling ice shelf cavities in a z coordinate ocean general circulation model, J. Geophys. Res.-Oceans, 113, C08043, https://doi.org/10.1029/2007JC004368, 2008.
Mathiot, P., Jenkins, A., Harris, C., and Madec, G.: Explicit representation and parametrised impacts of under ice shelf seas in the zâ´LU coordinate ocean model NEMO 3.6, Geosci. Model Dev., 10, 2849-2874, https://doi.org/10.5194/gmd-10-2849-2017, 2017.

In this paragraph we cite Dinniman et al. (2016), a review paper which references all existing ice shelf-ocean models known to its authors at the time of publication. This includes Losch et al. (2008). To make this clear, we have added "and references therein" to our citation of Dinnimann et al. (2016) (page 2, line 22). Since this publication predates Mathiot et al. (2017), we have also added this reference to our manuscript (page 2, line 22).

9:8 Actually FESOM uses a different version of EVP than Hunke and so I suggest you instead reference Bouillon et al. (2013), as Danilov does.

We have switched this reference as suggested (page 10, line 8).

9:12 Add a brief section on surface exchange scheme. Since these are forced models (ERA-I) then the difference in the bulk formula for surface exchange, particularly momentum, can contribute to differences in sea ice drift/formation, ACC transport and Ekman pumping/drift. A comparison of wind stress (on the ocean, not from ERA-I directly) from the two models would illustrate this.

We would like to thank the reviewer for this important point of comparison, which we had not previously investigated. We have added a brief section to the manuscript as suggested (page 10, lines 12-20):

**2.7 Surface exchange scheme**
While MetROMS and FESOM are forced with the same atmospheric state (see Section 3.2), the resulting surface fluxes differ based on the bulk formulae implemented by the models. Our configuration of FESOM uses constant exchange coefficients for heat and momentum fluxes, while MetROMS' exchange coefficients vary in time and space. For ocean/atmosphere fluxes (in ROMS), these coefficients are based on the COARE (Coupled-Ocean Atmosphere Response Experiment) protocol (Fairall et al., 1996). For sea-ice/atmosphere fluxes, CICE includes a stability-based atmospheric boundary interface (Hunke et al., 2015). These differences in bulk formulae may affect the simulations,

particularly the momentum fluxes which have consequences for ACC transport, Ekman pumping, and sea ice formation and drift. A comparison of ocean surface stress (not shown) reveals that these momentum fluxes are typically stronger in MetROMS, by up to 30%.

We have not included the surface stress comparison figure in the manuscript, but it is reproduced below. Results are shown for only the first year of each simulation, to minimise the influence of differently-drifting ocean states on the bulk fluxes.

Ocean surface stress (N/m$^2$), 1992 mean

[Figure]

We have also added several references to this section throughout the manuscript. In Section 4.1.1 (Drake Passage transport), page 13, lines 14-16:

Compared to the observations of Donohue et al. (2016), the values from all three of our simulations are too low, especially in MetROMS. This occurs despite MetROMS' stronger surface stress than in FESOM (Section 2.7).

In Section 4.1.2 (mixed layer depth), page 14, lines 14-15:

The tendency of MetROMS to have deeper mixed layers than FESOM may be influenced by the differing surface stress between the two models (Section 2.7).

In Section 4.1.4 (deep ocean drift), page 21, lines 26-29:

The cause of this increased CDW upwelling in MetROMS is not obvious. Warming and shoaling of CDW around most regions of Antarctica has been observed over recent decades, and attributed to changes in wind stress (Schmidtko et al., 2014; Spence et al., 2014, 2017). With these observations in mind, it is possible that this behaviour is due to MetROMS' surface exchange scheme, which leads to stronger surface stress than in FESOM (Section 2.7).

9:27-30 The Intermediate waters which are pumped up on to the shelves will not be spun-up and since the initialization of these waters is based on almost no observations, the fidelity of the experiments is it doubt. However, this does not invalidate the model inter-comparison. I

suggest a figure here which shows the far field anomaly of mean ocean temperature 300-1000m depth, compared with that of the initial conditions, for both models (see figure 14 of Mathiot et al., 2017).

We have created this figure, which is reproduced below. Note that regions with bathymetry shallower than 1000 m, and all ice shelf cavities, have been masked.

Change in temperature from initial conditions (°C), 300-1000 m average

[Figure]

We have decided not to include this figure in the manuscript as it does not convey any information which is not already present in Figure 5, apart from the spatial distribution of the temperature drift in this depth range. Figure 5 shows meridional slices of temperature and salinity through 0°E, which illustrates the drift in multiple water masses at different depths, including the erosion of AAIW which is only visible in the salinity field.

10: 15-30 Consider including a brief note on the downside of salinity restoring – bulk salinity drift and impacts on ACC.

We have added the following text to this section (page 12, lines 2-6):

Such restoring affects the salt budget and may contribute to drift in the total salt content of the ocean, although it prevents drift in the surface layer. This may impact the density structure of the Southern Ocean, and particularly the ACC, as well as damping interannual variability. However, these shortcomings were deemed preferable to spurious deep convection for the purposes of our analysis.

11:23 A lead in is required for this section describe the diagnostics that are going to be used and why. For example, the ACC "The Antarctic Circumpolar Current (ACC) is the most important current in the Southern Ocean, and the only current that flows completely around the globe. It is evaluated in models and observations at the Drake Passage, where by convention; all flow through the Passage is the ACC. The strength of the ACC is associated with the circumpolar winds (and the Southern Annular Mode - SAM) and the north-south oceanic salinity gradient. Various observations suggest the time-averaged Drake Passage transport is in between 134+-27 Sv (Cunningham et al., 2003) and 173+-11 Sv (Donohue et

al., 2016)" – likewise for the MLD (continental shelf break mixing, heat exchange and sea ice formation in coastal polynyas) and sea ice properties.

For Section 4.1.1 (Drake Passage transport), we have written a short introduction and reorganised the section so that observations are discussed before the model results (page 13, lines 4-10):

The ACC has the strongest transport of any ocean current in the world, and is key to the thermal isolation of Antarctica. Transport of the ACC is influenced by the Southern Hemisphere westerly winds as well as the density structure of the Southern Ocean. By convention, zonal transport of the ACC is evaluated through Drake Passage and is time-averaged to remove the seasonal cycle. With respect to observations, Drake Passage transport was previously thought to lie around 134 Sv (Cunningham et al., 2003). However, recent improvements in measuring systems have suggested a higher value. As part of the cDrake project (Dynamics and Transport of the Antarctic Circumpolar Current in Drake Passage), Donohue et al. (2016) estimated a Drake Passage transport of 173.3 +/- 10.7 Sv.

For Section 4.1.2 (mixed layer depth), we have written the following introduction (page 13, lines 24-27):

The surface mixed layer represents the portion of the ocean which is directly influenced by the atmosphere. The depth of the mixed layer is a key indicator of the strength of convection, and heat loss to the atmosphere resulting from convection will influence water mass properties. Regions of strong sea ice formation, such as coastal polynyas, are characterised by deep wintertime mixed layers.

For Section 4.1.3 (water mass properties), we have added a 1-sentence introduction (page 18, line 2):

Ice shelf melt rates and sea ice formation both influence, and are influenced by, water mass properties on the continental shelf.

For Section 4.2.1 (sea ice concentration and extent), we have written the following introduction (page 23, lines 3-6):

Sea ice concentration (the fraction of each grid cell covered by ice) and extent (the area of grid cells with concentration exceeding 0.15) are the most convenient variables for model evaluation, due to the availability of satellite observations. These variables are largely a reflection of atmospheric conditions, but are also influenced by ocean processes, such as upwelling of warmer water from below, and the pathway of the ACC.

For Section 4.2.2 (sea ice thickness), we have written the following introduction (page 26, lines 2-5):

Sea ice thickness is influenced by both thermodynamics (sea ice formation and melt) and dynamics (sea ice transport). Observations of sea ice thickness are scarce and have large

uncertainties (Holland et al., 2014). A comprehensive evaluation of MetROMS and FESOM with respect to sea ice thickness is therefore difficult, although a comparison of the two models can still be made.

12:1. Forced ocean simulations have significant drifts in the ACC because the bulk formulae do not constrain the surface fresh water balance, even with salinity restoration, and hence the density structure changes. A stronger note of caution about validating these models with no spin-up is required.

We have added the following text to this section (page 13, lines 19-22):

Furthermore, drift in the density structure may result from non-closure of the surface freshwater budget, which is globally unconstrained by the bulk-flux approach of our simulations. Since interior Southern Ocean processes operate on much longer timescales than our experiments, and would require long spin-ups to equilibrate, simulated ACC transport should be interpreted with caution and is not the focus of this manuscript.

13:17 – 15:34 There really is very little point in discussing the water mass properties and deep ocean when the model has not spun-up (a process requiring 300+yrs)! These do not add much to the discussion and should be removed to restrict the intercomparison to near surface characteristics.

Many of the water masses we discuss in this section are newly formed, rather than just determined by the initial conditions. Below is a temperature/salinity distribution (as in Figure 4) of the ECCO2 1992 annual average. Note that the January 1992 fields from ECCO2 were used as initial conditions for our model simulations.

Compared to the well-organised ISW seen in Figure 4 for MetROMS and FESOM, where characteristic sloped lines can be assigned to individual ice shelf cavities, ECCO2 shows no true ISW as it has no ice shelf cavities. Some water exists below the surface freezing point, but this is an artifact of data assimilation at the Antarctic coast. Furthermore, ECCO2 exhibits minimal HSSW which shows little connection to the Ross Sea Polynya or the (nonexistent) Ross Ice Shelf cavity. LSSW is restricted to the near-surface, in contrast to the filaments of approx. 400 m deep LSSW seen in MetROMS and FESOM. These three water masses are also known to be formed by processes with relatively short timescales, such as ice shelf melting (which stabilises within 5-10 years in our simulations) and water mass modification on the continental shelf. AASW also fits this criteria, as it is generally within the mixed layer. Therefore, we can consider ISW, HSSW, LSSW, and AASW to be newly formed, and not significantly influenced by the initial conditions. These are the water masses most relevant to the bulk of our manuscript.

[Figure]

The same cannot be said for the deeper water masses (AABW, CDW, and to some extent MCDW), which are not spun up, as noted by the reviewer. Nonetheless, a drifting field is still interesting and can tell us something about model behaviour. For example, Figure 4 clearly shows warmer CDW in MetROMS, which is due to increased southward spreading of this water mass, as expanded upon in Section 4.1.4. Additionally, RSBW has higher salinity in FESOM, consistent with its saltier Ross Sea Polynya and Ross Ice Shelf cavity. We thus do not agree that this entire discussion should be removed from the manuscript.

Instead, we have reorganised Section 4.1.3 to put more emphasis on the newly formed water masses, and to note that AABW and CDW are not spun up and should be evaluated with caution. The paragraphs on AABW and CDW have been moved to the end of the section and are now prefaced by the following paragraph (page 19, lines 1-3):

The remaining water masses, in the deep Southern Ocean, have much longer residence times and are therefore not fully spun up. Comparing their simulated properties is useful to assess model drift (see also Section 4.1.4), but they should be evaluated with caution.

16:11 Turner (2013) is refereeing to coupled models where the issue is more likely associated with the coupled forcing and mixed –layer depth, not due to the sea ice model itself. It is thus misleading to attribute this to the sea ice model, when it is rather the response of the sea ice to the climate model forcing. In the case of a standalone ocean model, where the surface forcing should act to restore the sea ice to that in ERA-I, such a lack of summer sea ice serious deficit in the models or bulk formula.

It should be noted that other standalone ocean/sea ice models forced with ERA-Interim show a similar underestimation of summer sea ice - see for example Kusahara et al. (2017). It has been suggested by Naud et al. (2014) that biases in ERA-Interim's summertime cloud cover lead to an overestimate of solar radiation reaching the ocean surface, which would contribute to low sea ice minima. This is discussed in Section 5 of our manuscript.

Nonetheless, we agree with the reviewer's point that these biases are not necessarily due to the sea ice model itself, but rather the atmospheric forcing and/or coupling. We have therefore revised the text as follows (page 23, lines 12-14):

All three of our simulations underestimate the sea ice minimum, which is a common bias seen in other standalone ocean/sea-ice models forced with ERA-Interim (Kusahara et al., 2017) as well as in fully coupled GCMs (Turner et al., 2013b).

We have also revised one sentence in Section 5 to remove the duplicated reference to Turner et al. (page 57, lines 22-23):

Simulated summer sea ice extent is too low in all three simulations, which exposes a larger area of the ocean to surface heating and drives increased summertime melting of ice shelf fronts.

16:16-19 Delete. It is not necessary to reiterate that the sea ice concentration is a time average (as it is for the inferred observations).

We have deleted these sentences as suggested.

16:21-22 This must happen because ERAI surface fluxes are attempting to restore sea ice to observations, so providing the mechanism them makes the variability self explanatory.

We agree this is the case, and have reworded the text as follows (page 23, lines 19-21):

However, they all display some of the observed interannual variability, such as the high in 2008 and the low in 2011, likely because observed sea ice cover is imprinted on the ERA-Interim atmospheric fields used to force the models.

16:23-26 The difference between high and low resolution is hardly significant in Fig 6b and not worthy of mention unless the bathymetry is causing the low ice extent in the first place. That ACC water is entering the Weddell Sea gyre from this direction would be unusual, and as a consequence be worth discussing as a model bias.

The spurious ACC excursion, due to smoother f/h contours in the low-resolution mesh, is the source of lower sea ice concentration along the Antarctic Peninsula compared to the high-resolution simulation. On a regional scale the differences are notable (Figure 6a), even if the total sea ice extent is not significantly affected (Figure 6b). These differences in sea ice concentration are also related to differences in other variables discussed later in the manuscript (Section 4.2.2 on sea ice thickness, and Section 4.3.8 on the Larsen Ice

Shelves). Therefore, we have decided to keep this passage in the text. However, we have added the word "spurious" (page 23, line 24) to stress that the ACC excursion is a model artefact.

16:27-34 Comparing sea ice concentrations is not valid unless there is a strong point to make. The observations are uncertain to 0.07 anyway and the concentration is consequently capped. The point about strong winter heat loss causing a sensible heat polynya and deep convection is supported by:
Goosse, H. and Fichefet, T.: Open-ocean convection and polynya formation in a large-scale ice-ocean model, Tellus A, 53, 94–11, doi:10.1034/j.1600-0870.2001.01061.x, 2001.
Timmermann, R. and Beckmann, A.: Parameterization of vertical mixing in the Weddell Sea, Ocean Model., 6, 83–100, doi:10.1016/S1463-5003(02)00061-6, 2004
Consequently, the authors could strengthen this argument, but it is really a sideline which should be avoided and allow a stronger continuity to the paper.

We acknowledge that observational uncertainty makes evaluation of sea ice concentrations difficult. However, there are no uncertainty ranges in model simulations (unless ensemble simulations are conducted), and the differences in sea ice concentration between MetROMS and FESOM are clear. The subsequent effects on surface fluxes could at least partially explain why MetROMS is susceptible to spurious deep convection in the Weddell Sea, while FESOM appears to be insensitive. Since this model bias ultimately shaped the design of our simulations (with respect to surface salinity restoring), we believe it is not a sideline to the paper, but rather a key difference between the models which is worth discussing.

We have reworded the text to focus on the differences between models, rather than evaluating them against observations (page 23, lines 27-30):

Sea ice concentrations throughout most of the ice pack are lower in MetROMS (approx. 0.94) than in both FESOM simulations (approx. 0.995). Observations from NSIDC fall in the middle (approx. 0.97), which is not significantly different from either model if observational uncertainty is considered. Nonetheless, this difference between the models influences the air-sea fluxes, which are modulated by the sea ice concentration.

We have already cited Timmermann and Beckmann (2004) in Section 3.3 where spurious deep convection is first discussed; we have now added a citation to Goosse and Fichefet (2001) in the same section (page 11, line 28). Neither of these references discusses the positive feedback between low sea ice concentration and convection, so we have not cited them in the current section.

17:7-11 Compared with coupled models the ice extents are pretty good, but they are still outside the decadal variability in the observations. Consequently the difference must be related to SST. This could be due to the path of the ACC (bathymetry or bulk formulae) or even due to the salinity restoration method being applied differently in the two models.

We agree that the path of the ACC may be playing a role here. Another possible explanation for differences in SST near the ice edge is mixed layer depth, which is deeper in MetROMS

(bringing more warm water to the surface to melt ice) than in FESOM (vice versa). We do not expect differences in the surface salinity restoration scheme between MetROMS and FESOM to have an effect, as the only difference between the two implementations is the depth of the surface layer over which the restoring is applied.

We have updated the text as follows (page 24, lines 7-9):

While the general pattern of both models' September sea ice agrees with observations, the northern edge of the ice pack is too far south in MetROMS and too far north in FESOM, which is possibly related to differences in mixed layer depth (Section 4.1.2) or in the path of the ACC.

17: 15 Ice thickness is indeed a good means to inter-compare processes between models. I would be far more concerned with the thick ice in FESOM (and MetROMS) sitting directly in front of the Filchner_Ronne ice shelf. This should be a region of thin ice in JJA due to the winter latent heat polynya, which will not form in the model due to the thick ice. Consequently the Weddell sea ice formation and drift would be abnormal as is evidenced by the ice extent and thickness. On the other hand the Ross Ice-shelf polynya looks fine.

Observations suggest that winter sea ice is thinner in front of the of the Ronne Ice Shelf and thicker in front of the Filchner Ice Shelf (see for example Figure 3 of Holland et al. 2014, doi:10.1175/JCLI-D-13-00301.1). Our simulations agree with this pattern. The Ronne Polynya has already been discussed in the text (page 26, lines 16-17). We have added a sentence discussing sea ice thickness in front of the Filchner Ice Shelf (page 26, lines 17-18):

Thicker ice is present directly in front of the Filchner Ice Shelf, especially in FESOM.

17:29 This launch into the cavity melt is too abrupt. It would be better to start with how this is assess observationally and the uncertainties (the Deporter and Rignot estimates are easily inside the error bounds and no comparison is required).

We have reorganised this section, as quoted in our response to the next comment.

17:29-18:34 The intriguing aspect of figure 8 is that both models deviate from Rignot in the same direction. We would expect differences associated with the individual model resolution (vertical and horizontal), bulk formulae of momentum at the continental shelf break, basal melt parameterisations (and bathymetry). A description of possible sources of discrepancies is required before launching into the individual shelf regions. For both models to deviate from Rignot in the same way either says that there are regional specific biases (e.g. lack of tides in models), ERAI is regionally biased from an unknown reality, or that Rignot is methodology is flawed. The latter could be tested by determining if the entire pattern changes with the Depoorter results. A comparison with Table 3 of Mathiot et al. (2017) might also be instructive.

We have reorganised Section 4.3 with the aim of (1) discussing observations prior to evaluating simulated melt rates, and (2) expanding our discussion of potential sources of biases in the simulations. The revised paragraphs are as follows (page 30, lines 2-29):

Basal melting of Antarctic ice shelves comprises a substantial source of freshwater entering the Southern Ocean. Rignot et al. (2013) estimate, based on observations for the period 2003-2008, that total ice shelf basal mass loss occurs at a rate of 1325 +/- 235 Gt/y. This estimate is prone to errors in the calculation of basal melting at ice shelf fronts (where separating basal melting from calving is not straightforward) and relies on atmospheric reanalyses which in turn have limited observations from which to downscale. Another observation-based estimate, by Depoorter et al. (2013), is similar at 1454 +/- 174 Gt/y.

All three model simulations underestimate total ice shelf basal mass loss with respect to these observations, roughly by a factor of two. The simulated mass loss, averaged over 2002-2016, is 642 Gt/y for MetROMS, 586 Gt/y for low-resolution FESOM, and 739 Gt/y for high-resolution FESOM. A closer examination of individual ice shelves shows that the bias in our simulations is a regional phenomenon. Table 1 compares simulated basal mass loss to Rignot et al.'s estimates for 25 ice shelves, organised into eight regions. The model biases are summarised in Figure 9, which plots the difference between the simulated values and Rignot et al.'s central estimates, as well as the uncertainty range, for each ice shelf. All three simulations underestimate mass loss for ice shelves in the Amundsen Sea, Bellingshausen Sea, and Australian Sector. These three regions include many warm-cavity ice shelves which, despite their small areas, exhibit substantial basal mass loss in observations. Ice shelves in the remaining five regions generally show either agreement between our model simulations and Rignot et al.'s observations, or an overestimation of mass loss by the models (with the main exception being MetROMS' underestimation of the Filchner-Ronne Ice Shelf). The following sections will analyse these eight regions in more detail.

While biases in ice shelf mass loss are largely region-specific, several overarching factors are worth mentioning here. First, neither MetROMS nor FESOM considers the effects of tides. Since the heat and salt transfer coefficients in both models depend on ocean velocity adjacent to the ice shelf base, tidal currents would be expected to increase melt rates in all ice shelf cavities. Tides also cause enhanced vertical mixing, which further influences melt rates (Gwyther et al., 2016). Next, insufficient horizontal resolution is likely to cause an underestimation of eddy transport of warm CDW onto the continental shelf; this phenomenon is discussed more fully in Section 4.3.6. Finally, biases in the ERA-Interim atmospheric forcing could affect water mass properties and therefore ice shelf melt rates; this is difficult to test due to a lack of observations around Antarctica. Note also that the area of a given ice shelf in model simulations does not necessarily agree with the area used in Rignot et al.'s calculations, particularly for small ice shelves which are not well resolved by the models. Such disagreements may bias our comparison. However, a comparison of area-averaged basal melt rates rather than area-integrated basal mass loss (not shown) shows essentially the same biases. Furthermore, a comparison with the mass loss estimates of Depoorter et al. (2013) yields a similar pattern of biases.

We decided not to compare our results to Mathiot et al. (2017) within this manuscript, as doing so would compel a comparison with the many other published circumpolar ocean/ice-shelf models: Dinniman et al. (2015), doi:10.1175/JCLI-D-14-00374.1; Kusahara et al. (2013), doi:10.1002/jgrc.20166; Timmermann et al. (2012), doi:10.3189/2012AoG60A156; and Beckmann et al. (1999), doi:10.1029/1999JC900194, to name a few. Such a comparison would be beyond the scope of this paper, which is already substantial. However, we have examined the results of Mathiot et al. (2017) with interest, and informally compared its Table 3 to our results for our own knowledge.

19-25 The descriptions of ice shelves by region This type of analysis lends itself to repetition and long-winded descriptions. Instead I suggest that the analysis focus on specific ice shelves that show particular characteristics in Figure 8. For example Pine Island shows all the models closely clustered. The analysis should then show that the actual behavior of the models is quite different. Another might be the Ross where the melt rates are quite different and there is resolution sensitivity. Another might be, say, West, which is a poorly resolved small (shallow draft) ice shelf.

We seriously considered reorganising the manuscript as the reviewer suggests, but ultimately decided that a comprehensive analysis was preferable to case studies of specific ice shelves. The choice of specific ice shelves is not obvious, as all eight regions in our analysis yield useful insights into model behaviour. The question of which ice shelves are the most worthy of analysis therefore highly depends on personal interest and expertise.

We expect the bulk of the readership of this manuscript to consist of ice-shelf/ocean researchers, who rarely hold a purely circumpolar perspective. More often such researchers specialise in particular regions or particular ice shelves. For example, the authors of this paper have collectively specialised in the Amery Ice Shelf, the Fimbul Ice Shelf, the Filchner-Ronne Ice Shelf, and multiple smaller ice shelves of the Australian sector and the Amundsen Sea. In each case, the authors feel it is essential for this manuscript to show the model behaviour in the given region, because seeing the results has informed their own research going forward.

In order for this paper to be as useful as possible to the target audience, we believe that the results should be shown for all regions of Antarctica. We understand that the manuscript is substantial in length and content, but this is often the case for model intercomparison papers, and is a format to which GMD is well-suited. Regardless, most readers will skip ahead to the sections about ice shelves in which they are most interested.

Finally, we do not expect that a re-organisation of the ice shelf analysis by process (i.e. the reasons for biases in simulated mass loss compared to Rignot et al.) would be any shorter, especially since most of the ice shelves in our simulations are affected by multiple such processes.

Keep the figures presented internally consistent, the style used for Figure 11 is good with a cross-section of water properties (including of-shelf) to reveal the bathymetry issues, perhaps showing the vertical overturning stream function and horizontal barotropic stream

function. The small vertically integrated velocity vectors are not easy to interpret and if this approach is used a schematic representation of circulation may be better.

During our initial analysis, we plotted all seven variables (from those shown in Figures 11-18) for all eight regions. However, each region exhibits different behaviour, which is best highlighted by a different combination of variables. In the interests of keeping the figures as concise as possible while still conveying the necessary information, we decided that a customised approach was preferable to choosing a standardised combination of variables. Cross-sections of temperature and salinity (as we show for the Fimbul and Ross Ice Shelves) can be valuable, but a latitude-longitude plot of bottom water temperature and salinity sometimes conveys more information, depending on the region.

We find velocity vectors to be more intuitive than streamfunctions, particularly since our audience is likely to be quite interdisciplinary, rather than purely oceanographic. We considered a schematic representation of circulation (i.e. overlaying the coloured absolute velocity values with manually drawn curved arrows) but some co-authors felt strongly that this represented a loss of information and should be avoided. Instead, we have remade Figures 11-18 with a higher DPI, which makes the vectors crisper, and details of circulation can be more easily examined when zooming into the electronic version of the manuscript.

27:20-30:19 Discussion It is certainly true that all models, even ERAI show a cloud bias associated with cyclones, see:
Bodas-Salcedo, A., T. Andrews, A. V. Karmalkar, and M. A. Ringer (2016), Cloud liquid water path and radiative feedbacks over the Southern Ocean, Geophys. Res. Lett., 43, 10,938–10,946, doi:10.1002/2016GL070770.
Williams, K. D., and Coauthors, 2013: The Transpose-AMIP II experiment and its application to the understanding of Southern Ocean cloud biases in climate models. J. Climate, 26, 3258–3274, doi:https://doi.org/10.1175/JCLI-D-12-00429.1.

We have added these references (page 57, line 30).

Include reference to (Graham et al., 2016) when discussing resolution.
Graham, JA, Dinniman, MS and Klinck, JM : Impact of model resolution for on-shelf heat transport along the West Antarctic Peninsula, J. Geophys. Res., 121, 7880-7897, doi: 10.1002/2016JC011875, 2016.

We have added a short discussion of Graham et al.'s results to Section 4.3.7 on the Bellingshausen Sea (page 52, lines 12-15):

The sensitivity of Bellingshausen Sea temperatures to model resolution was further demonstrated by Graham et al. (2016), whose ROMS simulations exhibited greater onshore heat transport at 1.5 km resolution compared to 4 km resolution, due to increased eddy activity.

Refer to Mathiot et al., 2017 to compare with a pure z-level model.

We have added the following sentence to expand our discussion of the benefits and drawbacks of different vertical coordinate systems (page 59, lines 14-16):

On the other hand, z-coordinate cavities are not susceptible to pressure gradient errors at the ice shelf front, and do not introduce time step limitations at the grounding line (Mathiot et al., 2017).

Some discussion of the variability is required. Is the melt rate dominated by a few years of periodic high melt, or is the interannual variability small (probably varies according to distance from the shelf break and strength of the ocean baratropic circulation on shelf

We have added a paragraph to Section 4.3 analysing interannual variability in ice shelf mass loss (page 31, lines 3-10):

Interannual variability in ice shelf melting is relatively small. In all three simulations, the standard deviation in annually averaged mass loss from individual ice shelves is typically 10-20% of their 2002-2016 mean. Furthermore, the mean and median of the annually averaged values are typically very similar (within 10% of each other) which indicates that the long-term average is not skewed by a few years of unusually high or low melt. The main exceptions are the Larsen C and D Ice Shelves in FESOM, which experience large spikes in mass loss in some summers but not others. This behaviour is tied to the sea ice cover, as FESOM occasionally has ice-free summers along the peninsula, allowing warmer AASW to develop. In MetROMS, the Shackleton Ice Shelf shows the highest interannual variability in mass loss. Here a few cells on the western edge of the ice shelf are undercut by the Australian Coastal Current, bringing periodic pulses of high melt.

We would like to thank Reviewer 1 for their prompt and thoughtful review, which led to numerous improvements in the manuscript.

**Reviewer 2**

Reviewer: Xylar Asay-Davis

I wish my name to be relayed to the authors, as I do not support the practice of anonymous review.

General comments:

This paper presents a comparison of two sea ice/ocean models with ice-shelf cavities, MetROMS and FESOM. Biases are analyzed in detail and across many regions where observations are available, including a discussion of directions for future model improvements that might reduce these biases. Where direct observations are not available, (e.g. for flow patterns under ice shelves), a comparison is made between models and biases are inferred from more indirect observations (e.g. the observed locations of inflow into and outflow from ice-shelf cavities). The paper is very well written, very clearly organized and

makes for a compelling read. I feel it is nearly ready for publication, requiring only a few minor changes.

Perhaps the most significant change I would wish for the authors to consider is the addition of a figure of the rate of sea-ice production from each model, possibly compared with satellite-derived estimates (see my specific comment below). While the figure of mixed-layer depth serves as a proxy for this quantity, I feel like a figure showing sea-ice production would more directly get at the source of biases seen in both models that are inferred to come from too much or too little (or incorrectly located) sea-ice production in different regions.

We have added this figure to the manuscript (see our response to the specific comment about sea ice formation below).

As an aside, given that it is not (any longer) a requirement from GMD to put the figures and tables at the end of the manuscript, as a reviewer I would have liked to have the figures interspersed with the text for easier reviewing and reference. This is just something to keep in mind for any future submissions to a Copernicus journal.

We agree this would be more readable, and we have rearranged the figures in the resubmitted version of the manuscript. Each figure is presented on its own page following the section which first references the figure (note that this leads to more white space between sections, and therefore an increase in the number of pages). We will also make use of this option in future submissions.

Specific comments:

p. 2 l. 3 "The rate of retreat of much of the AIS will be governed by the ocean." While I agree that the ocean will almost certainly play an important role in AIS retreat, there are a couple of reasons I would suggest toning down this statement. First, internal ice dynamics and topography may govern the rate of retreat at least as much if not more than the ocean, with the ocean acting more as a trigger mechanism or an intermittent forcing. Second, we should not discount the potentially important role that atmospheric forcing will likely play in this process (e.g. via surface melting and potential hydrofracture).

This is a good point, and we have changed this phrase to "The ocean is an important driver of AIS retreat" (page 2, line 3).

p. 3 l. 27-29 "how many features of fluid flow can be represented by a mesh of a certain spacing" and "the number of features resolved". Flow features are not easily countable in the way this wording implies, and increasing resolution does not typically lead to a binary transition from unresolved to resolved. Instead, there is a messy transition from unresolved through partially resolved to fully resolved. I would suggest something like, "smallest flow features that are captured by a mesh of a certain spacing" and "the smallest resolved feature" for these.

We have reworded the phrases as suggested (page 3, line 31 and page 4, line 1).

p. 4 l. 4-5 "These differences... dominates" My experience with MISOMIP and my own pan-Antarctic modeling is that eddy-permitting and eddy-resolving simulations do *not* necessarily behave more similarly, and other modeling choices still play an important (if not dominant role) even when eddies are included. For example, even fairly subtle differences in how topography is represented can lead to changes in how eddies are shed or small-scale currents interact with the topography.

We agree that this relationship may not be as simple as the original text perhaps made it seem. To avoid confusion, we have removed the quoted sentence as well as the sentence following.

p. 4 l. 21-22 and Fig. 5 "Bottom nodes are allowed to deviate from the standard z-levels in order to match the given bathymetry" This is not obvious in Fig. 5. If this is a plotting artifact in Fig. 5, it would be best to fix that so the true nature of how FESOM represents bathymetry is shown in the figure. If not, it is unclear why there are full-cell jumps in the bathymetry in Fig. 5, given that FESOM should have some kind of equivalent of the partial-cell methods used in other z-level models.

We would like to thank Dr Asay-Davis for noticing this. It turns out that the partial-cell option in the FESOM mesh machinery was not activated for this version of the mesh. Bathymetry in z-coordinate regions is instead represented in a classical stepwise fashion. Therefore, we have removed the quoted sentence.

p. 4 l. 24-25 "extremely high vertical resolution." I would suggest avoiding subjective phrases like this that include an implicit value judgement. What might seem today like "extremely high" vertical resolution is likely to become closer to standard resolution in the not-too-distant future. We frequently run global ocean simulations with the Model for Prediction Across Scales Ocean (MPAS-O) including 1 meter vertical resolution in the upper mixed layer.

This is a good point, and we have changed "extremely high vertical resolution" to "enhanced vertical resolution" (page 4, lines 25-26).

p. 4 l. 26 "with 30 barotropic timesteps for each baroclinic" This is largely an aside for your future work, not a request for a change in the paper. Did you experiment at all with fewer barotropic time steps per baroclinic? If vertical resolution is controlling the time step, it should follow that the barotropic time step would not be strongly affected and fewer subcycles might be possible.

We have not experimented with the ratio between barotropic and baroclinic time steps, but will consider it for future work.

p. 6 l. 10-11 "very minimal spurious sea ice formation" maybe remove "very" (since it sounds kind of subjective)

We have made this change (page 7, lines 16-17).

p. 6 l. 10-12 I'm kind of confused by this sentence. My understanding was that that the flux-limited advection scheme in Naughten et al. 2017 reduced the spurious sea-ice formation. Do you perhaps mean "comparable to" instead of "compared to"? Then, I would understand a bit better.

Indeed, the flux limiters in Naughten et al. 2017 essentially eliminate spurious supercooling, and are treated as a baseline to which other simulations are compared. We understand how this sentence may be confusing, and have changed "compared" to "comparable" as suggested (page 7, line 17).

p. 7 l. 9-11 It is a little unclear what "no significant impacts on Weddell Sea convection" means in this context. I take it to mean that KPP appears to work just as well as Pacanowski-Philander, at least over the 5 years. If that is the case maybe a comment is warranted here to the effect that this deserves further investigation. I'm a little unclear on what you (or I as the reader) should take from this.

We have replaced this sentence with the following (page 8, lines 15-17):

At least on the 5-year timescale, hydrography in the offshore Weddell Sea was very similar between KPP and the modified Pacanowski-Philander scheme. It is possible that longer simulations would show more divergence, and this warrants further investigation.

p. 8 l. 5-6: The values of $u^*$ need units (presumably m/s)

We have added these units (page 9, line 4).

p. 8 eq. (4): 530, 10^-3 and 10^-8 need units, since they appear to all be dimensional.

We have added units of $m^{-2}s$, m/s, and $s^{-1}$ respectively.

p. 10 l. 6-8: "which are not interpolated in time...as they represent total fluxes…" This is a fine approach but there would be ways to interpolate these data in time while preserving the 12-hourly mean.

We agree that a step change is not the only possible approach, however it is the current design of FESOM and so we have left the text unchanged.

p. 10 l. 13: "additional surface freshwater flux representing iceberg melt" Could you give a description in a sentence or 2 of what the characteristics are of this climatology? It seems like it comes from an iceberg model, rather than from observations, which is perfectly reasonable but probably deserves a mention.

We have updated this paragraph as follows (page 11, lines 20-25):

To account for the influence of iceberg calving on the Southern Ocean freshwater budget, both models are forced with an additional surface freshwater flux representing iceberg melt. For this field we use the output of Martin and Adcroft (2010), who modelled icebergs as interactive Lagrangian particles in the ocean component of a GCM simulation. The initial sizes of icebergs at calving fronts were determined from a statistical distribution constrained by observations. Martin and Adcroft's monthly climatology of iceberg melt is interpolated to each timestep in our simulations, and repeated annually. River runoff from other continents is not considered.

p. 11 l. 6-7 "...taken from the AVISO climatology...which is a single time record." If I understand right, you use the annual mean rather than the monthly climatology? (Near as I can tell, both are available from AVISO.)

We have changed "the AVISO climatology" to "the AVISO annual mean climatology" to clarify this (page 12, line 20).

p. 11 l. 9 "in y" I would probably change this to "in latitude" even though I understand that "y" is not exactly the same as latitude over the whole domain. At the northern boundary, they are presumably the same and I think that would be clearer to the reader.

We agree this is clearer, and have made this change (page 12, line 23).

p. 14 l. 3-4 "In both models RSBW temperatures more or less agree with observations...". This seems a little generous to me. MetROMS is clearly missing some colder temperatures, while FESOM reaches temperatures that are significantly colder than observed in WOCE. I guess that is what the "more or less" is meant to cover.

We have revised the text as follows to more fully account for the model-data disagreements (page 19, lines 18-20):

The colder varieties of RSBW are absent in MetROMS, while FESOM reaches temperatures which are significantly colder than WOCE observations. In both models, simulated WSBW is too warm.

p. 14 l. 16 LSSW isn't really analyzed here. Is this because Fig. 4 doesn't really have anything to say about this water mass?

We have added a sentence to discuss LSSW (page 18, lines 10-11):

LSSW shows similar properties in all three simulations, with minimum salinities around 33.75 psu.

p. 15 l. 24: "In FESOM, AAIW is slightly better preserved at low resolution than at high resolution." Do you care to speculate on why this is?

We have added the following sentence to this paragraph (page 21, lines 21-23):

This agrees with the results of Marchesiello et al. (2009) showing that in non-eddy-resolving regimes, spurious diapycnal mixing tends to increase as resolution is refined.

p. 20 l. 11: "extremely high resolution." Again, I would recommend a less subjective wording.

We have changed "extremely high resolution" to "higher resolution" (page 37, line 11).

p. 21 l. 4 and Fig. 11a: I'm not sure what can be one but I found the figure to be too small to clearly see the features that are being described. When I zoomed in to 400% in the pdf viewer, the figure was kind of pixelated, meaning this didn't really help.

To address this issue, we have remade all of the ice shelf maps (Figures 11-18) using a higher DPI. This makes the details of the figures much clearer when zooming into the electronic version of the manuscript. Individual mesh elements can now be discerned even for high-resolution FESOM, and the velocity vectors are much crisper. Note that vector graphics are not practical with FESOM, as every triangular element adds a new vector patch, leading to unworkably large file sizes.

p. 22 l. 15: "steeper ice draft" I would also suggest that better resolved currents may be the reason.

We have updated this sentence as follows (page 41, lines 27-28):

Increased velocities near the back of the cavity, possibly due to the steeper ice draft or better resolved currents, also have an effect.

p. 25 l. 2: "and this process requires resolutions of 1 km or less" This is stated in a couple of places. My experience is that model properties improve significantly even at eddy-permitting resolution (~2-4 km) even when the largest eddies are not fully resolved. My point is that eddy transport is not typically absent in eddy-permitting simulations, it is just diminished from what it would be at higher resolution.

We have revised this passage as follows (page 49, lines 10-15):

Eddy transport of heat is also an important factor for cross-shelf CDW exchange. In order to fully resolve this process, resolutions of 1 km or finer are required (St-Laurent et al., 2013), which none of our simulations have. A partial representation of eddy transport would be expected from eddy-permitting simulations (approx. 2-4 km on the Antarctic continental shelf), which high-resolution FESOM attains in the Amundsen and Bellingshausen Seas. The latitude-dependence of the Rossby radius of deformation means that eddies are much smaller, and therefore more computationally expensive to resolve, in the polar regions compared to the tropics and the mid-latitudes.

p. 27 l. 27-28: "First, the location and rate of sea ice formation impacts the properties of shelf water masses flowing into the ice shelf cavities." You did a very thorough job of plotting a lot

of fields from the 2 models. If there is one field I wished you'd included, it is the rate of sea-ice formation, perhaps comparing with a satellite-derived data set such as Tamura et al. (2016). I will not insist that you add such a plot but I would appreciate it if you would consider it at least.

This is a good idea which we had not previously considered, and we would like to thank Dr Asay-Davis for suggesting it. We have added a new section which compares simulated sea ice production to the dataset of Tamura et al. (page 28, lines 1-20), with an accompanying figure which is reproduced below:

**4.2.3 Sea ice production**

[revised manuscript text omitted]

p. 29 l. 10-12: "Therefore, in the future it may be worthwhile to experiment with different topographic smoothing methods, which may uncover options to minimise the trade-off between numerical stability and geometric accuracy." I would suggest adding to this that it might be worth investigating other numerical methods for computing the horizontal pressure-gradient force (HPGF) in FESOM, since this is an area of active research (Engwirda, Kelley, and Marshall 2017). I would have more confidence that improvements in the HPGF would lead to less topographic smoothing than that a better smoothing algorithm can solve the problem on its own.

This sounds like a promising direction for future development. We have added the following sentence (page 58, lines 27-29):

Another worthwhile approach would be to investigate alternative methods for the calculation of the horizontal pressure gradient force, such as that of Engwirda et al. (2017), which may permit more steeply sloping layers and therefore less topographic smoothing.

p. 29 l. 23-24: "None of our simulations resolve eddies on the Antarctic continental shelf, which would require resolution of approximately 1 km (St-Laurent et al., 2013)." See my previous comment about 1 km resolution. I think the higher resolution FESOM simulation is probably at least eddy permitting. This is suggested at least by some of the features shown in the zoomed-in velocity plots. If this is the case, it might be worth mentioning. If not, it might be worth mentioning that none of the simulations is even eddy permitting.

We have revised this sentence as follows (page 59, lines 5-7):

None of our simulations fully resolve eddies on the Antarctic continental shelf, which would require resolution of approximately 1 km (St-Laurent et al., 2013), although high-resolution FESOM is eddy-permitting (2-4 km) in the Amundsen and Bellingshausen Seas.

p. 30 l. 18-19: "Furthermore, alternative parameterisations of ice shelf basal melt are being explored by the community (Jenkins, 2011), which may provide valuable intercomparisons with the three-equation parameterisation in the future." I would suggest including Jenkins (2016) here. I think this work is more likely to lead to an alternative to the three equations than the plume-model approach as in Jenkins (2011).

We are interested to hear this prediction, and have added the citation to Jenkins 2016 (page 60, lines 3-4).

p. 30 l. 28-29: "Sea ice in both MetROMS and FESOM mostly agrees with observations..." Can you be more specific about which fields were compared with observations? For example, I don't think the rates and locations of sea-ice production are likely to agree with satellite-derived estimates (see my suggestion for a figure above), as you discuss in the context of mixed-layer depth in the ocean.

We have revised and expanded this passage as follows (page 60, lines 13-17):

Sea ice extent in both MetROMS and FESOM mostly agrees with observations, although both models underestimate the summer sea ice minimum, and MetROMS requires surface salinity restoring to prevent a spurious open-ocean polynya from forming in the Weddell Sea. Sea ice production is too strong in the Ross and Weddell Seas compared to observations, and too weak in the small coastal polynyas of the Australian Sector.

p. 30 l. 31: "In the interior Southern Ocean and the ACC, FESOM has an advantage due to its vertical coordinate system." I think it would be worth restating that FESOM's vertical coordinate is z-level (and perhaps also that MetROMS' is terrain-following) in this region.

We have revised this sentence as follows (page 60, lines 17-19):

In the interior Southern Ocean and the ACC, FESOM has an advantage due to its vertical coordinate system, which is locally z-coordinate compared to MetROMS' terrain-following coordinate which covers the entire domain.

p. 31 l. 1: "...more reliable atmospheric forcing..." I think this needs some clarification. What does this mean to you? I realize this has been covered in the discussion but it's worth summarizing in at least a little more detail here.

We have added the following sentence to the conclusion section (page 60, lines 19-21):

Our results are dependent on the ERA-Interim atmospheric reanalysis and are influenced by any biases it may contain over the Southern Ocean, including its known underestimation of summertime cloud cover which leads to excessive sea ice melt.

Furthermore, we have changed "more reliable atmospheric forcing" to "alternative atmospheric forcing datasets" in the last sentence of the conclusions (page 60, line 22).

Typographic and grammatical corrections:

p. 1 l. 15-16: In my experience, it is customary to refer to a model's "terrain-following coordinate" or "z-coordinate" (both singular). Plural would be appropriate if we were referring to coordinates in multiple dimensions (e.g. spherical coordinates).

We have made this change (page 1, line 15).

p. 4 l. 26-29 (and possibly elsewhere) I would suggest using only "time step" and not "timestep". You use both in this paragraph.

We have changed all instances of "timestep" in the manuscript to "time step".

p. 8 l. 1, 4, 12, 15: There is an incorrect new paragraph on each of these lines causing an indentation.

This was due to an equation formatting issue which we have corrected.

p. 21 l. 26: "which has the one of the deepest ice shelf drafts in Antarctica" an extra "the"

We have fixed this typo (page 41, line 5).

In particular, please note that for your paper, the following requirements have not been met in the Discussions paper:
• "The main paper must give the model name and version number (or other unique identifier) in the title."
• "All papers must include a section, at the end of the paper, entitled 'Code availability'. Here, either instructions for obtaining the code, or the reasons why the code is not available should be clearly stated. It is preferred for the code to be uploaded as a supplement or to be made available at a data repository with an associated DOI (digital object identifier) for the exact model version described in the paper. Alternatively, for established models, there may be an existing means of accessing the code through a particular system. In this case, there must exist a means of permanently accessing the precise model version described in the paper. In some cases, authors may prefer to put models on their own website, or to act as a point of contact for obtaining the code. Given the impermanence of websites and email addresses, this is not encouraged, and authors should consider improving the availability with a more permanent arrangement. After the paper is accepted the model archive should be updated to include a link to the GMD paper."

As evaluation results also depend on the model and the model version, please provide the model names and their version numbers in the title of your article. Additionally, please ensure that the exact model versions, the evaluation was performed with, are permanently archived (best in a permanent archive providing a DOI (e.g. Zenodo)). Please make the model code also available to the public. If legal issues prevent this, please state the explicit reasons in the code availability section.

Yours,
Astrid Kerkweg

We have renamed the manuscript "Intercomparison of Antarctic ice shelf, ocean, and sea ice interactions simulated by MetROMS-iceshelf and FESOM 1.4". Note that since our configuration of MetROMS is the only one to include Antarctic ice shelves, "MetROMS-iceshelf" is a unique identifier and does not require a version number.

We have archived the exact model versions using Zenodo as suggested, and also included links to the maintained versions of the code in the Code Availability section (page 60, lines 25-29):

**Code availability.** The source codes used for the simulations described here are archived at doi:10.5281/zenodo.1157229 for MetROMS-iceshelf and doi:10.5281/zenodo.1157227 for FESOM 1.4. These repositories also include the grid/mesh files and the configuration files. The atmospheric forcing, initial conditions, and ROMS northern boundary conditions can be obtained from the authors upon request. Additionally, the maintained version of MetROMS-iceshelf is publicly available at https://github.com/knaughten/metroms_iceshelf, and the maintained version of FESOM is available at

https://swrepo1.awi.de/svn/awi-cm/trunk following registration at https://swrepo1.awi.de/projects/fesom/.

**Other changes**

It was discovered that the standard deviations and trends in Drake Passage transport had been calculated over 5-day averages, not annual averages as stated in the text. The calculations were repeated with annual averages and the text was updated as follows (page 13, lines 12-18):

This time-averaged Drake Passage transport, including the standard deviation in annual averages, is 126.8 +/- 3.4 Sv in MetROMS, 158.6 +/- 2.8 Sv in low-resolution FESOM, and 152.6 +/- 3.1 Sv in high-resolution FESOM. Compared to the observations of Donohue et al. (2016), the values from all three of our simulations are too low, especially in MetROMS. This occurs despite MetROMS' stronger surface stress than in FESOM (Section 2.7). Additionally, the MetROMS and low-resolution FESOM simulations exhibit downward trends in Drake Passage transport over 2002-2016, which are statistically significant at the 95% level: -0.28 Sv/y in MetROMS and -0.17 Sv/y in low-resolution FESOM.

All other changes to the manuscript are typographical corrections or minor wording changes.